



# On the uncertainty of anthropogenic aromatic VOC emissions: evaluation and sensitivity analysis

Kevin Oliveira[1], Marc Guevara[1], Oriol Jorba[1], Hervé Petetin[1], Dene Bowdalo[1], Carles Tena[1], Gilbert Montané Pinto[1], Franco López[1], and Carlos Pérez García-Pando[1,2]

[1]Barcelona Supercomputing Center (BSC), Barcelona, Spain
[2]ICREA, Catalan Institution for Research and Advanced Studies, Barcelona, 08010, Spain

**Correspondence:** Kevin Oliveira (kevin.deoliveira@bsc.es)

**Abstract.** Volatile organic compounds (VOCs) significantly impact air quality and atmospheric chemistry, influencing ozone formation and secondary organic aerosol production. Despite their importance, the uncertainties associated with representing VOCs in atmospheric emission inventories are considerable. This work presents a spatiotemporal assessment and evaluation of benzene, toluene, and xylene (BTX) emissions and concentrations in Spain by combining bottom-up emissions, air qual-
ity modelling techniques and ground-based observations. The emissions produced by HERMESv3 were used as input to the MONARCH model to simulate surface concentrations across Spain. Comparing modelled and observed levels revealed uncertainty in the anthropogenic emissions, which were further explored through sensitivity tests. The largest levels of observed benzene and xylene were found in industrial sites near coke ovens, refineries and car manufacturing facilities, where the modelling results show large underestimations. Official emissions reported for these facilities were replaced by alternative estimates,
allowing to heterogeneously improve the model's performance, highlighting that uncertainties representing industrial emission processes remain. For toluene, consistent overestimations in background stations were mainly related to uncertainties in the spatial disaggregation of emissions from industrial use solvent activities, mainly wood paint applications. Observed benzene levels in Barcelona's urban traffic areas were five times larger than the ones observed in Madrid. MONARCH failed to reproduce the observed gradient between the two cities due to uncertainties in estimating emissions from motorcycles and mopeds.
Our results are constrained by the spatial and temporal coverage of available BTX observations, posing a key challenge in evaluating the spatial distribution of modelled levels and associated emissions.

## 1   Introduction

Volatile organic compounds (VOCs) significantly contribute to air pollution and pose serious health hazards to humans. These compounds originate from diverse sources, resulting from anthropogenic and biogenic activities (Kansal, 2009). They can also
be produced by the oxidation of other VOCs, this secondary formation being predominant for some VOCs (e.g. formaldehyde) (Parrish et al., 2012; Luecken et al., 2012). Anthropogenic sources include various human-driven activities, such as solvent use, traffic and fuel evaporation, industrial emissions, and biomass burning (Monks et al., 2015; Kansal, 2009). Within urban areas, VOC emissions are predominantly influenced by human activities, with vehicular emissions representing over 60 % of VOCs





in European urban areas (Borbon et al., 2018). Meanwhile, biogenic VOCs (BVOCs) also play a crucial role in atmospheric

chemistry, as they emit a larger fraction of total VOCs at a global scale and exhibit higher chemical reactivity compared
to many anthropogenic VOCs (Guenther et al., 2006). Additionally, it is important to note that human-induced atmospheric
changes increase oxidant levels which can also boost natural aerosol production (Kanakidou et al., 2000).

One essential aspect of VOCs is their contribution to tropospheric chemistry, as they are major precursors for ozone ($O_3$)
(Atkinson and Arey, 2003; Carter, 1990) and/or secondary organic aerosol (SOA) formation (Chen et al., 2022; Ziemann and

Atkinson, 2012). In Spain, for specific areas and conditions, VOCs contribute to high $O_3$ concentrations episodes (In't Veld
et al., 2024; Querol et al., 2018, 2017; Castell et al., 2008). For example, in Barcelona, In't Veld et al. (2024) estimated
that anthropogenic VOCs were significant contributors to $O_3$ formation, accounting for 38 and 49 % of the measured Ozone
Formation Potential (OFP) during winter and summer, respectively. Also, Oliveira et al. (2023) showed that in Spain, toluene
and benzene are in the top 5 main species contributing to OFP, while benzene, although having a low reactivity, is in the top

20 species contributing to OFP. Moreover, they play a key role in the formation of SOA(Baltensperger et al., 2005; Cui et al.,
2022; Sun et al., 2016; Jookjantra et al., 2022), which significantly contribute to fine particulate matter (PM2.5) (Srivastava
et al., 2022; Zhang et al., 2019; Ziemann and Atkinson, 2012). Huang et al. (2014) found that SOA contributes about 30 to 77
% of PM2.5 mass concentrations.

While some VOCs may not be associated with acute health impacts, they can still lead to chronic health risks (Shuai et al.,

2018; Alford and Kumar, 2021). Notably, aromatic compounds, such as benzene, toluene, and xylene (commonly referred
to as BTX), are of particular concern (Filley et al., 2004; Tagiyeva and Sheikh, 2014; Niaz et al., 2015; Ling et al., 2023),
benzene being classified as a Group A carcinogenic compound by the US Environmental Protection Agency (Bayliss et al.,
1998). This study focuses on BTX because (1) they are continuously measured, (2) they represent a substantial part of the total
anthropogenic emissions - more than 20% at global scale according to Yan et al. (2019), possibly more in urban areas (Bates

et al., 2021), (3) they are important precursors of $O_3$ and SOA.

Emission inventories are critical inputs for air quality (AQ) management and modelling. Despite the importance of VOCs,
as previously stated, among all the air pollutants reported by emission inventories (e.g. NOx, SOx, CO, $NH_3$, PM10, PM2.5),
VOCs are typically associated with the highest uncertainty. For example, when considering the combined uncertainty of both
emission factors (EF) and activity data, VOC emissions show uncertainties between 15.6 to 490 % across the different sectors,

which is more than for other species, such as for NOx emissions which report uncertainties between 3.6 to 175 % (MITERD,
2022). This is mainly due to limited efforts in updating the EF and limited information on the activity data of some key sectors
(e.g., use of solvents).

Chemical transport models (CTMs) are a powerful tool for assessing and forecasting atmospheric pollutant concentrations
(EC, 2008). They support the design of effective AQ control strategies by assessing the potential impacts of specific emission

reduction scenarios (Agency, 2011). When evaluated against observed pollutant concentrations, they typically provide key
insights on the validity of emission inventories, despite the other overlapping uncertainty sources. CTMs are most commonly
evaluated on the main criteria pollutants for which more observations are available (Badia and Jorba, 2015; Georgiou et al.,
2022; Skoulidou et al., 2021), but an important gap persists on the VOCs (She et al., 2023). As an example, Air Quality Expert





Group (2020) recently highlighted the absence of model-observation evaluations of VOCs over the UK in the literature. This
is due to several reasons; first, the models use simplified chemical mechanisms which group (or 'lump') the large number of
individual VOCs species to more generalised families based on their known reactions or number of carbon bonds (e.g., Carbon
Bond 2005 chemical mechanism (CB05; Yarwood et al., 2005) to reproduce the $O_3$-NOx chemistry with acceptable accuracy
and computational cost. This limitation not only restricts the number of species that can be individually evaluated against
observations but also impacts the model's accuracy in representing them. Second, emission inventories report total VOCs,
which then requires the need to use speciation profiles which are limited and commonly outdated (Oliveira et al., 2023). Third,
the availability and quality of observational data for VOCs are often limited in scope (von Schneidemesser et al., 2023; She
et al., 2023). Typically, continuous measurements of VOCs prioritise aromatics due to the recognised health risks associated
with their exposure. Despite the EU's AQ directive (AQD) recommending the measurement of 31 VOC species, only benzene
is currently regulated (EC, 2008). As for other VOCs, the measurements usually come from small campaigns which are limited
in time and location (von Schneidemesser et al., 2023).

This study aims to quantify limitations and uncertainties in Spanish anthropogenic BTX emissions and to improve its rep-
resentation in air quality modelling systems. To do so, we used the High-Elective Resolution Modelling Emission System
(HERMESv3; Guevara et al., 2019b, 2020) model to produce gridded bottom-up emissions, which were used as input in the
Multiscale Online Nonhydrostatic AtmospheRe CHemistry model (MONARCH; Badia et al., 2017; Klose et al., 2021a) chem-
75 ical transport model to simulate surface concentrations of benzene, toluene, and xylene (i.e. o-m-p xylene) across Spain. The
modelling results were then evaluated against official ground-based observation data for the year 2019. By conducting this
evaluation, we are able to identify the spatio-temporal characteristics of the observed concentrations, quantify the differences
with modelled results and link these divergences to uncertainties in the emissions used as input through a set of sensitivity
analyses.

The paper is structured into four sections. Section 2 describes the models used, i.e. HERMESv3 and MONARCH, along
with an overview of the air quality network and observational data used. Section 3 is dedicated to the discussion of results,
focusing on emissions and air quality concentrations of benzene, toluene, and xylene. The model's performance is evaluated
by comparing its outputs with observed data. This section also incorporates sensitivity analyses for the industrial sector and
mopeds and motorcycles. Finally, Section 4, summarises the main conclusions, limitations and uncertainties encountered in
this type of evaluation.

## 2 Data and Methods

### 2.1 Model Description

#### 2.1.1 HERMESv3

The HERMESv3 (Guevara et al., 2019b, 2020) is an open source, parallel and stand-alone multi-scale atmospheric emission
modelling framework that computes anthropogenic emissions for atmospheric chemistry models. The model consists of two



modules, a global-regional module (HERMESv3_GR) and a bottom-up module (HERMESv3_BU), that can work combined or separated depending on the working domain and scope of the applications. For this work, the HERMESv3_GR module is only used to process emissions reported by the CAMS regional gridded inventory (CAMS-REGv4.2; Kuenen et al., 2022) outside of Spain as well as for the shipping sector, while the HERMESv3_BU is used to produce bottom-up anthropogenic emissions in Spain (Guevara et al., 2020).

HERMESv3_BU module estimates anthropogenic emissions at high spatial- (e.g. road link, industrial facility level) and temporal- (hourly) resolution using state-of-the-art calculation methods, based on (but not limited to) the calculation methodologies reported by the European EMEP/EEA air pollutant emission inventory guidebook, that combine local activity and emission factors. The model computes bottom–up emissions from energy and manufacturing industrial facilities, road transport, residential and commercial combustion activities, other mobile sources (landing and take-off cycles in airports, agricultural machinery, recreational boats, shipping emissions in ports), fugitive emissions from fossil fuels (storage and transportation), domestic and industrial use of solvents, and agricultural activities (livestock and use of fertilisers).

The VOC speciation mapping disaggregates total VOCs to the species needed by the atmospheric chemistry model of interest and its corresponding gas-phase and aerosol chemical mechanism. Each individual sector is assigned with a set of profiles with numerical factors (mol of chemical species per gram of source pollutant) to convert the total emissions into the output model species. The number of speciation profiles considered varies according to the pollutant sector of interest. For instance, in the case of residential combustion, the number of speciation profiles proposed is equal to the number of fuel types considered (e.g. natural gas, biomass). In contrast, in the case of road transport, specific profiles are assigned to each vehicle category (Guevara et al., 2020). The speciation profiles used in HERMESv3 are based on the work done by Oliveira et al. (2023), which performed a collection, review and comparison of profiles available from state-of-the-art works.

### 2.1.2 MONARCH

The MONARCH model (Badia et al., 2017; Klose et al., 2021a) is an online multipurpose atmospheric-chemistry integrated system. MONARCH includes a gas-phase module combined with a hybrid sectional-bulk multi-component mass-based aerosol module to simulate the chemistry of the troposphere. MONARCH is coupled online with the Nonhydrostatic Multiscale Model on the B-grid (NMMB; Janjic and Gall, 2012) meteorological core.

The gas-phase chemistry in MONARCH solves the Carbon Bond 2005 chemical mechanism (CB05; Yarwood et al. (2005)) extended with chlorine chemistry (Sarwar et al., 2012) and additional oxidation pathways to form SOA from benzene, toluene, xylene, isoprene and terpenes. Benzene is implemented to form SOA, while xylene and toluene are also involved in $O_3$-NOx chemistry. The reactions solved by the model involving BTX are shown in Appendix A. The core CB05 mechanism considers 51 chemical species and solves 156 reactions. VOCs are lumped into several groups, such as: propionaldehyde and higher aldehydes (ALDX), internal olefin (IOLE), terminal olefin carbon bond (OLE), paraffin carbon bond (PAR), terpene (TERP). Notably, the mechanism also accounts for explicit species, namely acetaldehyde (ALD2), benzene (BENZENE), ethene (ETH), ethanol (ETOH), ethane (ETHA), formaldehyde (FORM), isoprene (ISOP), methanol (MEOH), toluene (TOL), and xylene (XYL).



The CB05 is well formulated for urban to remote tropospheric conditions, and it uses photolysis rates computed with the Fast-J scheme (Wild et al., 2000) considering the physics of each model layer (i.e., clouds, absorbers such as ozone). The dry deposition of gases uses a resistance scheme based on Wesely (1989) for the canopy or surface resistance while scavenging and wet deposition for precipitating and non-precipitating clouds follows the scheme of Byun and Ching (1999); Foley et al. (2010). Further details (i.e. aerosol processes) are available elsewhere (Spada, 2015; Klose et al., 2021a; Navarro-Barboza

et al., 2023).

For this work, MONARCH was set up with a rotated lat-lon projection, with a spatial resolution of 0.1° by 0.1° for Spain. The domain over Spain is presented in Fig. 1. The setup accounts for 24 vertical layers (top 50 hPa). The boundary conditions for the meteorological module come from the European Centre for Medium-Range Weather Forecasts (ECMWF) Reanalysis v5 (ERA5) for the year of 2019, and for the chemistry is derived from the Copernicus Atmospheric Monitoring Service

(CAMS) global atmospheric composition reanalyses, which is build on ECMWF's Integrated Forecasting System (IFS; Flemming et al., 2015). As previously mentioned, the anthropogenic emissions are feed by HERMESv3. The biogenic emissions are estimated with MEGANv2.04 (Guenther et al., 2006, 2012), which is fully implemented inside MONARCH. The speciation employed emits xylene (lumping p-cymene and o-cymene MEGAN species) and toluene (lumping toluene, benzaldehyde, methyl benzoate, phenylacetaldehyde, methyl salicylate, and indole MEGAN species) among other biogenic species.

## 2.2   Air Quality Network and Observational Data

In order to evaluate the performance of the MONARCH model, we used the data obtained for the year 2019 from the European Environment Agency (EEA) AQ e-reporting (EEA, 2023) and complemented with data provided by the Spanish ministry. To do so, we used the harmonised data set from Globally Harmonised Observations in Space and Time (GHOST). The GHOST data set is one of the most extensive collections of harmonised measurements of atmospheric composition at the surface level.

More detailed information can be found in Bowdalo et al. (2023). From this data set, our selection process only considered observations that meet the default quality assurance (QA) criteria specified by GHOST (the details of the QA applied can be found in Table C1 of the Appendix C) and have temporal coverage greater than 75 % in 2019. When applying these filters, 62 stations measuring benzene were dropped, along with 10 stations measuring toluene and 6 stations measuring xylene. The removal of these stations caused a reduction of spatial coverage, where the majority of the stations were located in south

and north-western part of Spain. The temporal coverage criteria was the main driver to remove these stations, although the QA criteria number 83 (see Appendix C for the full description) also had a substantial impact. This shows, mainly for stations measuring benzene, the low quality of the measurements available, and while this reduction may be perceived as a limitation for model evaluation, it is a critical measure to ensure the reliability of measurements. It is also crucial to recognise the potential measurement uncertainties, as exemplified by Gallego-Díez et al. (2016), which can significantly vary based on the chosen

methodology and instrumentation setup. The measurement methods for each station, when available, are detailed in Appendix F.





**Table 1.** Available number of stations by area classification and station type measuring benzene, toluene, and xylene in 2019. The number of stations that only measure with daily resolution is shown in parentheses.

| Station classification | Benzene | Toluene | Xylene |
|---|---|---|---|
| Urban Traffic | 22 (4) | 14 (2) | 13 (7) |
| Suburban Traffic | 1 | 1 | 1 (1) |
| Rural Traffic | 0 | 0 | 0 |
| Urban Industrial | 1 | 1 | 1 (1) |
| Suburban Industrial | 8 (2) | 7 (4) | 7 (6) |
| Rural Industrial | 1 | 0 | 0 |
| Urban Background | 8 (1) | 8 (1) | 6 (3) |
| Suburban Background | 5 | 2 | 1 |
| Rural Background | 1 | 1 | 1 (1) |
| TOTAL | 47 | 34 | 30 |

Applying the temporal coverage of 75 % and the QA criteria resulted in a final dataset of 47 stations (40 stations measuring hourly and 7 only measuring daily) measuring benzene, 34 stations (27 stations measuring hourly and 7 only measuring daily) measuring toluene and 30 stations (11 stations measuring hourly and 19 only measuring daily) measuring xylene. Table 1

shows the available number of stations per station classification measuring benzene, toluene and xylene in 2019. The number of stations measuring with daily resolution is shown in parentheses. The locations and spatial distribution of the stations, along with their respective classifications, are shown in Fig. 1. Examining the spatial distribution of the final dataset reveals a heterogeneous coverage. Notably, there is a lack of monitoring stations in the southern and north-western regions of Spain, as well as in Barcelona, specifically for stations measuring toluene and xylene. A complete description of the location, area and

station classification, and the measuring time resolution of each station can be found in Appendix F.





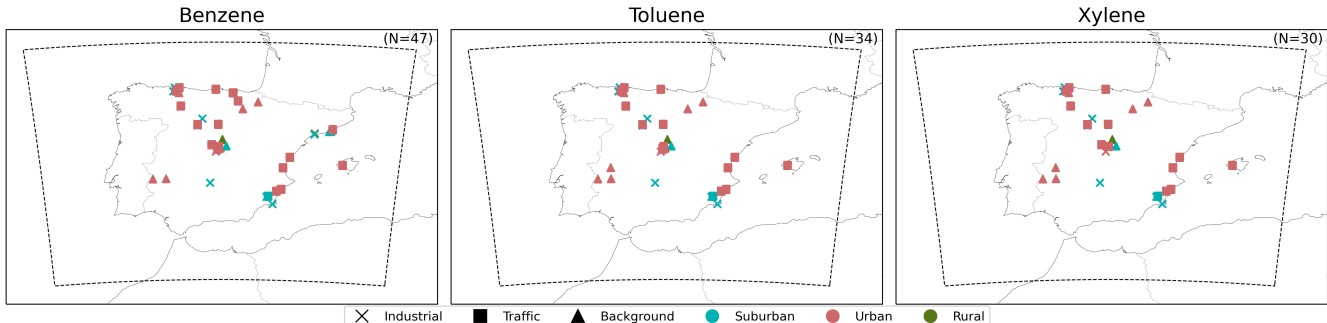

**Figure 1.** Location of the air quality stations measuring benzene, toluene and xylene (with an temporal coverage over 75 %) by type and area classification in Spain for 2019. The station classification is represented by distinct markers, while the area classification is indicated by varying colours. The characteristics of each station are listed in Appendix F.

## 3 Results

### 3.1 Emissions

Figure 2 shows the total anthropogenic emissions of benzene, toluene and xylene in Spain per grid cell (4 km by 4 km) for the year 2019. The analysis of the emission results excludes the estimated MEGAN biogenic emissions, as it does not report any emissions for benzene, and their contribution to total toluene and xylene was found to be minimal (Henrot et al., 2017). To support the discussion, a map identifying the name and location of the different Spanish NUTS2 (Nomenclature of Territorial Units for Statistics) administrative regions and the main cities mentioned in this work is provided in Fig. B1 in Appendix B.

The total estimated emissions in Spain are 11 kilotons (kt) for benzene, 36 kt for toluene and 25 kt for xylene. The main Spanish NUTS2 administrative regions emitting benzene are Andalusia (1.7 kt, 17% of the total emitted benzene), Aragon (1.6 kt, 15%), and Catalonia (1.4 kt, 14%). Notably, each AC exhibits distinct emission characteristics. In Andalusia, the residential sector accounts for the majority of emissions, contributing 74 % (1.3 kt), primarily from wood combustion. In contrast, for Aragon and Catalonia, the industrial sector is the main source of emissions, comprising 82 % (1.3 kt) and 53 % (0.8 kt), respectively. Aragon's substantial industrial contribution is primarily due to the presence of a large paper and pulp manufacturing industry, which is responsible for 40 % of industrial benzene emissions in this region.

For toluene emissions, the top-emitting NUTS2 regions are Catalonia (8 kt, 23 % of the total emitted toluene), the Valencian Community (4 kt, 11 %), and Andalusia (4 kt, 11 %). Similarly, in the case of xylene, the top emitters are Catalonia (5 kt, 19 % of the total emitted xylene), Andalusia (3 kt, 12 %), and the Valencian Community (3 kt, 11 %). For both pollutants, the solvent sector predominates, accounting for 72 % and 89 % of total emissions, respectively. In Andalusia, the solvent sector contributes less to xylene emissions (50 %) due to the presence of several petrochemical complexes, which account for approximately 25 % of these emissions. It is worth noting that Madrid has significant emissions, with 3.5 kt of toluene and 2.6 kt of xylene. However, due to its smaller area compared to other ACs, it does not rank in the top 3. Nevertheless, when





looking at the emission intensity, Madrid ranks first for toluene (emitting 0.437 t.km$^{-2}$) and xylene (emitting 0.328 t.km$^{-2}$) and second for benzene (emitting 0.057 t.km$^{-2}$).

Overall, regions with higher emission values in Spain tend to coincide with more populated areas or significant industrial activity. For instance, regions with lowest population density (about 26 hab.km$^{-2}$), namely Castilla y León, Castilla–La Mancha and Extremadura (INE, 2021), despite collectively covering 44 % of peninsular Spain's total area, contribute only 15 %, 14 %, and 13 % of total benzene, toluene, and xylene emissions, respectively. Regarding spatial distribution, benzene hotspots are located in specific industrial areas, such as Toledo in the south of Madrid and Tarragona in the south of Catalonia. In contrast, hotspots for toluene and xylene are primarily located in urban areas.

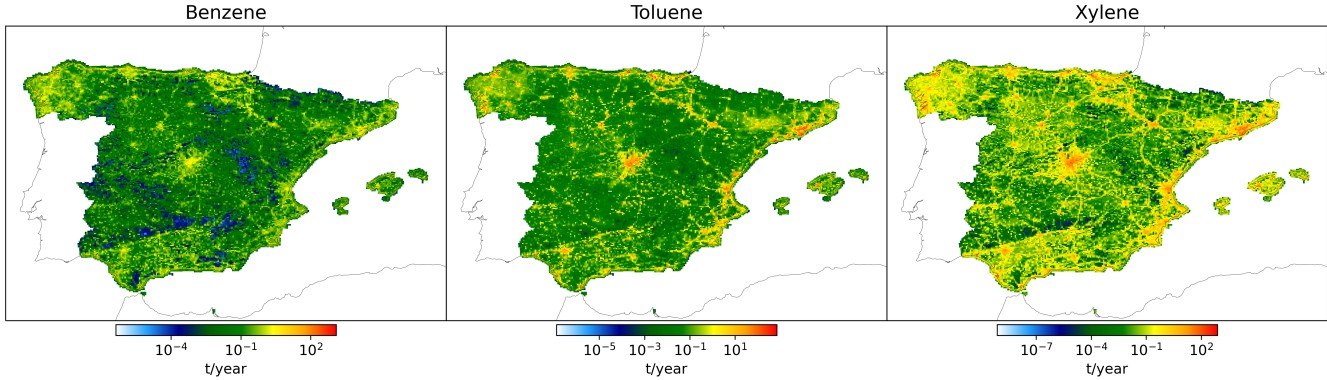

**Figure 2.** Bottom-up estimated emissions (t.year$^{-1}$) of benzene, toluene and xylene in Spain at 4 km by 4 km.

To quantify the contribution of the different anthropogenic activities to total BTX emissions, we conducted individual runs of HERMESv3_BU, grouping its emission sectors as follows: road transport; the solvent sector was divided depending on its application resulting in domestic use and industrial, which accounts for the solvent production and use in the industrial context; residential, which corresponds to the residential and commercial combustion sector; industry, which accounts for energy and manufacturing industrial emissions except for VOC process emissions linked to the use of solvents (e.g., manufacturing of 200 chemical products or paint application); and 'other', which accounts for the remaining emission sectors, including, aviation (landing and take-off cycles in airports), shipping in port areas, recreational boats, livestock, agricultural machinery, and the fugitive fossil fuel sector.

Figure 3 shows the contribution for each grouped sector per day over the year of 2019 and VOC species for urban areas in Spain. The information regarding urban settlements was sourced from Schiavina et al. (2023). Benzene emissions show 205 notable variations throughout the year, with industries being the primary contributor, accounting for 35 % to 44 % on average. Another relevant source is the road transport, which contributes on average 29 %. However, due to the weekend effect, the contribution from this sector lowers during weekends. It is worth mentioning that the emissions from the residential sector show a big seasonality, as they are primarily driven by the burning of wood for residential heating. While the contribution is higher in more rural areas, it reduces from 18 % in winter to 6 % in summer.



The contribution of the different sources to toluene emissions remains quite constant along the year, with the industrial solvent sector being the primary contributor, averaging around 85 % annually. Within the industrial solvent sector, the primary activities emitting toluene are the fabrication and treatment of chemical products, such as paint manufacturing (50 %), industrial paint application (14 %), and non-industrial paint application (10 %). The domestic use of solvent, despite having an important contribution to total VOCs (around 11 % in 2019), has a very limited contribution to BTX emissions compared to industrial

use of solvent because the speciation profile used assumes less than 1 % for each individual specie.

     Similar to toluene, xylene is also primarily dominated by the solvent sector, with a contribution averaging around 77 % annually. Overall, the shares show a relatively constant pattern, where transport contributes from 10 to 16 %, reaching higher values in winter. For xylene, the main activity from the industrial solvent sector is the paint application, such as industrial paint application (41 %), wood paint application (16 %) and paint application in the manufacturing of automobiles (11 %).

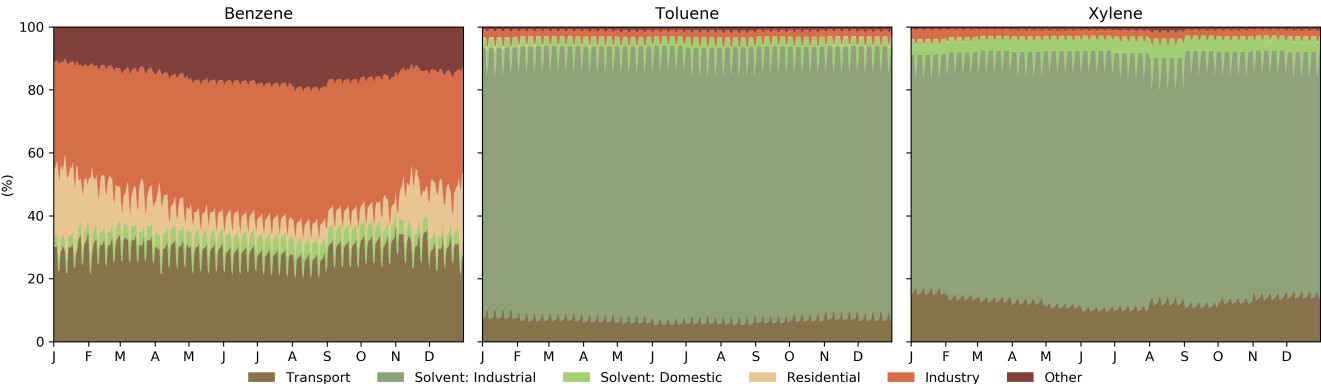

**Figure 3.** Temporal source contribution (%) to total emissions of benzene, toluene and xylene in Spanish urban areas in 2019.

Figure 4 shows the contribution (%) of each sector to total benzene, toluene and xylene emissions per grid cell at a resolution of 4 by 4 km. A similar figure is presented in Appendix D showing the emissions (t) of benzene, toluene and xylene by emission sector. For benzene, the residential sector is the dominant sector in Spain, with an overall contribution of over 86 %, except in urban areas, where the contribution is generally under 20 %. This is more evident in Madrid and Barcelona, although it can be seen in all major cities. This spatial pattern can be attributed to the predominant emission of benzene from residential

biomass combustion, which in HERMESv3 is allocated based on rural population density due to the minimal usage of this fuel in urban areas, where natural gas dominates the energy mix. In urban areas, the main contribution comes from the transport sector (overall, over 60 %) and to a smaller extent from the domestic solvent use (up to 24 %). As expected, over major interurban roads, the transport sector is usually the main contributor. In the more rural areas, agriculture activity dominates benzene emissions, contributing around 94 %. Furthermore, the industrial sector makes a substantial contribution (>90 %) to

specific grid cells, particularly in areas where large facilities and industrial hotspots are situated. This is mostly evident over the Catalonia region but also over main industrial areas such as Valencia and the central region of Andalusia.





For toluene and xylene, the contribution by sectors shows a similar spatial pattern. The industrial solvent sector dominates the urban areas with overall values ranging between 63 - 97 %. However, the cells intersecting main interurban roads are dominated by the traffic sector, contributing around 44 - 80 %. The residential emissions have an essential contribution in more rural areas, mainly due to biomass burning, where they are more relevant for xylene (up to 76 %) than for toluene (up to 70 %). The main differences between toluene and xylene spatial patterns are seen for 'other'. Despite having lower values (see Fig. 2), rural areas are predominantly characterised by livestock emissions, primarily from cattle. In several locations, grid cells are entirely dominated by the livestock sector. The variations in cattle emissions between toluene and xylene can be attributed to the differences in their respective speciation, with toluene accounting for a greater percentage (1 %) than xylene (<0.1 %) (Oliveira et al., 2023).

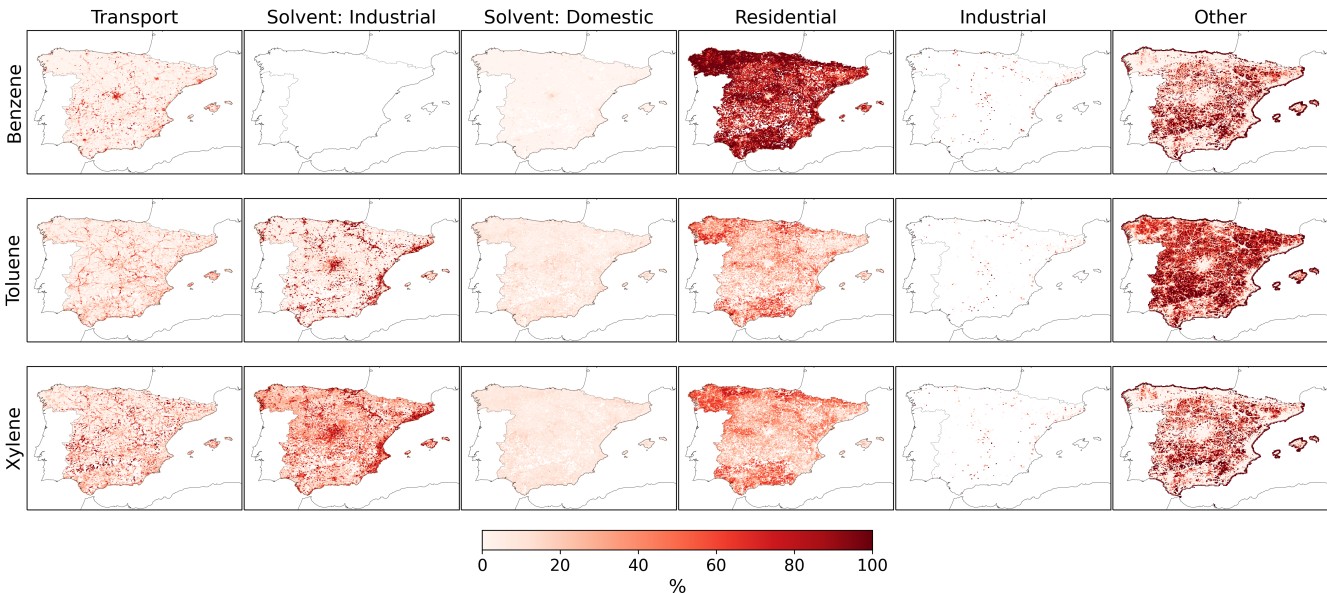

**Figure 4.** Relative contribution (%) of individual anthropogenic emission sources to total benzene, toluene and xylene annual emissions in Spain in 2019 per grid cell (4 km by 4 km).

## 3.2 Surface BTX Concentrations

The following subsection presents the comparison between modelled BTX concentrations estimated with MONARCH and observations reported by air quality stations. The location and characteristics of each specific station is presented in Appendix F. To analyse the impact of the different emission sectors and reduce the number of variables, we evaluated the model by aggregating the measurement stations by station type and area classification as follows:

– Urban and suburban traffic stations, as there is only one suburban traffic station and due to the similarity in trends observed between the two. Although the model has limitations in replicating traffic sites accurately, analysing these sites can still yield valuable information.





- Urban and suburban industrial stations were also aggregated but only for the daily resolution, as the urban industrial
station and the rural industrial only present daily values.

- Urban and suburban background stations were aggregated due to their similar observed trends, and this consolidation
     was necessary given the limited availability of suburban background stations measuring toluene (N=2) and xylene (N=1).
     Rural background stations were excluded from the analysis as only one site with these characteristics was available.

Annual and seasonal mean statistics are computed for each group of stations, with seasons corresponding to winter (January,
February and December), spring (March, April and May), summer (June, July and August) and autumn (September, October
and November). The statistics used in this work are commonly used to evaluate model performances against observations. For
this, we selected N (sample size), observed mean, modelled mean, Mean Bias (MB), Normalised Mean Bias (NMB), Root
Mean Squared Error (RMSE) and the pearson correlation coefficient (r).

### 3.2.1 Benzene

As previously mentioned, benzene is the only compound currently regulated in the EU by the AQD, with a limit value of 5
$\mu$g.m$^{-3}$ (annual mean). Air quality monitoring stations in Spain consistently report values below this regulatory limit, and
none record levels near the limit. Notably, the highest recorded benzene concentration is observed in Barcelona at a traffic
station (ES1438A, 3.00 $\mu$g.m$^{-3}$), followed by an industrial station in Asturias near facilities with coke ovens (ES2075A,
2.89 $\mu$g.m$^{-3}$), and another Barcelona urban traffic station (ES1480A, 2.59 $\mu$g.m$^{-3}$). The World Health Organization (WHO)
also establishes a reference level (RL) of 1.7 $\mu$g.m$^{-3}$ for the annual mean concentration of benzene (WHO, 2019), which is
surpassed only by the three monitoring stations mentioned above, underscoring the overall compliance with the air quality
standards but not completely for the health guidelines.

The spatial distribution of annual mean benzene concentrations in both observed and modelled data shows a fairly consistent
alignment (see Fig. 5). However, the model struggles to replicate isolated hotspots, primarily attributable to its resolution and
270 the sporadic occurrence of high peaks (mainly in industrial areas), which, using classical emission inventory approaches, is
challenging to reproduce. The maximum annual mean value modelled by MONARCH is 2.29 $\mu$g.m$^{-3}$, also below the AQD
limit value, and located south of Madrid, very close to two chemical facilities, namely one that produces chemical organic and
inorganic products and another that produces pharmaceutical chemical products. Unfortunately, no stations are located in this
area to validate the modelled results.



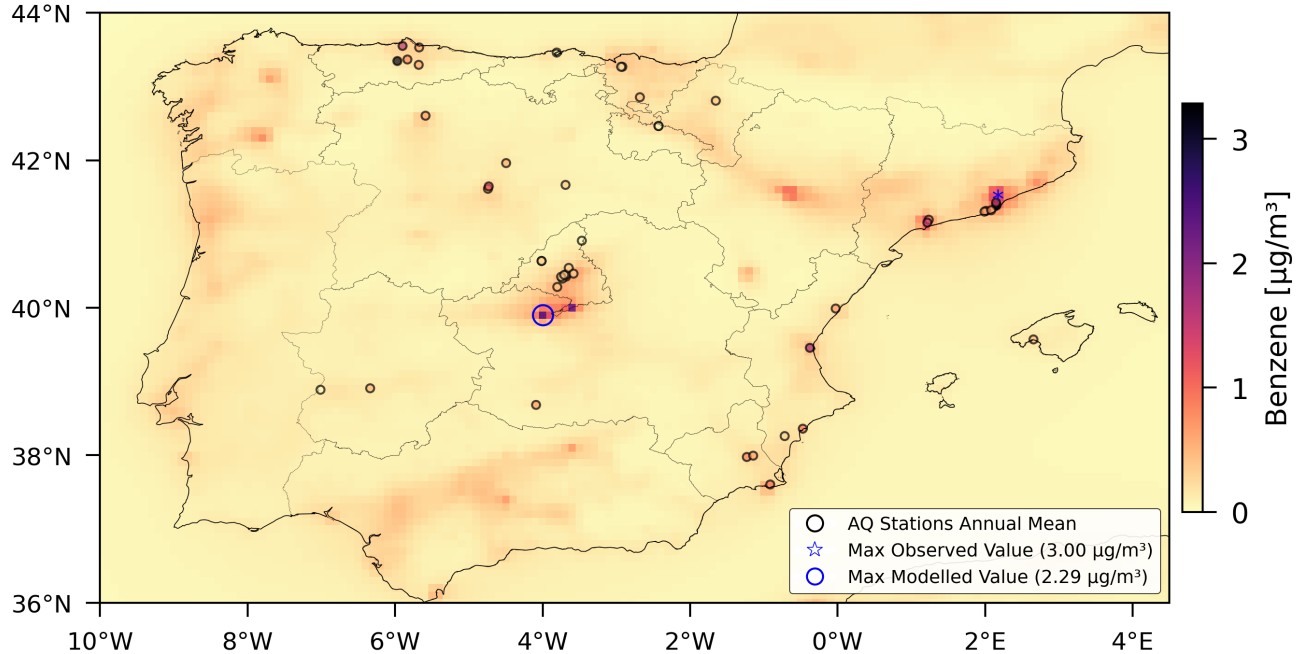

**Figure 5.** Annual mean of modelled (0.1° by 0.1°) and observed benzene concentrations ($\mu$g.m$^{-3}$) for 2019. Black circles represent air quality station locations, while colours the annual mean observed values. The star indicates the station with the highest observed value, and the blue circle marks the maximum modelled value.

Figure 6 shows the annual average MB ($\mu$g.m$^{-3}$) at urban (on the left) and suburban stations (on the right). In general, as depicted in the figure, the model tends to underestimate benzene concentrations. For the urban stations, the MB is generally low (from -0.21 to 0.92 $\mu$g.m$^{-3}$), showing an adequate model performance in these areas. However, the model tends to be highly underestimated for some specific stations where the largest concentrations are observed. This is mainly attributed to underestimations of VOCs, although to a lesser extent the model's coarse resolution could also limit its capacity to capture intense urban and industrial hotspots accurately. It is worth nothing that the resolution plays a smaller role since the models performs well in replicating NO2 concentrations in industrial areas (see Appendix E). The largest underestimations are mainly observed in the East coast, particularly in Barcelona, at two traffic stations, ES1438A (MB = -2.17 $\mu$g.m$^{-3}$ and NMB= -72.3 %) and ES1480A (MB = -1.75 $\mu$g.m$^{-3}$ and NMB= -67.4 %), as well as one traffic station in the Valencian Community (ES1239A, MB = -1.26 $\mu$g.m$^{-3}$ and NMB = -76.2 %). The same happens for suburban stations measuring high concentration levels, which occur in the northern part of the country in Asturias. For instance, industrial station ES2075A (MB = 1.21 $\mu$g.m$^{-3}$ and NMB = -93.0 %) is located near two coke oven facilities, and industrial station ES0879A (MB = -2.69 $\mu$g.m$^{-3}$ and NMB = -80.5 %) is located near one coke oven facility. Of the 13 suburban stations, 8 are near major industrial facilities like refineries and coke ovens, showing larger underestimations. The suburban stations located in Madrid and Barcelona in background environments show low MB values (-0.24 to 0.54 $\mu$g.m$^{-3}$).





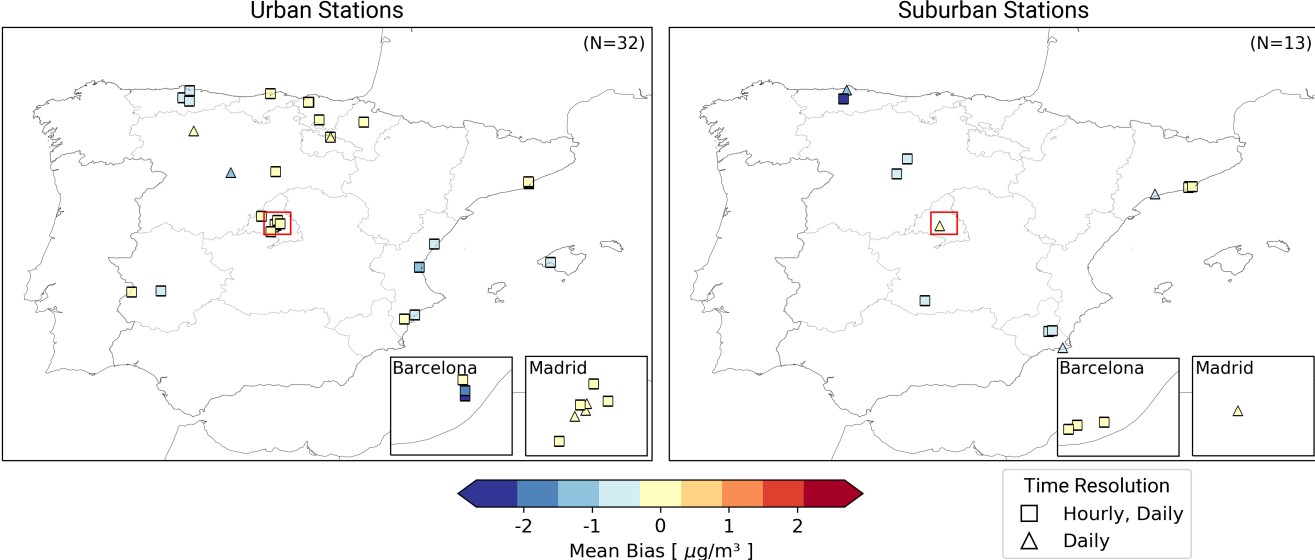

**Figure 6.** Benzene annual Mean Bias (MB) ($\mu$g.m$^{-3}$) between observed and modelled concentrations at urban (left) and suburban stations (right).

Figure 7 shows the comparison between observed and modelled hourly, daily, and monthly benzene cycles per station classification, including the variability of the standard deviation as the shaded region. Table 2 provides statistical metrics for all stations and by station classification, both for the entire year and for each season, including N, observed mean, modelled mean, MB, NMB, RMSE, and r.

Observed benzene levels show marked seasonal variations, with the highest levels occurring during winter across all stations, a characteristic that is also captured by MONARCH. Correlation coefficients (r) vary by season, with the highest value observed in summer (0.45) and the lowest in winter (0.28). The annual NMB indicates that, on average, the model underestimates benzene levels by approximately -51.5 %. This underestimation is more pronounced in industrial stations (-61.9 %) and much less in background stations (-20.7 %). The NMB is also heterogeneous across seasons, winter being the period in which the underestimation is at the lowest level (-38.7 % for all stations) when compared to the other seasons (e.g., -59.2 % in autumn). The model slightly overestimates levels during winter at background stations, where the NMB is 2.1 %. Background stations generally exhibit lower underestimations and higher correlation coefficients, suggesting a better model performance in this type of environment. This is also observed in the comparison between modelled and observed background hourly, monthly, and weekly cycles, with results indicating generally good performance of MONARCH, except for the small overestimation of the morning peak.

The model consistently underestimates benzene levels in traffic stations (MB = -0.4 $\mu$g.m$^{-3}$, and NMB = -58%) while effectively replicating the observed hourly, monthly, and weekly cycles. These underestimations can be partially attributed to the model's resolution (i.e., approximately 10x10 km$^2$), as this can lead to a 'smoothing' of emissions and subsequently result in



underestimations of concentrations in these stations. Nevertheless, the performance of the model is not consistent across urban traffic sites. While in Madrid, the model exhibits a good performance and low MB values (between -0.20 and 0.14 $\mu$g.m$^{-3}$), in other large cities like Barcelona and Valencia, the model's underestimations are more significant (see Fig. 6), indicating a potential misrepresentation of local sources that are characteristic of these cities. This aspect is further analysed and discussed in Section 3.3.2. Notably, underestimations are more pronounced during winter, suggesting a potential underestimation of road traffic cold start emissions.

Among the different station classifications, industrial stations display the highest underestimations (NMB = -61.9 %) and the lowest correlation values (0.13 annual), indicating poor model performance in these areas. This is also evident when looking at the standard deviation variability of the industrial observations (see Fig. 6), which shows the largest variability compared to other areas. The low correlation and substantial RMSE (1.4 $\mu$g.m$^{-3}$) between observed and modelled values is partially explained by the occurrence of sporadic high peaks reported by observations, which are potentially linked to situations when industries are not under normal operating conditions (e.g., maintenance activities, leaks) and that are not captured adequately by the emission inventory. After applying a filter to remove possible outliers in the observational data, the model's performance slightly improves, with the annual NMB increasing to -56.3 % and the correlation to 0.22. However, even removing these points, the model still exhibits substantial underestimation. To better understand the limitations of MONARCH in capturing observed industrial benzene levels, we identified and located these stations and the surrounding industrial facilities. Stations near large refineries, coke ovens, and car manufacturing facilities consistently exhibit flat-modelled profiles and systematic underestimations, which could indicate underestimations in emissions from these activities. This aspect is further analysed in Section 3.3.1.





**Figure 7.** Observed (black line) and modelled (blue line) benzene hourly, weekly, and monthly cycles ($\mu$g.m$^{-3}$) per station classification. The shaded region corresponds to the standard deviation variability.





**Table 2.** Seasonal and annual statistics for N, observed mean, modelled mean, MB, NMB, RMSE and r, per station classification for benzene for Spain in 2019 obtained with MONARCH.

| Station classification | Period | Statistical metrics | | | | | | |
|---|---|---|---|---|---|---|---|---|
| | | N | Obs. mean | Mod. mean | MB | NMB | RMSE | r |
| | | (-) | ($\mu g/m^3$) | ($\mu g/m^3$) | ($\mu g/m^3$) | (%) | ($\mu g/m^3$) | (-) |
| **All Stations** | Annual | 16461 | 0.7 | 0.3 | -0.4 | -51.5 | 0.9 | 0.38 |
| | Winter (DJF) | 4056 | 1.1 | 0.6 | -0.4 | -38.7 | 1.2 | 0.28 |
| | Spring (MAM) | 4144 | 0.6 | 0.3 | -0.4 | -57.9 | 0.8 | 0.34 |
| | Summer (JJA) | 4193 | 0.5 | 0.2 | -0.3 | -58.8 | 0.6 | 0.45 |
| | Autumn (SON) | 4068 | 0.7 | 0.3 | -0.4 | -59.2 | 0.9 | 0.37 |
| **Traffic** | Annual | 7915 | 0.8 | 0.3 | -0.4 | -58.0 | 0.9 | 0.56 |
| | Winter (DJF) | 1912 | 1.1 | 0.6 | -0.5 | -47.0 | 1.2 | 0.45 |
| | Spring (MAM) | 1946 | 0.7 | 0.2 | -0.4 | -63.6 | 0.8 | 0.57 |
| | Summer (JJA) | 2026 | 0.5 | 0.2 | -0.3 | -64.0 | 0.6 | 0.70 |
| | Autumn (SON) | 2031 | 0.7 | 0.2 | -0.5 | -65.2 | 0.9 | 0.63 |
| **Industrial** | Annual | 3421 | 1.1 | 0.4 | -0.7 | -61.9 | 1.4 | 0.13 |
| | Winter (DJF) | 864 | 1.5 | 0.7 | -0.8 | -54.2 | 1.8 | 0.07 |
| | Spring (MAM) | 836 | 0.9 | 0.3 | -0.6 | -66.5 | 1.3 | 0.00 |
| | Summer (JJA) | 855 | 0.8 | 0.3 | -0.6 | -65.9 | 1.0 | 0.16 |
| | Autumn (SON) | 866 | 1.0 | 0.3 | -0.7 | -65.7 | 1.4 | 0.05 |
| **Background** | Annual | 4538 | 0.5 | 0.4 | -0.1 | -20.7 | 0.5 | 0.54 |
| | Winter (DJF) | 1146 | 0.7 | 0.7 | 0.0 | 2.1 | 0.7 | 0.38 |
| | Spring (MAM) | 1181 | 0.4 | 0.3 | -0.1 | -32.6 | 0.3 | 0.66 |
| | Summer (JJA) | 1142 | 0.4 | 0.2 | -0.1 | -35.3 | 0.2 | 0.77 |
| | Autumn (SON) | 1069 | 0.5 | 0.3 | -0.2 | -33.9 | 0.4 | 0.52 |

### 3.2.2 Toluene

The highest toluene concentrations in 2019 were measured at two urban traffic stations located in Valencia (ES1239A, 5.95 $\mu$g.m$^{-3}$) and Valladolid (ES1631A, 3.77 $\mu$g.m$^{-3}$), respectively, followed by a suburban industrial station in Murcia (ES1627A, 3.63 $\mu$g.m$^{-3}$). It is worth mentioning that although toluene is one of the recommended VOCs for measurement in the AQD, no legal or recommended limits are defined by the EC or the WHO.





The modelled results (see Fig. 8) show that the maximum annual mean value is 9.65 $\mu$g.m$^{-3}$ and is located in Barcelona, where no measurements are available. Overall, the spatial pattern between observed and modelled annual mean values shows a good agreement, with urban areas being depicted as the main hotspots according to both sources of information.

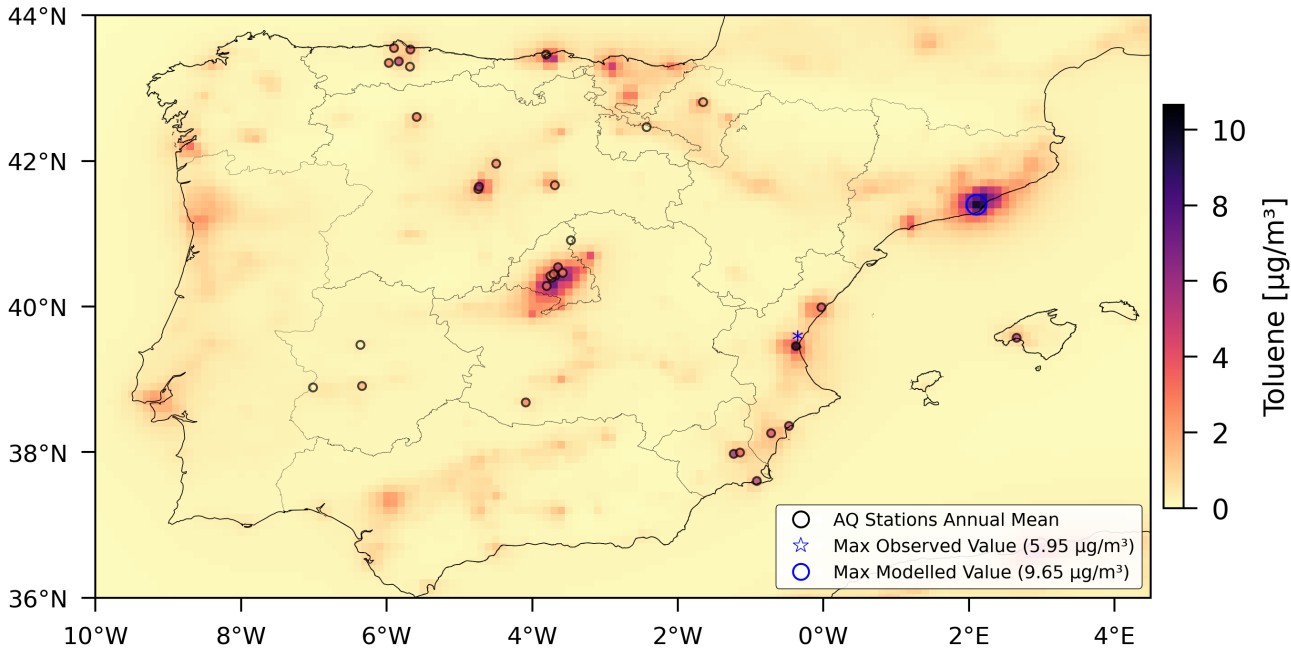

**Figure 8.** Annual mean of modelled (0.1° by 0.1°) and observed toluene concentrations ($\mu$g.m$^{-3}$) for 2019. Black circles represent air quality station locations, while colours the annual mean observed values. The star indicates the station with the highest observed value, and the blue circle marks the maximum modelled value.

Figure 9 shows the toluene annual MB ($\mu$g.m$^{-3}$) for urban stations (on the left) and suburban stations (on the right). Oppositely to benzene, for which results were fairly consistent across sites (negative or close to 0 MB), heterogeneous results are observed for toluene (a mix of positive and negative MB). A significant overestimation of the modelled concentrations is observed in Madrid's urban and suburban areas (between 0.46 and 4.17 $\mu$g.m$^{-3}$). A similar, albeit to a lesser extent, overestimation is shown in central northern urban regions, specifically Cantabria (ES1580A, MB = 1.80 $\mu$g.m$^{-3}$ and NMB = 150 %)

and Navarre (ES1580A, MB = 1.80 $\mu$g.m$^{-3}$ and NMB = 150 %, and ES1472A, MB = 1.56 $\mu$g.m$^{-3}$ and NMB = 61 %). Most of the underestimations observed in urban stations occur at traffic monitoring stations. The biggest underestimation is found at one of these stations in Valencia (ES1239A, MB= -2.61 $\mu$g.m$^{-3}$). In suburban stations, the model tends to underestimate, with the exception of the station in Madrid (ES1193A, MB= 3.14 $\mu$g.m$^{-3}$) and, to a lesser extent, the station in Valladolid (ES0651A, MB= 2.23 $\mu$g.m$^{-3}$). In other areas, modelled values are more aligned with observations, with MB being much

lower. As previously shown, modelled toluene concentrations in the urban area of Barcelona are even higher than the ones





modelled in Madrid. However, due to the lack of measurements in this region, we cannot conclude if these modelled levels are also overestimated, as in the case of Madrid.

The observed overestimations in urban stations suggest a potential limitation in the spatial disaggregation of specific dominant emission sources in these areas. This includes specific activities within the solvent sector, mainly from wood paint

application and, to a lesser extent, industrial paint application and paint manufacturing. These sectors are of particular interest as they are the main toluene-emitting sources (see Section 3.1). The proxies considered in HERMESv3 for the spatial disaggregation of these three sectors include land use information related to industrial and commercial areas, population density information, and a shapefile containing the geographical location of individual paint manufacturing facilities, respectively. For example, for the wood paint application, we use the population as a proxy and, coupled with a coarse model resolution, causes

the monitoring stations to fall in emission hotspots, leading to overestimations. However, validating or refining this information is challenging due to the scarcity of detailed and available data concerning these specific activities. Additionally, it is worth noting that this overestimation is not observed for benzene concentrations, as the speciation used for these sectors establishes zero benzene emissions.

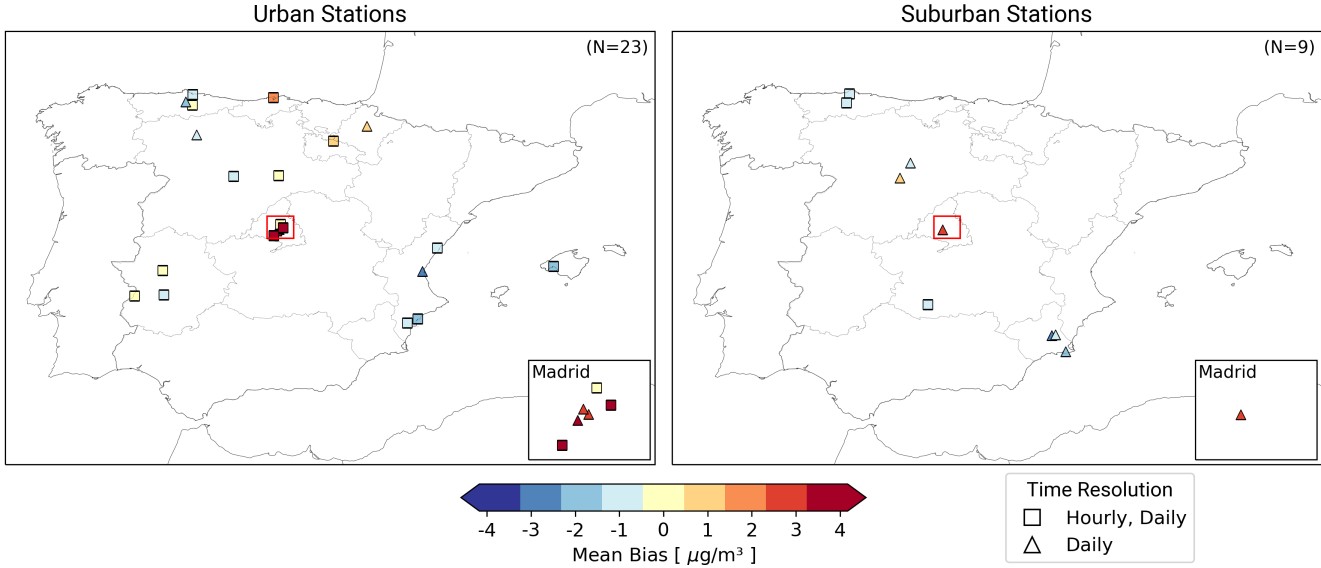

**Figure 9.** Toluene annual Mean Bias (MB) ($\mu$g.m$^{-3}$) between observed and modelled concentrations at urban (left) and suburban stations (right) in 2019.

Figure 10 shows the comparison between modelled and observed toluene hourly, daily, and monthly cycles per station

classification, including the variability of the standard deviation as the shaded region. Modelled cycles are very consistent across stations. At the hourly scale, the model consistently reaches its morning peak between 1-2 hours before the observations. This shift between modelled and observed morning peak is also reported for benzene concentrations (see Fig. 7) and could be due to uncertainties in the chemical processes affecting VOCs and not related as much to the temporal profiles as the



afternoon peak is well replicated. At traffic stations, the model effectively replicates hourly, monthly, and weekly cycles,
although it tends to overestimate concentrations during the nighttime period. It is important to note that the results obtained
at traffic stations compensate for the overestimation observed in the Madrid urban region with underestimations reported in
other urban traffic stations (see Fig. 9). For industrial stations, the model accurately reproduces all observed cycles. However,
it notably underestimates concentrations during weekends, suggesting a larger drop in concentrations during Saturday and
Sunday than the one observed. This behaviour could be linked to an inadequate weekly distribution of emissions in certain
facilities surrounding the industrial stations located in Tarragona, for which we assume that they close or significantly reduce
their activity during weekends, while in reality, they operate continuously. At background stations, a consistent overestimation
of the model is observed, potentially attributed to an inaccurate spatial proxy applied to wood paint application. Currently,
these emissions are disaggregated using population data, although they should be associated with industrial areas.

Table 3 provides statistical metrics for toluene, including N, observed mean, modelled mean, MB, NMB, RMSE, and r, for
all stations and by station classification, both for the entire year and for each season. When considering all stations, the annual
MB (0.1 $\mu$g.m$^{-3}$) and NMB (5.1 %) indicate a slight overestimation. Nevertheless, and as mentioned before, this is a result of
compensation between the slight underestimation observed in traffic (MB = -0.4 $\mu$g.m$^{-3}$) and industrial (MB = -0.5 $\mu$g.m$^{-3}$)
stations and the larger overestimation reported in background sites (MB = 1.2 $\mu$g.m$^{-3}$). During winter, the model exhibits a
larger positive bias (0.4 $\mu$g.m$^{-3}$) and a higher NMB (14.3 %) than the other seasons. For traffic and industrial stations, the
model consistently underestimates in all seasons. However, the underestimations are more pronounced for industrial stations,
but the NMB values remain below -30 %. The industrial stations generally have the highest RMSE values across all time
periods, especially during winter (5.4 $\mu$g.m$^{-3}$). Furthermore, industrial stations consistently demonstrate the lowest r (below
0.24), which is close to zero during spring and summer. Background stations have the highest correlation but show a significant
positive bias in annual and seasonal means and extremely high NMB values (greater than 100 %). As previously mentioned, the
significant overestimation is related to overestimations of the emissions from the solvent sector, mainly from paint application.







**Figure 10.** Observed (black line) and modelled (blue line) toluene hourly, weekly, and monthly cycles ($\mu$g.m$^{-3}$) per station classification. The shaded region corresponds to the standard deviation variability.





**Table 3.** Seasonal and annual statistics for N, observed mean, modelled mean, MB, NMB, RMSE and r, per station classification for toluene in Spain in 2019 obtained with MONARCH.

| Station classification | Period | Statistical metrics | | | | | | |
|---|---|---|---|---|---|---|---|---|
| | | N | Obs. mean | Mod. mean | MB | NMB | RMSE | r |
| | | (-) | ($\mu g/m^3$) | ($\mu g/m^3$) | ($\mu g/m^3$) | (%) | ($\mu g/m^3$) | (-) |
| **All Stations** | Annual | 11719 | 2.0 | 2.1 | 0.1 | 5.1 | 3.2 | 0.30 |
| | Winter (DJF) | 2847 | 2.8 | 3.2 | 0.4 | 14.3 | 4.9 | 0.22 |
| | Spring (MAM) | 2964 | 1.5 | 1.6 | 0.1 | 4.4 | 2.1 | 0.23 |
| | Summer (JJA) | 2969 | 1.4 | 1.5 | 0.1 | 4.7 | 1.8 | 0.29 |
| | Autumn (SON) | 2939 | 2.2 | 2.0 | -0.1 | -5.6 | 3.0 | 0.35 |
| **Traffic** | Annual | 5221 | 2.7 | 2.2 | -0.4 | -16.2 | 3.0 | 0.35 |
| | Winter (DJF) | 1242 | 3.9 | 3.3 | -0.6 | -15.1 | 4.5 | 0.26 |
| | Spring (MAM) | 1307 | 2.0 | 1.8 | -0.2 | -10.8 | 1.9 | 0.27 |
| | Summer (JJA) | 1321 | 1.8 | 1.6 | -0.2 | -10.6 | 1.7 | 0.35 |
| | Autumn (SON) | 1351 | 2.9 | 2.2 | -0.7 | -24.7 | 3.0 | 0.38 |
| **Industrial** | Annual | 2745 | 2.2 | 1.7 | -0.5 | -21.1 | 3.5 | 0.16 |
| | Winter (DJF) | 691 | 3.0 | 2.7 | -0.3 | -10.3 | 5.4 | 0.12 |
| | Spring (MAM) | 665 | 2.0 | 1.4 | -0.6 | -29.4 | 2.8 | 0.07 |
| | Summer (JJA) | 694 | 1.6 | 1.3 | -0.4 | -22.4 | 2.2 | -0.01 |
| | Autumn (SON) | 695 | 2.2 | 1.6 | -0.6 | -27.3 | 3.0 | 0.24 |
| **Background** | Annual | 3459 | 0.8 | 2.1 | 1.2 | 146.3 | 3.2 | 0.58 |
| | Winter (DJF) | 869 | 1.1 | 3.5 | 2.3 | 206.3 | 5.2 | 0.54 |
| | Spring (MAM) | 900 | 0.6 | 1.5 | 0.9 | 139.2 | 1.8 | 0.61 |
| | Summer (JJA) | 878 | 0.7 | 1.4 | 0.7 | 106.0 | 1.5 | 0.66 |
| | Autumn (SON) | 812 | 1.0 | 2.1 | 1.1 | 107.8 | 2.8 | 0.63 |

### 3.2.3 Xylene

In Spain, the highest xylene concentrations in 2019 were measured at two industrial stations, in Palencia (ES1298A, 6.18 $\mu$g.m$^{-3}$) and Valladolid (ES1356A, 5.98 $\mu$g.m$^{-3}$), followed by a traffic station also located in Valladolid (ES1631A, 3.72 $\mu$g.m$^{-3}$). The two industrial air quality stations are located near (< 1 km) two car manufacturing facilities.

Modelled maximum concentrations (5.42 $\mu$g.m$^{-3}$) are reported in Galicia on the northeast coast of Spain (see Fig. 11). The levels are mainly driven by the presence of a car manufacturing facility. However, similar concentrations are also modelled in



the urban areas of Barcelona (5.37 $\mu$g.m$^{-3}$) and Madrid (5.36 $\mu$g.m$^{-3}$). Out of these three regions, measured values are only available in Madrid, showing smaller concentrations (average of 1.16 $\mu$g.m$^{-3}$) compared to MONARCH. As in the case of toluene, and despite being recommended for VOC measurement in the AQD, no legal limits are established for xylene.

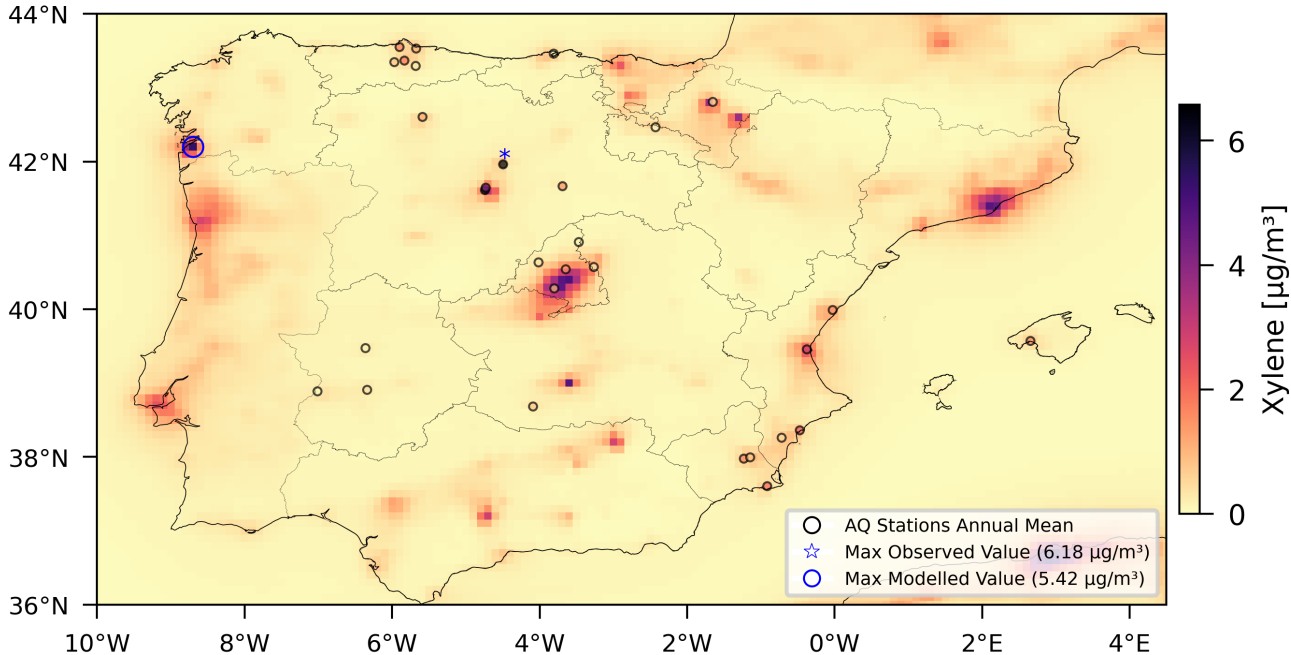

**Figure 11.** Annual mean of modelled (0.1° by 0.1°) and observed xylene concentrations ($\mu$g.m$^{-3}$) for 2019. Black circles represent air quality station locations, while colours the annual mean observed values. The star indicates the station with the highest observed value, and the blue circle marks the maximum modelled value.

Figure 12 shows the xylene annual MB ($\mu$g.m$^{-3}$) for urban (on the left) and suburban stations (on the right). The pattern observed in urban stations is similar to the one reported for toluene, with overestimations occurring in Madrid and Navarre regions and slight underestimations or MB close to 0 reported elsewhere. Looking at the suburban stations, we can see two of them that stand out as highly underestimated (ES1298A, MB = -5.62 $\mu$g.m$^{-3}$; ES1356A, MB = -3.93 $\mu$g.m$^{-3}$). These stations are the ones that report the largest annual mean values and that are located near car manufacturing industries, as described

before. These underestimations were also identified for benzene, although they were more pronounced for xylene. The strong bias suggests that the emissions from this industrial activity are highly underestimated. This aspect is further analysed in Section 3.3.1.



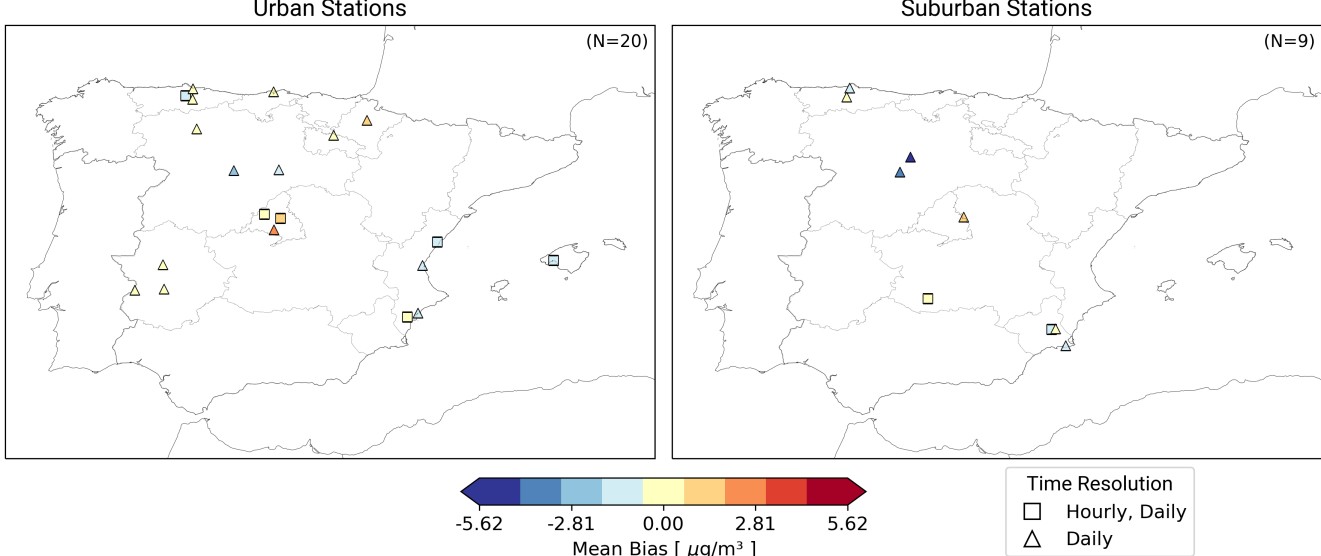

**Figure 12.** Xylene annual Mean Bias (MB) ($\mu$g.m$^{-3}$) between observed and modelled concentrations at urban (left) and suburban stations (right) in 2019.

Figure 13 shows the comparison between modelled and observed xylene hourly, daily, and monthly cycles per station classification, including the variability of the standard deviation as the shaded region.

For traffic stations, we observe that the underestimation of the model is significantly more pronounced during the winter months, indicating an underestimation of the traffic emissions related to cold-start emissions, as the levels observed during spring and summer are being well reproduced. The model effectively reproduces the weekly cycle, accurately capturing the weekend decrease. At the hourly scale, the already discussed shift between observed and modelled peaks for benzene and toluene also occurs for xylene, indicating an issue related to VOC chemistry in MONARCH.

The hourly cycle is not presented in the figure for industrial stations as only one station measuring xylene hourly is available. This limitation makes drawing conclusive insights for this type of station challenging. When examining the monthly cycles, distinctive patterns are observed for measured and modelled values. In the first case, noticeable peaks occur during the summer months, which are not captured by the model. On the other hand, MONARCH successfully replicates the observed peaks in February, October, and December. As previously mentioned, the industrial stations near car manufacturing facilities show a

significant negative bias. This indicates uncertainty in the emission estimates for these activities (see section 3.3.1 for more details). Regarding the weekly cycle, the model presents a flatter profile compared to the observed one. This suggests that the facilities near the stations either operate primarily during the week or have lower production on weekends.

    For background stations, we observe significantly lower values when compared to other station types. This is primarily attributable to the fact that most of these stations are located in small urban areas (e.g. Langreo in Asturias, and Mérida and



Badajoz in Extremadura) and, therefore, are not so heavily influenced by main urban hotspots such as Madrid or Barcelona. Analysing the cycles, we note that the model performs reasonably well but tends to overestimate slightly.

Table 4 provides statistical metrics for xylene, including N, observed mean, modelled mean, MB, NMB, RMSE, and r, for all stations and by station classification, both for the entire year and for each season. The annual statistics for all stations show an average xylene concentration of 1.4 $\mu$g.m$^{-3}$ with a model mean of 0.9 $\mu$g.m$^{-3}$, where the NMB is -36.7 %, indicating a mod-
erate model underestimation. The correlation coefficient (r) for the year is 0.21, suggesting a weak positive correlation while showing a moderate RMSE of 2.9 $\mu$g.m$^{-3}$. The correlation is lower during spring and summer (0.15 and 0.10, respectively) and bigger during autumn and winter (both 0.27). However, seasonal NMB values show the highest underestimation during summer (-46.4 %) and the lowest during winter (-33.1 %). The low correlation between observations and modelling results is mainly attributable to the low correlation registered at industrial stations.

Traffic stations exhibit a high seasonal variability, with winter having the highest observed mean concentration (2.3 $\mu$g.m$^{-3}$) and spring/summer (0.9 $\mu$g.m$^{-3}$) the lowest. Modelled means show 1.2 $\mu$g.m$^{-3}$ during winter and for spring/summer 0.7 $\mu$g.m$^{-3}$. During winter, the model performs the poorest with the highest NMB at -47.8 % but maintains a moderate correlation (r) of 0.46. During spring, the model shows the lowest NMB (-19.6 %) but also in terms of correlation (0.21). The low correlation implies that the model during the spring season does not accurately capture the variations in xylene concentrations, which might be missing important non-linear patterns or variability more pronounced during spring. Despite this, traffic stations
generally have the highest correlation values.

Among all the classifications, industrial areas exhibit the biggest annual observed (2.4 $\mu$g.m$^{-3}$) and modelled (1.1 $\mu$g.m$^{-3}$) values but also the most significant underestimation, with an NMB of -53.0 % and the highest RMSE of 5 $\mu$g.m$^{-3}$. Additionally, these areas consistently display the lowest correlation, which is generally weak across all seasons. The model consistently
underestimates xylene concentrations in industrial areas throughout the year, with the highest NMB occurring during summer (-70.4 %) and the lowest during winter (-30.0 %). The weak model performance in these areas could be due to underestimating emission sources but also linked to specific industrial operations that are not captured by our model, such as maintenance activities or leaks.

For background stations, unlike the other station classifications, the model tends to overestimate with a remarkably high
NMB of 146.2 %. In terms of seasonality, the largest overestimations occur during autumn exhibiting the highest MB (0.5 $\mu$g.m$^{-3}$). The correlation between observed and modelled values generally leans towards the positive but remains weak, ranging from 0.27 to 0.38, except for winter, where the correlation drops to approximately 0. This particular anomaly in winter can be primarily attributed to big peaks observed in a specific station in Asturias (ES1353A). Filtering these values that could be affected by specific surrounding activities from a car manufacturing facility.





**Figure 13.** Observed (black line) and modelled (blue line) xylene hourly, weekly, and monthly cycles ($\mu$g.m$^{-3}$) per station classification. The shaded region corresponds to the standard deviation variability.



**Table 4.** Seasonal and annual statistics for N, observed mean, modelled mean, MB, NMB, RMSE and r, per station classification for xylene in Spain in 2019 obtained with MONARCH.

| Station classification | Period | Statistical metrics | | | | | | |
|---|---|---|---|---|---|---|---|---|
| | | N | Obs. mean | Mod. mean | MB | NMB | RMSE | r |
| | | (-) | ($\mu g/m^3$) | ($\mu g/m^3$) | ($\mu g/m^3$) | (%) | ($\mu g/m^3$) | (-) |
| **All Stations** | Annual | 10159 | 1.4 | 0.9 | -0.5 | -36.7 | 2.9 | 0.21 |
| | Winter (DJF) | 2435 | 1.8 | 1.2 | -0.6 | -33.1 | 3.0 | 0.27 |
| | Spring (MAM) | 2585 | 1.1 | 0.7 | -0.4 | -33.3 | 3.0 | 0.15 |
| | Summer (JJA) | 2594 | 1.2 | 0.6 | -0.6 | -46.4 | 2.9 | 0.10 |
| | Autumn (SON) | 2545 | 1.4 | 0.9 | -0.5 | -35.5 | 2.7 | 0.27 |
| **Traffic** | Annual | 4805 | 1.4 | 0.9 | -0.5 | -39.0 | 1.7 | 0.49 |
| | Winter (DJF) | 1130 | 2.3 | 1.2 | -1.1 | -47.8 | 2.5 | 0.46 |
| | Spring (MAM) | 1204 | 0.9 | 0.7 | -0.2 | -19.6 | 1.1 | 0.21 |
| | Summer (JJA) | 1234 | 0.9 | 0.7 | -0.3 | -30.1 | 0.9 | 0.39 |
| | Autumn (SON) | 1237 | 1.6 | 0.9 | -0.7 | -43.5 | 1.9 | 0.56 |
| **Industrial** | Annual | 2717 | 2.4 | 1.1 | -1.3 | -53.0 | 5.0 | 0.08 |
| | Winter (DJF) | 675 | 2.5 | 1.7 | -0.7 | -30.0 | 4.4 | 0.14 |
| | Spring (MAM) | 661 | 2.4 | 0.9 | -1.5 | -60.9 | 5.7 | 0.12 |
| | Summer (JJA) | 690 | 2.6 | 0.8 | -1.9 | -70.4 | 5.5 | -0.01 |
| | Autumn (SON) | 691 | 2.2 | 1.1 | -1.1 | -49.2 | 4.4 | 0.12 |
| **Background** | Annual | 2637 | 0.3 | 0.6 | 0.4 | 146.2 | 1.0 | 0.19 |
| | Winter (DJF) | 630 | 0.4 | 0.8 | 0.4 | 104.1 | 1.3 | 0.05 |
| | Spring (MAM) | 720 | 0.2 | 0.6 | 0.3 | 148.2 | 0.8 | 0.34 |
| | Summer (JJA) | 670 | 0.2 | 0.5 | 0.3 | 141.4 | 0.6 | 0.27 |
| | Autumn (SON) | 617 | 0.2 | 0.7 | 0.5 | 216.8 | 1.1 | 0.38 |

## 3.3 Sensitivity analysis

Based on the discussion and findings presented in Section 3.2, we have conducted a series of sensitivity runs to explore and identify less-represented or poorly understood processes and emissions from specific sectors. The selected sectors, including the manufacturing industry (i.e., refineries, coke ovens, and car manufacturing facilities) and road transport (i.e., mopeds and motorcycles), were previously identified as potentially driving some of the main discrepancies between observed and modelled results. This identification was based on their proximity to air quality stations that were consistently underestimating, leading





to the hypothesis that these sectors are likely the primary contributors to the observed bias. To assess this, we either estimated alternative emissions (using available information on EF and activity data) or used different available datasets. The following subsections describe for each sector the approaches considered to derive alternative emission estimates and their impact on the modelling results.

### 3.3.1 Industrial Emissions: Refineries, Coke Ovens, and Car Manufacturing Industries

The sensitivity analysis for the industrial sector focuses on emissions from three types of facilities: refineries, coke ovens and car manufacturing industries. As previously discussed in Section 3.2.1 and 3.2.3, the benzene and xylene industrial stations where the model exhibits the largest bias are all located near these types of facilities. Figure G1 in Appendix G displays the locations of the facilities and air quality stations nearby that are considered for the sensitivity analysis.

The HERMESv3 plant-level industrial emissions considered in the results discussed in Section 3.2 are derived from the national Large Point Sources (LPS) database for refineries, car manufacturing industries and one coke oven facility, and from the Spanish Pollutant Release and Transfer Register (PRTR-Spain) for the remaining coke oven facilities. Both databases are compiled and maintained by the Spanish Ministry for the Ecological Transition and the Demographic Challenge (Ministerio para la Transición Ecológica y el Reto Demográfico, MITERD), which estimate the emissions using the information provided by the corresponding industrial facilities (MITERD, 2023, 2022). It is important to note that the NMVOC emissions reported by PRTR are generally calculated or estimated, while for $NO_2$ the values are measured and, in fewer cases, are calculated. Appendix E shows hourly, weekly, and monthly cycles per station classification for the stations measuring VOCs. As shown in Fig.E1, the model generally performs better for $NO_2$ than for VOCs. While a direct comparison may be challenging due to potential differences in sources and processes, this still highlights the considerable uncertainty in VOC emission estimates. For the sensitivity test, alternative emissions were proposed for each facility type, and they are discussed in detail later.

Figure 14 shows a comparison of the annual NMVOC emissions (i.e., BTX and other NMVOC) derived from LPS and PRTR-Spain for each Spanish refinery. Note that both LPS and PRTR-Spain only report total NMVOC emissions and that the split of these onto BTX and other VOCs was done considering the profiles reported by Oliveira et al. (2023). The comparison between the two datasets reveals dramatic discrepancies. LPS total emissions (1.6 kt) in this type of facility are more than 7 times lower than the estimated derived from PRTR-Spain (11.8 kt). Except for two facilities, PRTR-Spain consistently reports much larger emissions than LPS, with differences being up to a factor of 38. The source of these discrepancies are currently unknown as both datasets should follow a similar methodology. In all the facilities, benzene is the largest emitted species out of the BTX group (80 %). This is consistent with the fact that the speciation profiles used for the activities emitting NMVOCs in refineries (e.g., petroleum products processing or storage and handling of products) consistently account for a bigger fraction of benzene compared to toluene and xylene (Oliveira et al., 2023). For the sensitivity test, we replaced the original LPS emissions with the values reported by PRTR-Spain.

For car manufacturing facilities, total NMVOC emissions reported by LPS (1.27 kt) and PRTR-Spain (1.39 kt) are very similar. The emissions calculated in LPS for these facilities used a Tier 3 approach from the EMEP/EEA guidelines (EEA, 2019). This approach is based on surveys of individual facilities and considers the amount of VOCs produced minus the



amount of VOCs eliminated, resulting in the total VOCs emitted per facility. Considering the large underestimation of xylene concentrations in the stations located near this type of facility (Section 3.2 and the fact that independent speciation profiles from the literature indicate a similar amount of xylene emitted from this activity (between 5 % and 10 %, Oliveira et al. (2023)), we hypothesise that the low performance of the model is linked to an underestimation of the total NMVOC reported by LPS for these facilities. Therefore, we calculated alternative emissions by multiplying the EF reported by EMEP/EEA guidelines by the

annual number of cars produced at each facility in 2019 (País, 2020). New emission estimates (3.8 kt) are around 240 % larger than the LPS estimates (1.1 kt) (see Fig. 14). The considerable differences observed when applying the different approaches, i.e. tier 1 and tier 3, show the significant uncertainty associated with VOC emission estimation. The differences could be due to a missing fraction of the products used in the mass balance or uncertainties in the emission processes and emission factors. For the paint application activity, the speciation profiles used reports zero benzene, so as a result, no impacts are expected on

benzene observations. Consequently, this sensitivity analysis for these facilities does not address the underestimations observed for benzene. It is hypothesised that emissions from car manufacturing facilities are underestimated, although we were unable to estimate emissions for other activities.

For coke ovens, no alternative emission estimate exists besides the ones provided by PRTR-Spain. For two of the three facilities, the emissions reported by PRTR were measured and therefore should not be underestimated. The fugitive emissions

from several activities in coke oven plants were identified to be a significant source of pollution (Aries et al., 2007). So, we hypothesise that the large benzene bias observed in the two industrial stations located near coke oven facilities (see Sect. 3.2) is linked to the non-inclusion of fugitive emissions in the PRTR-Spain (i.e. from charging, door and lid leaks, off-take leaks). This hypothesis is consistent with the fact that most of these NMVOC fugitive emissions are composed of benzene, with an average contribution of 57 % (Aries et al., 2007). To fill in this gap, we performed a bottom-up estimation of these emissions

using information on the total Spanish coke production for 2019 (MITECO, 2019), the production capacity of each facility to distribute the total production and EF sourced from (Aries et al., 2007) and references therein, covering five distinct activities linked to fugitive emissions: fugitive door (7.1 g.t$^{-1}$ of coke), charging hole lid (4.4 g.t$^{-1}$ of coke), soaking (7.1 g.t$^{-1}$ of coke), and coal charging (7.7 g.t$^{-1}$ of coke). This increased total NMVOC and benzene emissions in coke ovens of 32 t (+ 25 %) and 21 t (+ 33 %), respectively (Fig. 14). Compared to the increases in emissions for the other sectors, the addition of the fugitive

emissions in coke ovens is expected to have a limited impact on the results.





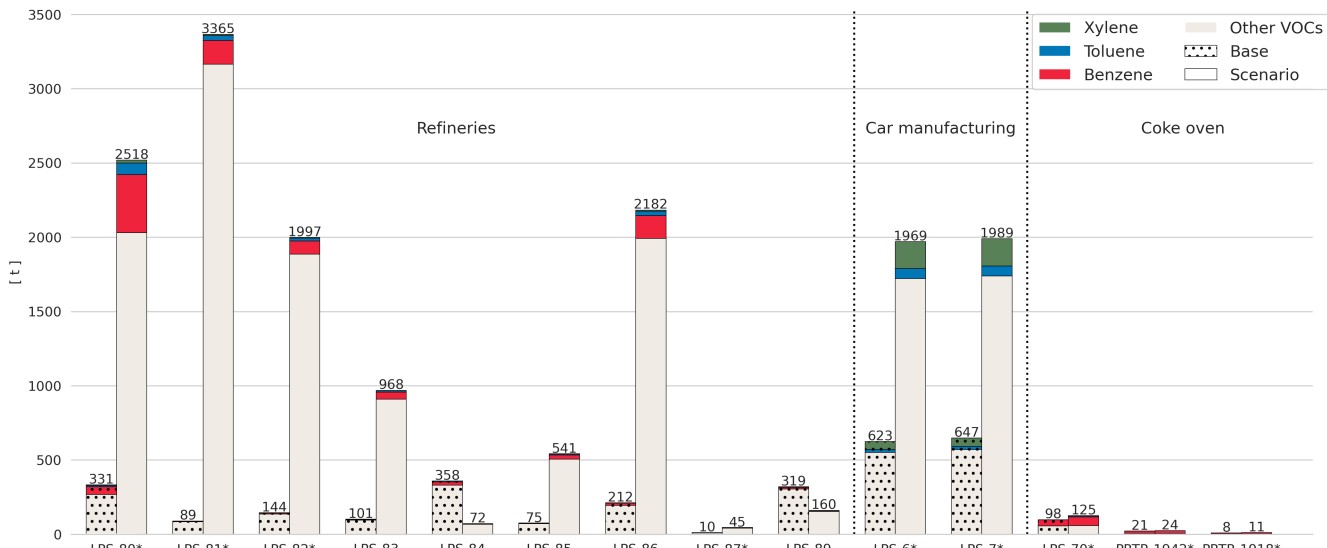

**Figure 14.** Comparison of total VOC emissions (t) in 2019 for the baseline and for the different facilities in Spain accounted in the industrial scenario. The code indicates the HERMESv3 code of the facility and the * indicates that the facility is located near an industrial air quality monitoring station.

Figure 15 shows the comparison between observed and modelled benzene, toluene, and xylene averaged monthly cycles at each selected industrial air quality stations, and also including the variability of the standard deviation as the shaded region (see Fig. G1). Modelled concentrations include the results obtained using the baseline and modified emissions. To ensure the robustness of our sensitivity analysis, observed values were filtered to exclude outliers by retaining data points within three standard deviations from the mean (i.e. z-score threshold of 3). Complementary, Table 5 and Table 6 summarises the comparison between annual statistics (MB, NMB, RMSE, and r) obtained in each station when using each emission dataset.




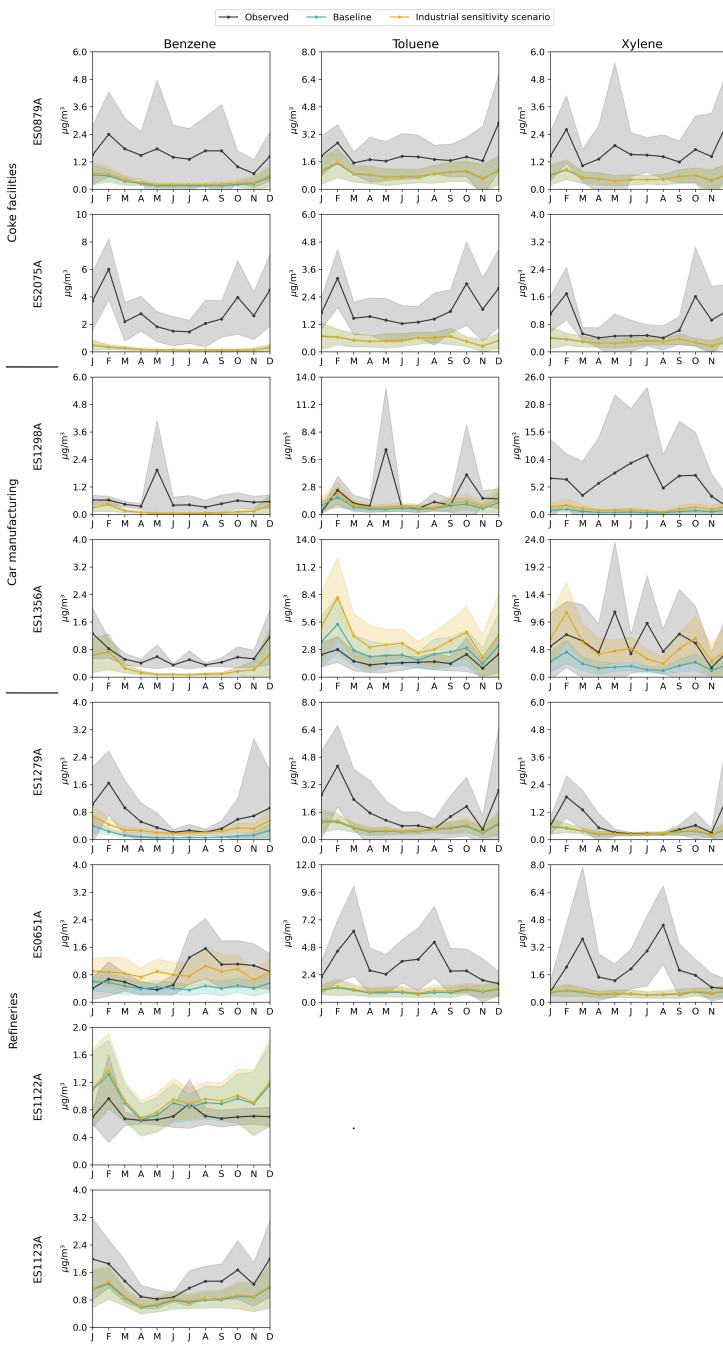

**Figure 15.** Comparison of the monthly variability of benzene, toluene and xylene concentrations from baseline (blue) and the industrial sensitivity scenario (orange) against observations (black) in 2019 for each industrial monitoring stations. The shaded region corresponds to the standard deviation variability.





For benzene, the impact of modifying the industrial emissions on the model performance is heterogeneous across stations. Considerable improvements are observed in stations next to refineries, while the ones located near coke ovens continue presenting similar underestimations. Overall, industrial benzene concentrations remain significantly underestimated, the NMB being shifted from -59.5 % to -50.7 %, reflecting a modest improvement primarily attributed to the limited increases in emissions from coke ovens (see Fig. 14). The low improvements observed in stations near coke ovens (distance below 700m) and the fact that these stations report the highest annual mean concentrations indicate that the refined emissions are still missing relevant processes emitting benzene in these types of facilities. The improvements are considerably more pronounced for stations close to refineries when compared to those near coke ovens. For instance, at station ES1279A, the NMB decreased from -76.4 % to -43.2 %, and the RMSE improved from 0.8 to 0.7 $\mu$g.m$^{-3}$. As previously mentioned, the changes in emissions affect total NMVOCs, so the impacts observed for each VOC species vary due to the speciation profile used. For example, for station ES0615 (see Fig. 15), the model's performance significantly improved for benzene, reducing the NMB from -45.9 % to -2.6 % and increasing the correlation from 0.31 to 0.47. In contrast, the impacts on toluene and xylene are minimal. This could indicate a limitation arising from the speciation profiles used, although the fractions were similar compared to other available profiles. The impact is less pronounced for the other stations near refineries, likely due to their positioning relative to the emission sources (distance between 2 km and 3 km).





**Table 5.** Comparison of annual statistical metrics (N, observed mean, modelled mean, MB, NMB, RMSE, and r) for benzene in Spain in 2019 obtained with MONARCH for the baseline and the industrial sensitivity analysis.

| Station reference | Run | Benzene | | | | | |
|---|---|---|---|---|---|---|---|
| | | Obs. mean | Mod. Mean | MB | NMB | RMSE | r |
| | | ($\mu g/m^3$) | ($\mu g/m^3$) | ($\mu g/m^3$) | (%) | ($\mu g/m^3$) | (-) |
| **Refineries** | | | | | | | |
| ES1279A | Baseline | 0.6 | 0.1 | -0.4 | -76.4 | 0.8 | 0.48 |
| | Industrial scenario | | 0.3 | -0.3 | -43.2 | 0.7 | 0.40 |
| ES0651A | Baseline | 0.9 | 0.5 | -0.4 | -45.9 | 0.7 | 0.31 |
| | Industrial scenario | | 0.9 | 0.0 | -2.6 | 0.6 | 0.47 |
| ES1122A | Baseline | 0.7 | 0.9 | 0.2 | 32.9 | 0.5 | 0.09 |
| | Industrial scenario | | 1.0 | 0.3 | 38.9 | 0.5 | 0.09 |
| ES1123A | Baseline | 1.3 | 0.9 | -0.5 | -34.4 | 0.7 | 0.56 |
| | Industrial scenario | | 0.9 | -0.4 | -31.9 | 0.7 | 0.55 |
| **Coke oven** | | | | | | | |
| ES0879A | Baseline | 1.4 | 0.3 | -1.1 | -78.3 | 1.6 | 0.20 |
| | Industrial scenario | | 0.4 | -1.0 | -73.8 | 1.5 | 0.21 |
| ES2075A | Baseline | 2.8 | 0.2 | -2.6 | -92.8 | 3.3 | 0.43 |
| | Industrial scenario | | 0.2 | -2.6 | -92.1 | 3.2 | 0.42 |
| **All Industrial** | Baseline | 1.0 | 0.4 | -0.6 | -59.5 | 1.2 | 0.18 |
| | Industrial scenario | | 0.5 | -0.5 | -50.7 | 1.1 | 0.17 |

For toluene, Table 6 compares statistical metrics for the baseline and the industrial sensitivity analysis for the two stations near car manufacturing and the overall impact on industrial stations. For the station ES1298A, the MB and NMB slightly decreased from -0.4 $\mu g.m^{-3}$ to -0.2 $\mu g.m^{-3}$ and -34.0 % to -17.8 %, respectively. At station ES1356A, the model was already overestimating with an NMB of 56.6 %, and now it has increased to 128.4 %, while the correlation increased from 0.76 to 0.77. The degradation of the model's performance in this station can be explained mainly by overestimations from the solvent sector in some areas previously identified in Section 3.2.2. So, when we increase the emissions from this facility, the overestimations are even larger (see Fig. 15).

For xylene, the adjustment of industrial emissions improved the model's capability to reproduce the observed monthly cycle for stations near car manufacturing facilities (see Fig. 15). Table 6 summarises the changes in the MONARCH's performance at the two stations close to the car manufacturing industries, as these are the facilities where xylene emissions have increased the most when adjusting the industrial VOC emissions (see Fig. 14). Improvements in MB, NMB and RSME can be observed in both locations, but most notably in ES1356A, with the MB and NMB decreasing from -3.2 $\mu g.m^{-3}$ to -0.1 $\mu g.m^{-3}$ and from -60.6 % to -1.0 %, respectively. For station ES1298A, there are slight improvements, the NMB decreases from -89.0





550  % to -79.0 %, and the RMSE reduces from 8.2 $\mu$g.m$^{-3}$ to 7.9 $\mu$g.m$^{-3}$, although the model is still highly underestimated. Despite the overall improvements, it is evident that the model still underestimates xylene concentrations, suggesting potential missing sources or emissions underestimations. It is worth noting that in both stations, the model still struggles to reproduce the changes in observed concentrations as the the correlation stays very low. This is mainly due to frequent high observed episodes, evidenced by the highest standard deviation variability. Daily means exceed 15 $\mu$g.m$^{-3}$ in both stations, even after

removing outliers. These occurrences may be linked to specific activities within the nearby facilities, which are not accounted in our model.

**Table 6.** Comparison of annual statistical metrics (observed mean, modelled mean, MB, NMB, RMSE, and r) for toluene and xylene in Spain in 2019 obtained with MONARCH for the baseline and the industrial sensitivity analysis.

| | Station reference | Run | Toluene | | | | | | Xylene | | | | | |
|---|---|---|---|---|---|---|---|---|---|---|---|---|---|---|
| | | | Obs. mean | Mod. Mean | MB | NMB | RMSE | r | Obs. mean | Mod. Mean | MB | NMB | RMSE | r |
| | | | ($\mu g/m^3$) | ($\mu g/m^3$) | ($\mu g/m^3$) | (%) | ($\mu g/m^3$) | (-) | ($\mu g/m^3$) | ($\mu g/m^3$) | ($\mu g/m^3$) | (%) | ($\mu g/m^3$) | (-) |
| Car manuf. | ES1298A | Baseline | 1.3 | 0.9 | -0.4 | -34.0 | 1.3 | 0.38 | 5.2 | 0.6 | -4.7 | -89.0 | 8.2 | 0.05 |
| | | Industrial scenario | | 1.1 | -0.2 | -17.8 | 1.3 | 0.37 | | 1.1 | -4.1 | -79.0 | 7.9 | 0.01 |
| | ES1356A | Baseline | 1.7 | 2.6 | 0.9 | 56.6 | 1.7 | 0.76 | 5.2 | 2.1 | -3.2 | -60.6 | 6.2 | 0.28 |
| | | Industrial scenario | | 3.8 | 2.1 | 128.4 | 3.0 | 0.77 | | 5.2 | -0.1 | -1.0 | 5.9 | 0.27 |

### 3.3.2 Road Traffic Sector: Mopeds and Motorcycles

The sensitivity analysis concerning the traffic sector, particularly emissions from mopeds and motorcycles, arises from the underestimations identified in Section 3.2. We identified that urban traffic stations in large cities, such as Barcelona and Va-

lencia, exhibited significant underestimations for benzene, while in other major urban areas, such as Madrid, the modelled and observed results are pretty in line. This discrepancy suggests a potential misrepresentation of local traffic sources characteristic of these cities, specifically due to the significant contribution of mopeds and motorcycles to the road transport sector, which is further described next.

    Figure 16 shows the monthly comparison of NO$_2$ and benzene concentrations for urban traffic stations in Madrid, Barcelona

and Valencia. For the traffic stations located in Madrid, the model was able to reasonably reproduce both NO$_2$ (NMB = -23.96 %) and benzene (NMB = -9.64 %) levels. In the case of Barcelona and Valencia, we observed that while NO$_2$ levels (NMB = -24.77 %, and NMB = -14.77 %, respectively) are also reasonably well reproduced, benzene concentrations are significantly underestimated (NMB = -69.97 %, and NMB = -76.53 %, respectively). Measured benzene levels in Barcelona are much higher (2.81 $\mu$g.m$^{-3}$) than the ones reported for Madrid (0.51 $\mu$g.m$^{-3}$). This is a characteristic that MONARCH fails to reproduce, as

the modelled concentrations are similar in all urban traffic sites. Observed benzene concentrations in Barcelona and Valencia achieve their maximum values during winter (up to 5 $\mu$g.m$^{-3}$), showing a marked seasonality that is not registered in any of the Madrid urban traffic stations and that the model is not capable of reproducing. Notably, looking at background levels in Barcelona, as seen in station ES1856A (see Fig. G2), the model can perform reasonably well for both NO$_2$ (NMB = 13.26 %) and benzene (NMB = -18.20 %). For Valencia, there are no available background stations measuring benzene. Since the



background levels are correctly reproduced in Barcelona, this leads to the conclusion that the main issue should be related to a local traffic source.

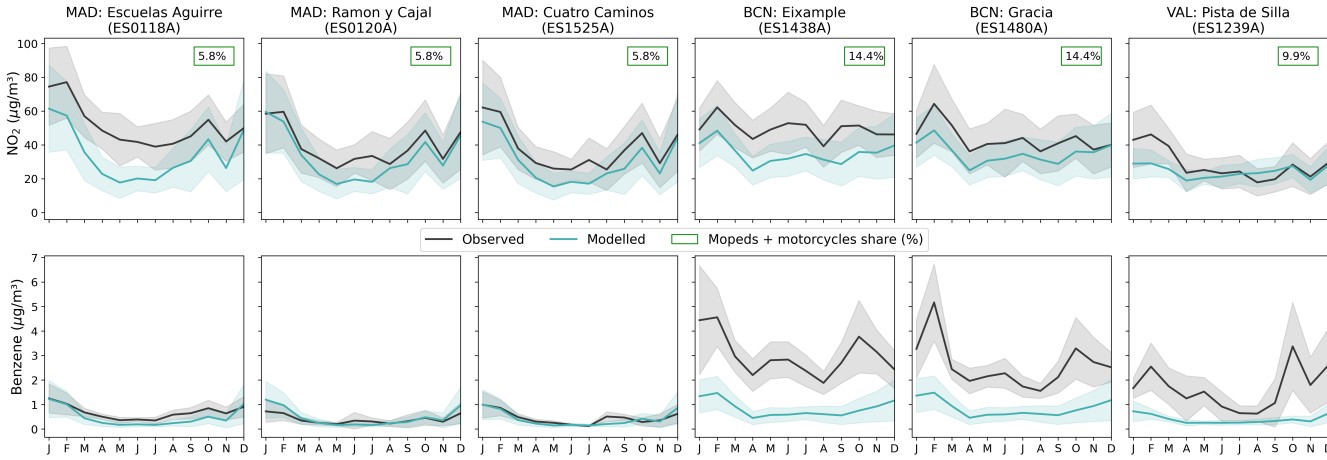

**Figure 16.** Monthly comparison of $NO_2$ and benzene concentrations ($\mu$g.m$^{-3}$) at urban traffic stations in Madrid (MAD), Barcelona (BCN), and Valencia (VAL). The share of mopeds and motorcycles per city used in HERMESv3 is included.

Focusing on Madrid and Barcelona, both exhibit significantly different fleet compositions, with Madrid having a moped and motorcycle share of 5.8 % and Barcelona having a share of 14.4 %. Moreover, according to HERMESv3 results in Barcelona, mopeds and motorcycles contribute around 63 % to road transport benzene emissions. These results are in line with the remote

sensing results obtained by Dallmann et al. (2019) for Paris in 2015, where the contribution of motorised 2-wheeled vehicles was 46 % of road transport non-methane hydrocarbon emissions. Moreover, a quick analysis of measured benzene in European urban traffic stations showed that similar high annual mean values are found in other Mediterranean coastal cities, such as Genoa, Italy (2.70 $\mu$g.m$^{-3}$) and Athens, Greece (2.81 $\mu$g.m$^{-3}$). Notably, these countries are also known for having a significant share of mopeds and motorcycles (Yannis et al., 2007; Scorrano and Danielis, 2021). This reinforces the hypothesis of a possible

benzene emission underestimation from mopeds and motorcycles.

The EF from mopeds and motorcycles reported by "COmputer Programme to calculate Emissions from Road Transport version 5" (COPERT 5), which are used in HERMESv3, seem to align with alternative estimates (Saxer et al., 2006; Adam et al., 2010). Therefore, we assume the issue is related to the degradation factor, which in HERMESv3 is considered for gasoline and diesel-powered vehicles but not mopeds and motorcycles.

Following the methodology presented by (Ntziachristos et al., 2018) for petrol cars and light commercial vehicles, we used Eq. 1 to estimate correction factors accounting for different vehicle ages. To accomplish this, we utilised the linear regression constants for hydrocarbons (HC) emission factors based on odometer mileage (km) as provided by Tsai et al. (2018). We calculated the mean mileage for each moped/motorcycle category using the data provided by the ministry on kilometres travelled per category and year. The values used per vehicle category are in the supplementary material in Table G1.





$$M_{corr} = AM \times M_{MEAN} + BM \tag{1}$$

Where, $M_{corr}$ is the mileage correction factor for a given mileage ($M_{MEAN}$), $M_{MEAN}$ is the mean fleet mileage of vehicles for which correction is applied (km), $AM$ is the degradation of the emission performance per kilometre, and $BM$ is the emission level of a fleet of brand new vehicles. $AM$ and $BM$ are assumed to be constant and do not depend on the vehicle category.

Implementing the mileage degradation for VOCs increased mopeds and motorcycle emissions by a factor of 2-3. Despite this, the impact on air quality results is quite limited, with the model still significantly underestimating observed values in Barcelona and Valencia. Table 7 presents a comparison between the baseline and sensitivity analysis scenario which incorporates the mileage degradation. The table summarises annual statistics (MB, NMB, RMSE, and r) obtained for traffic stations in Madrid, Barcelona, and Valencia. Overall, the NMB reduced (between -0.4 % and 1.0 %) for all the stations except for station ES1525A, 605 located in Madrid, where the performance was slightly degraded (from NMB = 13.5 % to NMB= 14.8 %) with the scenario. Station ES1438A is the only station where the correlation increased, rising from 0.37 to 0.38.

We believe the model coarse spatial resolution (0.1° by 0.1°) may be playing a significant role in these results, as it is not capable of capturing the large influence of mopeds and motorcycles in the urban traffic areas of the city. Unlike vehicle passenger cars, whose spatial distribution across Barcelona's urban fabric is relatively homogeneous, mopeds and motorcycles 610 in Barcelona mainly circulate in the inner city, where the two air quality urban traffic stations are located (Rodriguez-Rey et al., 2022). Performing simulations at a much finer spatial resolution (i.e., 1km or less) would be needed to confirm this point. Despite this, it is evident that the emissions are still underestimated, suggesting that some sources are either not accurately represented in our model or are unaccounted for. So, to improve the model's performance in these sites, it would be important to include other relevant sources that could differentiate cities. For instance, Yang et al. (2021) estimated that uphill driving 615 can significantly impact EF, with an estimated factor of approximately 12 compared to baseline or downhill driving. Due to the different topography of Barcelona compared to Madrid, introducing this could in the model could lead to further improvements in the performance of the model.





**Table 7.** Comparison of annual statistical metrics (N, observed mean, modelled mean, MB, NMB, RMSE, and r) for benzene in Spain in 2019 obtained with MONARCH for the baseline and the mileage degradation sensitivity analysis.

| Station reference | | Run | Benzene | | | | | |
| | | | Obs. mean $(\mu g/m^3)$ | Mod. Mean $(\mu g/m^3)$ | MB $(\mu g/m^3)$ | NMB (%) | RMSE $(\mu g/m^3)$ | r (-) |
|---|---|---|---|---|---|---|---|---|
| Madrid | ES0118A | Baseline | 0.7 | 0.5 | -0.2 | -24.2 | 0.4 | 0.79 |
| | | Scenario | | 0.5 | -0.2 | -23.3 | 0.4 | 0.79 |
| | ES0120A | Baseline | 0.4 | 0.4 | 0.0 | -6.5 | 0.3 | 0.69 |
| | | Scenario | | 0.4 | 0.0 | -5.9 | 0.3 | 0.69 |
| | ES1525A | Baseline | 0.5 | 0.5 | 0.1 | 13.5 | 0.4 | 0.73 |
| | | Scenario | | 0.5 | 0.1 | 14.8 | 0.4 | 0.73 |
| Barcelona | ES1438A | Baseline | 3.0 | 0.9 | -2.0 | -68.3 | 2.4 | 0.37 |
| | | Scenario | | 1.0 | -2.0 | -67.6 | 2.4 | 0.38 |
| | ES1480A | Baseline | 2.6 | 1.0 | -1.6 | -62.2 | 1.9 | 0.48 |
| | | Scenario | | 1.0 | -1.6 | -61.3 | 1.9 | 0.48 |
| Valencia | ES1239A | Baseline | 1.6 | 0.4 | -1.2 | -74.3 | 1.7 | 0.51 |
| | | Scenario | | 0.4 | -1.2 | -73.9 | 1.7 | 0.51 |

## 4 Conclusions

Volatile Organic Compounds (VOCs) significantly impact air quality and atmospheric chemistry, and are key precursors for
the formation of in ozone ($O_3$) and secondary organic aerosol (SOA). Aromatic VOCs, such as benzene, toluene, and xylene (BTX), pose distinct health risks, which further justifies the importance of understanding their variability and mitigating their negative impacts. Nevertheless, comprehensive model-based evaluation studies remain scarce in the literature.

The present work presents an assessment and evaluation of BTX primary emissions and concentrations in Spain by combining observations, emission inventories and air quality modelling techniques. We run the HERMESv3 emission model to
625 produce a gridded bottom-up inventory of VOC emissions and use it as input in the MONARCH chemical transport model to analyse the spatial and temporal variability of BTX surface concentrations across Spain. Estimated results are then compared to observed values reported by the Spanish official national air quality monitoring network. The intercomparison between modelled and observed levels allows identifying sources of uncertainty within the emission input, which we further explored through specific sensitivity test runs.
The following conclusions are obtained from the analysis of the bottom-up BTX emissions estimated with HERMESv3:





- Annual emissions of benzene, toluene and xylene over Spain reach 11 kt, 36 kt, and 25 kt, respectively. The spatial distribution of these emissions across Spanish regions varies across chemical compounds, but Andalusia, Catalonia, and the Valencian Community appear as major contributors for the three species.

- Regarding dominating sources in urban areas, benzene is primarily emitted by the industrial sector (35 - 44 %), followed by traffic (29 %). While both xylene and toluene are primarily emitted by the industrial solvent sector (contributing around 89 % for toluene and 78 % for xylene), mainly from the industrial paint application. While domestic solvent use is one of the main emitting activities of total VOCs (11 %), its contribution to BTX emissions is relatively limited, constituting less than 1 % in the speciation profiles considered.

- Regarding the spatial distribution, benzene emissions in urban areas are primarily attributed to the transport sector (over 60 %), while outside of urban areas, the residential sector dominates (> 90 %). For toluene and xylene, both urban and suburban areas are largely influenced by the industrial solvent sector (comprising 63-97 % of emissions), while in more rural areas, the livestock and residential sectors play a more significant role.

The comparison between modelled and observed BTX concentrations and sensitivity test runs lead to the following conclusions:

- Both observed and modelled Spanish benzene concentrations remain below the EU annual limit value.

- Comparisons results indicate a generally good spatial correlation between observations and model outputs. However, the highest modelled values were observed in areas lacking monitoring stations.

- The model shows a good performance in replicating benzene concentrations within non-industrial areas, albeit with a tendency to underestimate, especially during winter. In contrast, the performance diminishes noticeably in industrial zones due to uncertainties linked to specific industrial activities, such as refineries, car manufacturing facilities, and coke ovens.

- A sensitivity analysis was tested for these activities by estimating alternative emissions considering different sources. The industrial sensitivity analysis overall showed some improvements, although the model is still strongly underestimating near some industrial sites, which could be linked to underestimations in the total emissions. While, in some cases, the EF may be more generic, deviating from industry-specific nuances and leading to uncertainty.

- The model's more pronounced underestimations identified during winter can be attributed to several uncertainties and modelling limitations. For instance, in urban and suburban areas, underestimations are primarily influenced by residential wood combustion, which is less relevant in large cities. Additionally, it is important to note that the emission model only accounts for cold start emissions in Madrid and Barcelona. Therefore, the model's performance in traffic stations during winter could be enhanced by incorporating cold start emissions in all urban areas.





- Observed benzene levels in Barcelona's urban traffic areas were five times larger than the ones observed in Madrid. As for the background station in Barcelona, the model is performing well; this indicates a potential underestimation of mopeds and motorcycle emissions.

- Another sensitivity analysis was performed by incorporating mileage degradation factors for mopeds and motorcycles. The impact of this sensitivity analysis showed to be very limited, which in part could be related to the resolution as it fails to capture the changes in urban areas. Despite this, from the results we conclude that the emissions from mopeds and motorcycles are still underestimated although due to a lack of available literature, including EF, we could not conduct further investigations and more work is needed in this area.

- MONARCH's performance in traffic sites could be improved by incorporating additional sources related to the transport
sector. Other authors have identified relevant factors, for example, accounting for varied road grade characteristics impact on the EF (Yang et al., 2021).

- For toluene, the model shows a good level of accuracy in replicating the hourly, monthly, and weekly cycles across various station classifications. However, it tends to underestimate the levels in traffic stations while significantly overestimating them in background stations.

- The toluene overestimations are particularly pronounced in the Madrid area and, to a lesser extent, in stations located in central northern regions, such as Cantabria and Navarre. The model's limited performance in these regions may be attributed to inadequate proxies to spatially disaggregate emissions from activities within the industrial solvent sector, namely for wood paint application which uses the population as a spatial proxy.

- For xylene, the model effectively reproduced the hourly, monthly, and weekly cycles across different station classifica-
tions. However, significant underestimations were observed, particularly for traffic stations during the winter months, where the model struggled to replicate peak concentrations.

- The most significant xylene underestimations were identified in industrial facilities near car manufacturing facilities, and these were particularly pronounced during the spring and summer months, especially on weekdays.

- The industrial sensitivity analysis showed important improvements in the model's performance for stations near car
manufacturing. However, it is still underestimating, indicating incorrect characterisation of specific activities contributing to VOC emissions.

- MONARCH performs better in reproducing $NO_2$ than VOCs in the stations considered. This could potentially be related to the fact that NOx industrial emissions are more robust due to better measured-based EF.

As highlighted in the introduction, a significant challenge in assessing the performance of the models arises from the inherent
uncertainties from emissions, such as EF and activity data, chemical mechanisms, and limitations associated with observational data. For this work, we considered a total of 47 stations measuring benzene, 34 stations measuring toluene and 30 stations



measuring xylene. The spatial distribution reveals a notable absence of monitoring stations in south and north-western Spain, as well as in Barcelona for toluene and xylene. Another prominent limitation comes from the absence of quality continuous measurements. For example, in the case of stations monitoring benzene, after applying a 75 % temporal coverage threshold

and implementing the GHOST quality filters, approximately 55 % of the available stations were dropped mainly due to the low temporal coverage. Furthermore, continuous measurements primarily target aromatic compounds, while data for other VOCs are often measured from short-duration campaigns and confined to specific locations. These limitations are crucial to consider since a comprehensive evaluation of VOC emissions and concentrations requires consistently high-quality data, both in terms of spatial and temporal coverage. In addition, the usage of satellite data to monitor VOCs is emerging, e.g. Franco et al. (2022)

obtained emission rates for ethylene from satellite data, which was linked to big industrial facilities. This could be used in the future to better assess and evaluate the spatial distribution of modelled VOCs levels and associated emissions.

To conclude, this research not only enhances our knowledge on VOCs emissions and uncertainties but also contributes to improve the performance of air quality models when simulating these compounds, which, in turn, can lead to better support for the design of effective pollution control strategies. This is crucial as the upcoming study will evaluate the effects of specific

VOC emission reduction strategies on air quality. Although the initial set of measures examined by Petetin et al. (2023) indicated limited impacts of anthropogenic VOC emission reductions on $O_3$ Spanish levels, additional scenarios will be explored, focusing on assessing the impact on PM2.5.

*Code availability.* The HERMESv3_BU code package is accessible through the GitLab repository: https://earth.bsc.es/gitlab/es/hermesv3_bu (last access: November 2023) (https://doi.org/10.5281/zenodo.3521897, Guevara et al. (2019a)). The MONARCH source code used in

this work is accessible through the GitLab repository: https://earth.bsc.es/gitlab/es/monarch (last access: November 2023) (https://doi.org/10.5281/zenodo.5215467, Klose et al. (2021b)). Both repositories have a wiki with the model instructions.

## Appendix A: MONARCH: CB05 mechanism

This Appendix provides the CB05 mechanism reactions and extensions implemented in MONARCH involving benzene (BENZENE), toluene (TOL) and xylene (XYL), and the respective species names in Table A1. Further information on the original

CB05 mechanism and reactions is available in Yarwood et al. (2005).





| Species name | Description |
|---|---|
| OH | Hydroxyl radical |
| BENZ/TOL/XYLRO2 | First generation products from TOL, XYL and BENZ that further react with NO and $NO_2$ to produce SVOC |
| HO2 | Hydroperoxy radical |
| XO2 | NO to $NO_2$ conversion from RO2 |
| CRES | Cresol and higher molecular weight phenols |
| MGLY | Methylglyoxal and other aromatic products |
| PAR | Paraffin carbon bond (C-C) |
| TO2 | Toluene-hydroxyl radical adduct |

**Table A1.** Species names used for the CB05 mechanism reactions and extensions implemented in MONARCH regarding benzene, toluene and xylene.

$$\text{BENZENE} + \text{OH} \rightarrow \text{OH} + 0.764\text{*BENZRO2} \quad (\text{Rate constant: } k = 2.47 \times 10^{-12} \times e^{-206/T}) \tag{A1}$$

$$\text{XYL} + \text{OH} \rightarrow 0.700\text{HO}_2 + 0.500\text{XO}_2 + 0.200\text{CRES} + 0.800\text{MGLY}$$
$$+ 1.100\text{PAR} + 0.300\text{TO}_2 + 0.804\text{XYLRO2} \quad (\text{Rate constant: } k = 1.7 \times 10^{-11} \times e^{-116/T}) \tag{A2}$$

$$\text{TOL} + \text{OH} \rightarrow 0.440\text{HO}_2 + 0.080\text{XO}_2 + 0.360\text{CRES} + 0.560\text{TO}_2$$
$$+ 0.765\text{TOLRO2} \quad (\text{Rate constant: } k = 1.8 \times 10^{-12} \times e^{-355/T}) \tag{A3}$$





## Appendix B: Spanish NUTS2 administrative regions

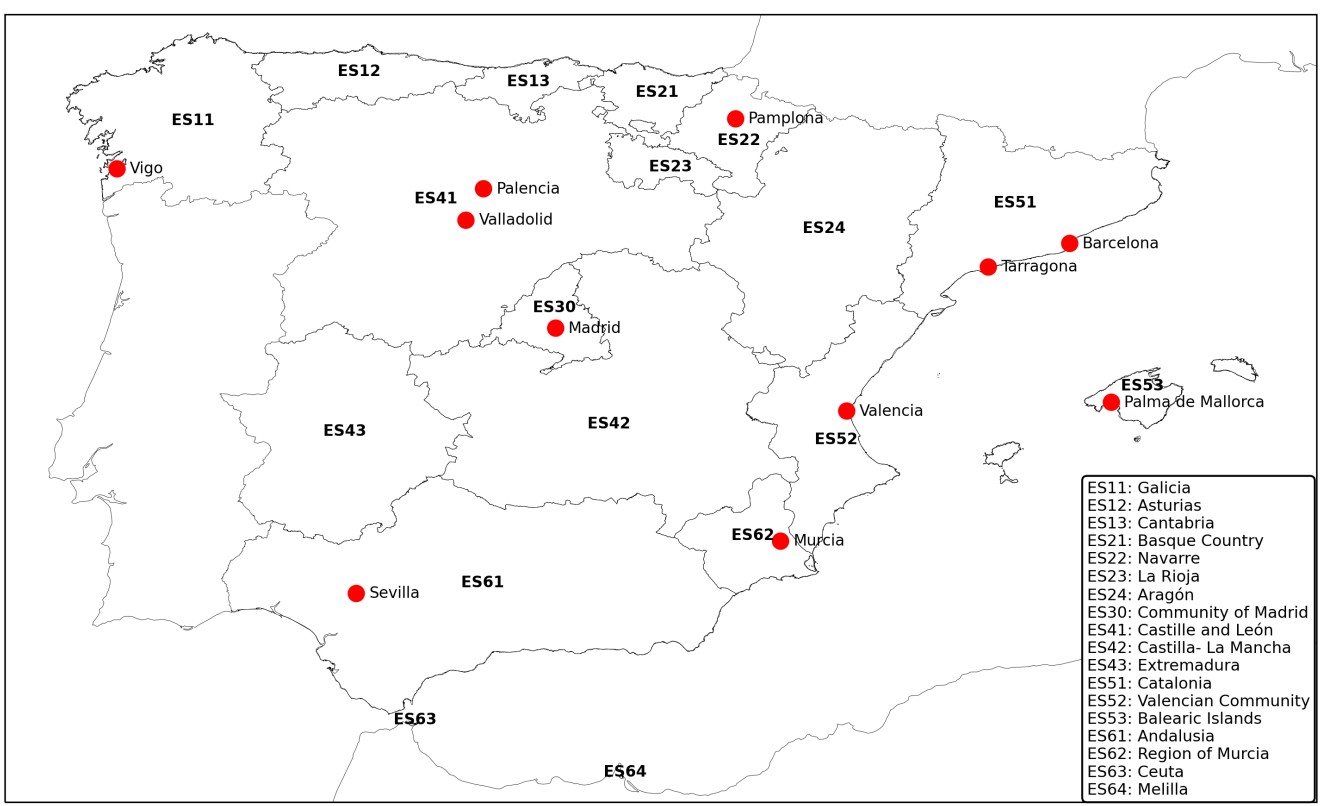

**Figure B1.** Administrative Spanish NUTS2 regions and main cities mentioned in this work.

## Appendix C: GHOST quality assurance (QA) flags

A quality assurance screening was applied to all observations using Globally Harmonised Observational Surface Treatment (GHOST) metadata, and all observations flagged were removed.





**Table C1.** Description of the GHOST quality-assurance flags used on the air quality observational data set.

| Flag | Description |
| --- | --- |
| 0 | Measurement is missing (i.e. NaN). |
| 1 | Value is infinite – occurs when data values are outside of the range that *float32* data type can handle (-3.4E+38 to +3.4E+38). |
| 2 | Measurement is negative in absolute terms. |
| 6 | Measurements are associated with data quality flags given by the data provider which have been decreed by the GHOST project architects as being associated with substantial uncertainty/bias. |
| 8 | After screening by key QA flags, no valid data remains to average in the temporal window. |
| 20 | The primary sampling is not appropriate to prepare the specific parameter for subsequent measurement. |
| 21 | The sample preparation is not appropriate to prepare the specific parameter for subsequent measurement. |
| 22 | The measurement methodology used is not known to be able to measure the specific parameter. |
| 72 | Measurement is below or equal to the preferential lower limit of detection. |
| 75 | Measurement is above or equal to the preferential upper limit of detection. |
| 82 | The preferential resolution for the measurement is coarser than a set limit (variable by measured parameter). |
| 83 | The resolution of the measurement is analysed month by month. If the minimum difference between observations is coarser than a set limit (variable by measured parameter), measurements are flagged. |
| 110 | The measured value is below or greater than scientifically feasible lower/upper limits (variable by parameter). |
| 111 | The median of the measurements in a month is greater than a scientifically feasible limit (variable by parameter). |
| 112 | Data has been reported to be an outlier through data flags by the network data reporters (and not manually checked and verified as valid). |
| 113 | Data has been found and decreed manually to be an outlier. |
| 132 | 4 out of 6 months' distributions are classed as Zone 6 or higher, suggesting there are potentially systematic reasons for the inconsistent distributions across the 6 months. |
| 133 | 8 out of 12 months' distributions are classed as Zone 6 or higher, suggesting there are potentially systematic reasons for the inconsistent distributions across the 12 months. |



## Appendix D: Emissions

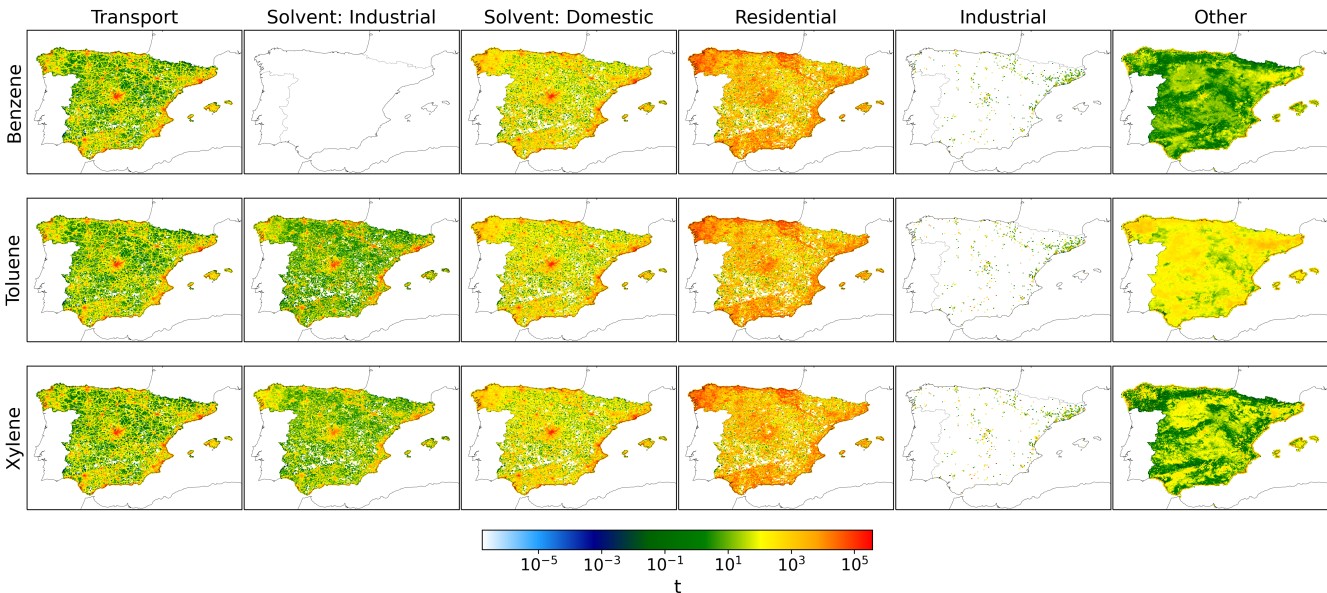

**Figure D1.** Absolute contribution (t) of individual anthropogenic emission sources to total benzene, toluene and xylene annual emissions in Spain in 2019 per grid cell (4 km by 4 km).





# Appendix E: NO$_2$ surface concentrations



**Figure E1.** Observed (black line) and modelled (blue line) NO$_2$ hourly, weekly, and monthly cycles ($\mu$g.m$^{-3}$) per station classification for stations also measuring VOCs. The shaded region corresponds to the standard deviation variability.

# Appendix F: Air quality stations

The tables in this appendix provide detailed information about each station, including the measurement methods employed. The following abbreviations are used to denote different measurement techniques: Gas Chromatography (GC), Mass Spectrometry (MS), Flame Ionisation Detection (FID), Photoionisation Detection (PID), and if the method is unknown (UNK).





**F1    Stations measuring benzene**





**Table F1.** Location and characteristics of selected stations measuring benzene in 2019.



| Station reference | Meas. method | Lon | Lat | City | Area classification | Station classification | Time resolution |
|---|---|---|---|---|---|---|---|
| 28047002 | GC-FID | -4.013 | 40.634 | Collado-Villalba | Urban | Traffic | Daily |
| 48020062 | GC-MS | -2.918 | 43.267 | Bilbao | Urban | Traffic | Daily |
| ES0118A | GC-MS | -3.682 | 40.422 | Salamanca | Urban | Traffic | Hourly, Daily |
| ES0120A | GC-MS | -3.677 | 40.452 | Chamartin | Urban | Traffic | Hourly, Daily |
| ES0126A | GC-MS | -3.732 | 40.395 | Carabanchel | Urban | Background | Hourly, Daily |
| ES0651A | GC-MS | -0.913 | 37.603 | La Union | Suburban | Industrial | Hourly, Daily |
| ES0879A | GC-MS | -5.899 | 43.550 | Aviles | Suburban | Industrial | Hourly, Daily |
| ES1122A | GC-MS | 1.237 | 41.194 | La Pobla De Mafumet | Rural | Industrial | Hourly, Daily |
| ES1123A | GC-MS | 1.218 | 41.155 | Constanti | Suburban | Industrial | Hourly, Daily |
| ES1161A | GC-MS | -5.587 | 42.604 | Leon | Urban | Traffic | Hourly, Daily |
| ES1193A | GC-MS | -3.749 | 40.420 | Moncloa-Aravaca | Suburban | Background | Hourly, Daily |
| ES1239A | GC-MS | -0.376 | 39.456 | Valencia | Urban | Traffic | Hourly, Daily |
| ES1244A | GC-MS | -2.935 | 43.268 | Bilbao | Urban | Traffic | Hourly, Daily |
| ES1269A | GC-MS | -5.833 | 43.366 | Oviedo | Urban | Traffic | Hourly, Daily |
| ES1272A | GC-MS | -5.673 | 43.530 | Gijon | Urban | Traffic | Hourly, Daily |
| ES1279A | GC-PID | -4.089 | 38.682 | Puertollano | Suburban | Industrial | Hourly, Daily |
| ES1298A | GC-FID | -4.494 | 41.961 | Villamuriel De Cerrato | Suburban | Industrial | Daily |
| ES1353A | GC-MS | -5.680 | 43.294 | Sama | Urban | Background | Hourly, Daily |
| ES1356A | GC-MS | -4.741 | 41.613 | Laguna De Duero | Suburban | Industrial | Daily |
| ES1438A | GC-MS-FID | 2.154 | 41.385 | Barcelona | Urban | Traffic | Daily |
| ES1472A | GC-MS | -1.651 | 42.807 | Iturrama | Urban | Background | Hourly, Daily |
| ES1480A | GC-MS-FID | 2.153 | 41.399 | Gracia | Urban | Traffic | Daily |
| ES1502A | GC-MS | -2.681 | 42.855 | Gasteiz / Vitoria | Urban | Traffic | Hourly, Daily |
| ES1525A | GC-MS | -3.706 | 40.445 | Chamberi | Urban | Traffic | Hourly, Daily |
| ES1536A | GC-MS | -3.265 | 40.571 | Azuqueca De Henares | Suburban | Background | Hourly |
| ES1564A | GC-FID | -3.645 | 40.540 | Alcobendas | Urban | Traffic | Hourly, Daily |





| ES1565A | GC-FID | -3.801 | 40.282 | Fuenlabrada | Urban | Industrial | Hourly, Daily |
|---------|--------|--------|--------|-------------|-------|------------|---------------|
| ES1580A | GC-FID | -3.809 | 43.461 | Santander | Urban | Traffic | Hourly, Daily |
| ES1601A | GC-MS | -7.011 | 38.888 | Badajoz | Urban | Background | Hourly, Daily |
| ES1602A | GC-MS | -2.428 | 42.464 | Logrono | Urban | Background | Hourly, Daily |
| ES1610A | GC-MS-FID | 2.656 | 39.570 | Palma | Urban | Traffic | Hourly, Daily |
| ES1627A | GC-MS | -1.231 | 37.976 | Alcantarilla | Suburban | Industrial | Hourly, Daily |
| ES1631A | GC-MS | -4.730 | 41.646 | Valladolid | Urban | Traffic | Hourly, Daily |
| ES1633A | GC-MS | -1.145 | 37.994 | Murcia | Suburban | Traffic | Hourly, Daily |
| ES1635A | GC-MS | -0.472 | 38.359 | Alicante | Urban | Traffic | Hourly, Daily |
| ES1802A | GC-FID | -3.468 | 40.909 | Patones | Rural | Background | Hourly, Daily |
| ES1803A | GC-FID | -4.014 | 40.634 | Collado-Villalba | Urban | Traffic | Hourly, Daily |
| ES1819A | GC-MS | -6.338 | 38.907 | Merida | Urban | Background | Hourly, Daily |
| ES1834A | GC-MS | -0.026 | 39.989 | Castello De La Plana | Urban | Traffic | Hourly, Daily |
| ES1849A | GC-FID | -0.718 | 38.259 | Elche | Urban | Traffic | Hourly, Daily |
| ES1856A | GC-MS-FID | 2.148 | 41.426 | Horta-Guinardo | Urban | Background | Daily |
| ES1903A | GC-MS | 2.014 | 41.313 | Viladecans | Suburban | Background | Hourly, Daily |
| ES1910A | UNK | 1.992 | 41.303 | Gava | Suburban | Background | Hourly, Daily |
| ES1942A | GC-MS | -3.581 | 40.462 | Barajas De Madrid | Urban | Background | Hourly, Daily |
| ES1983A | GC-MS | 2.082 | 41.322 | El Prat De Llobregat | Suburban | Background | Hourly, Daily |
| ES1986A | GC-MS | -3.689 | 41.666 | Aranda De Duero | Urban | Traffic | Hourly, Daily |
| ES2075A | GC-MS | -5.970 | 43.346 | Castandiello | Suburban | Industrial | Hourly, Daily |

**F2  Stations measuring toluene**





**Table F2.** Location and characteristics of selected stations measuring toluene in 2019.

| Station reference | Meas. method | Lon | Lat | City | Area classification | Station classification | Time resolution |
|---|---|---|---|---|---|---|---|
| ES0118A | GC-MS | -3.682 | 40.422 | Salamanca | Urban | Traffic | Hourly, Daily |
| ES0126A | GC-MS | -3.732 | 40.395 | Carabanchel | Urban | Background | Hourly, Daily |
| ES0651A | GC-MS | -0.913 | 37.603 | La Union | Suburban | Industrial | Hourly, Daily |
| ES0879A | GC-MS | -5.899 | 43.550 | Aviles | Suburban | Industrial | Daily |
| ES1161A | GC-MS | -5.587 | 42.604 | Leon | Urban | Traffic | Hourly, Daily |
| ES1193A | GC-MS | -3.749 | 40.420 | Moncloa-Aravaca | Suburban | Background | Hourly, Daily |
| ES1239A | GC-MS | -0.376 | 39.456 | Valencia | Urban | Traffic | Hourly, Daily |
| ES1269A | GC-MS | -5.833 | 43.367 | Oviedo | Urban | Traffic | Daily |
| ES1272A | GC-MS | -5.673 | 43.530 | Gijon | Urban | Traffic | Daily |
| ES1279A | GC-MS | -4.089 | 38.682 | Puertollano | Suburban | Industrial | Hourly, Daily |
| ES1298A | GC-FID | -4.494 | 41.961 | Villamuriel De Cerrato | Suburban | Industrial | Daily |
| ES1353A | GC-MS | -5.680 | 43.294 | Sama | Urban | Background | Daily |
| ES1356A | GC-MS | -4.741 | 41.613 | Laguna De Duero | Suburban | Industrial | Daily |
| ES1472A | GC-PID | -1.651 | 42.807 | Iturrama | Urban | Background | Hourly, Daily |
| ES1525A | GC-MS | -3.706 | 40.445 | Chamberi | Urban | Traffic | Hourly, Daily |
| ES1536A | GC-MS | -3.265 | 40.571 | Azuqueca De Henares | Suburban | Background | Hourly |
| ES1564A | GC-FID | -3.645 | 40.540 | Alcobendas | Urban | Traffic | Hourly, Daily |
| ES1565A | GC-FID | -3.801 | 40.282 | Fuenlabrada | Urban | Industrial | Hourly, Daily |
| ES1580A | GC-FID | -3.809 | 43.461 | Santander | Urban | Traffic | Hourly, Daily |
| ES1601A | GC-MS | -7.011 | 38.888 | Badajoz | Urban | Background | Hourly, Daily |
| ES1602A | GC-MS | -2.428 | 42.464 | Logrono | Urban | Background | Hourly, Daily |
| ES1610A | GC-MS-FID | 2.656 | 39.570 | Palma | Urban | Traffic | Hourly, Daily |
| ES1615A | GC-MS | -6.360 | 39.473 | Caceres | Urban | Background | Hourly, Daily |
| ES1627A | GC-MS | -1.231 | 37.976 | Alcantarilla | Suburban | Industrial | Hourly, Daily |
| ES1631A | GC-MS | -4.730 | 41.646 | Valladolid | Urban | Traffic | Hourly, Daily |
| ES1633A | GC-MS | -1.145 | 37.994 | Murcia | Suburban | Traffic | Hourly, Daily |
| ES1635A | GC-MS | -0.472 | 38.359 | Alicante | Urban | Traffic | Hourly, Daily |





|  |  |  |  |  |  |  |  |
|---|---|---|---|---|---|---|---|
| ES1802A | GC-FID | -3.468 | 40.909 | Patones | Rural | Background | Hourly, Daily |
| ES1819A | GC-MS | -6.338 | 38.907 | Merida | Urban | Background | Hourly, Daily |
| ES1834A | GC-MS | -0.026 | 39.989 | Castello De La Plana | Urban | Traffic | Hourly, Daily |
| ES1849A | UNK | -0.718 | 38.259 | Elche | Urban | Traffic | Hourly, Daily |
| ES1942A | GC-MS | -3.581 | 40.462 | Barajas De Madrid | Urban | Background | Hourly, Daily |
| ES1986A | GC-MS | -3.689 | 41.666 | Aranda De Duero | Urban | Traffic | Hourly, Daily |
| ES2075A | GC-MS | -5.970 | 43.346 | Castandiello | Suburban | Industrial | Daily |

**F3 Stations measuring xylene**



**Table F3.** Location and characteristics of selected stations measuring xylene in 2019.

| Station reference | Meas. method | Lon | Lat | City | Area classification | Station classification | Time resolution |
|---|---|---|---|---|---|---|---|
| ES0651A | GC-MS | -0.913 | 37.603 | La Union | Suburban | Industrial | Daily |
| ES0879A | GC-MS | -5.899 | 43.550 | Aviles | Suburban | Industrial | Daily |
| ES1161A | GC-MS | -5.587 | 42.604 | Leon | Urban | Traffic | Hourly, Daily |
| ES1239A | GC-MS | -0.376 | 39.456 | Valencia | Urban | Traffic | Daily |
| ES1269A | GC-MS | -5.833 | 43.367 | Oviedo | Urban | Traffic | Daily |
| ES1272A | GC-MS | -5.674 | 43.530 | Gijon | Urban | Traffic | Daily |
| ES1279A | GC-PID | -4.089 | 38.682 | Puertollano | Suburban | Industrial | Hourly, Daily |
| ES1298A | GC-FID | -4.494 | 41.961 | Palencia | Suburban | Industrial | Daily |
| ES1353A | GC-MS | -5.680 | 43.294 | Sama | Urban | Background | Daily |
| ES1356A | GC-MS | -4.741 | 41.613 | Valladolid | Suburban | Industrial | Daily |
| ES1472A | GC-MS | -1.651 | 42.807 | Iturrama | Urban | Background | Daily |
| ES1536A | GC-MS | -3.265 | 40.571 | Azuqueca de Henares | Suburban | Background | Hourly, Daily |
| ES1564A | GC-FID | -3.645 | 40.540 | Alcobendas | Urban | Traffic | Daily |
| ES1565A | GC-FID | -3.801 | 40.282 | Fuenlabrada | Urban | Industrial | Daily |
| ES1580A | GC-FID | -3.809 | 43.461 | Santander | Urban | Traffic | Daily |
| ES1601A | GC-MS | -7.011 | 38.888 | Badajoz | Urban | Background | Hourly, Daily |
| ES1602A | GC-MS | -2.428 | 42.464 | Logrono | Urban | Background | Daily |
| ES1610A | GC-MS-FID | 2.656 | 39.570 | Palma | Urban | Traffic | Hourly, Daily |
| ES1615A | GC-MS | -6.360 | 39.473 | Caceres | Urban | Background | Hourly, Daily |
| ES1627A | GC-MS | -1.231 | 37.976 | Alcantarilla | Suburban | Industrial | Daily |
| ES1631A | GC-MS | -4.730 | 41.646 | Valladolid | Urban | Traffic | Daily |
| ES1633A | GC-MS | -1.145 | 37.994 | Murcia | Suburban | Traffic | Daily |
| ES1635A | GC-MS | -0.472 | 38.359 | Alicante | Urban | Traffic | Hourly, Daily |
| ES1802A | GC-FID | -3.468 | 40.909 | Patones | Rural | Background | Daily |
| ES1803A | GC-FID | -4.014 | 40.634 | Collado-Villalba | Urban | Traffic | Daily |
| ES1819A | GC-MS | -6.338 | 38.908 | Merida | Urban | Background | Hourly, Daily |
| ES1834A | GC-MS | -0.026 | 39.989 | Castello De La Plana | Urban | Traffic | Hourly, Daily |





| ES1849A | GC-FID | -0.717 | 38.259 | Elche | Urban | Traffic | Hourly, Daily |
| ES1986A | GC-MS | -3.689 | 41.666 | Aranda De Duero | Urban | Traffic | Hourly, Daily |
| ES2075A | GC-MS | -5.970 | 43.346 | Castandiello | Suburban | Industrial | Daily |

**Appendix G: Sensitivity Analysis**

**G1 Industrial**



**Figure G1.** Location of the industrial AQ stations and the facilities accounted for the industrial sensitivity analysis, with the respective HERMESv3 code.





## G2 Mopeds and Motorcycles

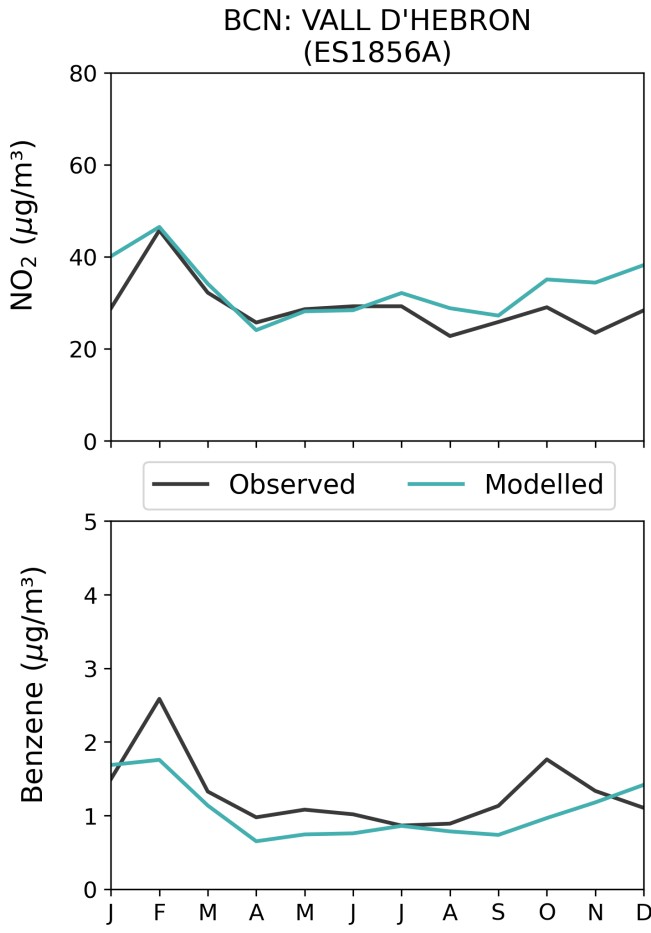

**Figure G2.** Monthly $NO_2$ and benzene concentrations ($\mu$g.m$^{-3}$) at the urban background station ES1856A in Barcelona (BCN).



**Table G1.** Data used to estimate correction factors accounting for different vehicle categories ages. Mcorr is the mileage correction factor for a given mileage, Mmean is the mean fleet mileage of vehicles for which correction is applied, AM is the degradation of the emission performance per kilometre, and BM is the emission level of a fleet of brand new vehicles, which is assumed to be 1.

| COPERT category | AM | BM | Mmean | Mcorr |
|---|---|---|---|---|
| Mopeds 2-stroke <50 cm$^3$_Euro 1 | 0.0000216 | 1 | 8207 | 1.18 |
| Mopeds 2-stroke <50 cm$^3$_Euro 2 | 0.0000216 | 1 | 22537 | 1.49 |
| Mopeds 2-stroke <50 cm$^3$_Euro 3 | 0.0000216 | 1 | 55812 | 2.21 |
| Mopeds 2-stroke <50 cm$^3$_Euro 4 | 0.0000216 | 1 | 3970 | 1.09 |
| Motorcycles 2-stroke >50 cm$^3$_Conventional | 0.0000216 | 1 | 2744 | 1.06 |
| Motorcycles 2-stroke >50 cm$^3$_Euro 2 | 0.0000216 | 1 | 5099 | 1.11 |
| Motorcycles 2-stroke >50 cm$^3$_Euro 3 | 0.0000216 | 1 | 27358 | 1.59 |
| Motorcycles 2-stroke >50 cm$^3$_Euro 4 | 0.0000216 | 1 | 6290 | 1.14 |
| Motorcycles 4-stroke <250 cm$^3$_Conventional | 0.0000216 | 1 | 90500 | 2.96 |
| Motorcycles 4-stroke <250 cm$^3$_Euro 1 | 0.0000216 | 1 | 26780 | 1.58 |
| Motorcycles 4-stroke <250 cm$^3$_Euro 2 | 0.0000216 | 1 | 95064 | 3.06 |
| Motorcycles 4-stroke <250 cm$^3$_Euro 3 | 0.0000216 | 1 | 127280 | 3.75 |
| Motorcycles 4-stroke <250 cm$^3$_Euro 4 | 0.0000216 | 1 | 17283 | 1.37 |
| Motorcycles 4-stroke >750 cm$^3$_Conventional | 0.0000216 | 1 | 21376 | 1.46 |
| Motorcycles 4-stroke >750 cm$^3$_Euro 1 | 0.0000216 | 1 | 15175 | 1.33 |
| Motorcycles 4-stroke >750 cm$^3$_Euro 2 | 0.0000216 | 1 | 24297 | 1.53 |
| Motorcycles 4-stroke >750 cm$^3$_Euro 3 | 0.0000216 | 1 | 43869 | 1.95 |
| Motorcycles 4-stroke >750 cm$^3$_Euro 4 | 0.0000216 | 1 | 7936 | 1.17 |
| Motorcycles 4-stroke 250 - 750 cm$^3$_Conventional | 0.0000216 | 1 | 57704 | 2.25 |
| Motorcycles 4-stroke 250 - 750 cm$^3$_Euro 1 | 0.0000216 | 1 | 22192 | 1.48 |
| Motorcycles 4-stroke 250 - 750 cm$^3$_Euro 2 | 0.0000216 | 1 | 48776 | 2.05 |
| Motorcycles 4-stroke 250 - 750 cm$^3$_Euro 3 | 0.0000216 | 1 | 83049 | 2.8 |
| Motorcycles 4-stroke 250 - 750 cm$^3$_Euro 4 | 0.0000216 | 1 | 14224 | 1.31 |

*Author contributions.* KO: Conceptualization, Investigation, Data curation, Methodology, Writing; MG: Conceptualization,Investigation, Supervision, Data curation, Methodology, Writing; OJ: Funding acquisition,Supervision, Software, Methodology, Writing and reviewing; HP: Methodology, Software, Writing and reviewing; DB: Methodology, Data curation; CT: Software; GMP: Software; FL: Software; CPGP: Supervision, Funding acquisition, Project administration, Reviewing



*Competing interests.* The contact author has declared that none of the authors has any competing interests.

*Acknowledgements.* The research leading to these results has received funding from the Ministerio para la Transición Ecológica y el Reto Demográfico (MITECO) as part of the Plan Nacional del Ozono project (BOE-A-2021-20183, BOE-A-2023-24403); from the Ministerio de Ciencia, Innovación y Universidades (MICINN) as part of the BROWNING project (Grant No. RTI2018-099894-B-I00 funded by MCIN/AEI/10.13039/501100011033 and by EDRF A way of making Europe); from the VITALISE project (Grant No. PID2019-108086RA-I00 funded by MCIN/AEI/10.13039/501100011033); from the Red Temática ACTRIS España (Grant No. CGL2017-90884-REDT); from the AXA Research Fund; and from the European Research Council (Grant No. 773051, FRAGMENT). K. Oliveira acknowledge the support received by the grant PRE2020-092616 funded by MCIN/AEI/10.13039/501100011033 and ESF Investing in your future. BSC also acknowledge the computer resources at MareNostrum and the technical support provided by the Barcelona Supercomputing Center through the RES (AECT-2022-2-0003, AECT-2023-3-0018).



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
