# Peer review of "On the uncertainty of anthropogenic aromatic VOC emissions: evaluation and sensitivity analysis"

_EGUsphere, 2023_

## Author Comment (AC1)

**Answers to the reviewers**
* * *
The authors would like to thank the reviewers for the opportunity to revise and provide insightful comments and constructive feedback on the manuscript. In response, we have meticulously addressed each point raised, as detailed in the attached file. Our responses are provided in red text below each respective comment. Furthermore, to enhance the manuscript's quality, we have incorporated additional minor changes and supplementary information.

**RC1**

Oliveria et al. utilize HERMES, an emissions model that calculates an anthropogenic inventory and outputs gridded output, and MONARCH, a Chemical Transport Model, to simulate ambient benzene, toluene, and xylene concentrations. These modeled concentrations are then compared to ground-level observations in many locations throughout Spain. The modeling system is then perturbed to generate BTX concentrations for a few sector-specific sensitivity tests.

Overall, I think it deserves publication in EGUsphere if my comments below are addressed. In particular, I do not think the authors consider other variables that can lead to model bias beyond incorrect emissions to a large enough degree. There is no discussion related to potential biases in modeled meteorology, which can have a large impact on simulated concentrations of atmospheric constituents. Below, I list all my comments and attempt to provide guidance, where possible.

**Major Comments**:

- 5-7, 8-10, and 11-13 feature a lot of repetition in what is being presented. I feel like both spatial plots in each set can be removed and are largely (and more easily digestible) represented by what is shown in Figs. 7/10/13. Or generate a scatter plot (obs/model) with traffic/industrial/background each represented by a different marker. That would be much easier to follow and extract information from. For example, it isn't clear to me how good the agreement is between model and observations, as alluded to on Line 334.

Answer: The authors agree that a scatter plot could quickly provide information regarding the model's performance. Although, by doing so, we would lose the temporal information, which is crucial in the scope of this work. So, the authors believe that keeping the trends in Figs. 7/10/13 and, accompanied by the table, should provide sufficient information to the reader regarding the model performance. The structure of the results was also mentioned by another reviewer, so to address this, we made the following changes:

- Figures 5, 8, and 11 were merged and moved to the beginning of section 3.2.
- Figures 6, 9 and 12 related to the spatial mean bias were merged and moved to Appendix E since they provided similar information as Fig. 5,8 and 11.
- In several instances, the text was adapted accordingly.

With these changes, we condense the result section, which should improve the clarity and readability of the paper.

- Line 476: I am extremely confused by this comparison. From the proceeding paragraph and based on reading the figure, it looks like 12 facilities from the LPS database and 2 facilities from PRTR are shown. So, where are LPS and PRTS being compared in the figure?

Answer: The authors understand that using the HERMESv3 code in the x-axis could induce confusion. To address this issue, we re-did the figure. Now the facilities are numbered as "F1, F2, …." and to differentiate the source of the data for the baseline/scenario, we added the information, i.e. LPS, PRTR, or if it was estimated for this work, on the top of each bar along the total emissions. The caption was improved: "Comparison, between the baseline and the industrial scenario, of total VOC emissions (t) in 2019 for 14 facilities in Spain. Each bar shows the total emissions and the source of the data (i.e. PRTR or LPS) or if it was estimated ("Est.") for this work. The * indicates that the facility is located near an industrial air quality monitoring station.". By doing so, the authors believe that now the figure shows a clear comparison for the 14 facilities.

**Minor Comments**:

- "Heterogeneously" in line 10 seems awkward. I'm not sure what the author's mean when they use that word.

Answer: We have revised the phrase for better clarity as follows: "Official emissions reported for these facilities were replaced by alternative estimates, resulting in varied improvements in the model's performance across different stations. However, uncertainties associated with industrial emission processes persist, emphasising the need for further refinement.".

- Line 74: "model chemical transport model". Remove the first "model" in the sentence.

Answer: We have removed the redundant "model" from the sentence.

- Line 136: Feed à fed

Answer: We have corrected "Feed" to "Fed" in the text.

- 3: Please include "road" in "road transport" on the figure legend.

Answer: In the revised figure, the legend now includes "road transport". In some instances throughout the manuscript this was also changed.

- Line 362: The model appears low biased in the afternoon, and I would not dismiss an issue with the temporal profiles as quickly as the author's seem to.

Answer: The authors agree with the reviewer that there might still be relevant uncertainties related to the temporal profiles. Nevertheless, we see that for $NO_2$, the model reproduces the morning peak well. This suggests other processes related with meteorology or chemistry could be more responsible for the time shift than the temporal profiles. Analysing the main processes in our CTM, we identified that one of the main issues could be related to the dry deposition. The dry deposition scheme employed in MONARCH is based on the Wesely resistance scheme which adjusts the canopy resistances of individual model species based on their solubility (using the Henry's law constant) from observationally-based $O_3$ resistances. While effective for certain species, this approach may not accurately represent the deposition of VOCs. The Henry Law constant values reported for VOCs are two orders of magnitude higher than for $NO_2$, and a big range is reported in the literature, e.g. for benzene 0.12-0.22, toluene 0.13-0.21. (https://webbook.nist.gov/cgi/cbook.cgi?ID=C108883&Units=SI&Mask=10#Solubility) . While this is out of the scope of this work, we believe it could explain the biases identified and clearly deserves further investigations.

We introduced this information to the manuscript:

"While the shift between modelled and observed morning peaks is also reported for benzene concentrations (see Fig. 6), the model performs well for $NO_2$. This suggests that, while there could be issues with the temporal disaggregation of emissions or the reproduction of key meteorological parameters (e.g., PBL height), the primary concern lies in chemical processes affecting VOCs, such as dry deposition. This should be further investigated as the concentrations were found to be quite sensitive to this poorly constrained process."

- Broadly, when describing the observation stations using city names, I'm largely unsure what part of the spatial figures the authors are discussing (I'm an American with limited geographical knowledge of Spain).

Answer: The authors agree that the current plots may need to provide more context, especially for readers with limited knowledge of the geography of Spain. To address this, we have added the names of the cities in the maps to facilitate better identification of the stations.

- Line 405: The authors attribute bias from "traffic stations" to issues related to cold-start emissions. While possible, near-road stations are not only influenced by mobile emissions. In addition, there are other mobile source

processes beyond cold-start emissions that could affect wintertime model performance. I wouldn't jump to such definitive conclusions.

Answer: The authors agree that the near-road stations are not only influenced by road transport emissions. During winter, the residential sector makes a big contribution when burning biomass. However, in Spain, it is important to note that the main fuel used in urban areas is natural gas. Therefore, the impact of the residential sector on these stations should be limited.

Regarding the emissions from other non-road mobile sources (e.g., construction machinery, lawn and garden equipment), this could impact, so we added to the text as follows: "For traffic stations, the levels observed during spring and summer have a smaller bias than during the winter months. This suggests that traffic emissions related to cold-start emissions, as well as emissions from other non-road mobile sources (e.g., construction machinery, lawn and garden equipment) influencing measured levels in these environments, might be underestimated."

- Line 409: Again, I don't think you can conclude this issue is due to model chemistry and not attributable to (at least partially) temporal allocation of emissions. Also, what about meteorology? Perhaps the meteorological data has a bias in something like PBL, which can influence the diurnal pattern of modeled concentrations.

Answer: The authors agree with the reviewer that, while the scope of this work focuses on emissions uncertainties, it is important to acknowledge other potential sources of uncertainties. In several instances of the manuscript, we mentioned possible uncertainties related to meteorology and their impact on modelled concentrations. Additionally, we added references (Brunner et al., 2015; Sicard et al.; 2021) in section 2.1.2, which assessed the performance of MONARCH in reproducing different meteorological parameters, such as temperature, relative humidity, wind speed, wind direction, radiation, and PBL.

As commented previously, focusing on the shift between modelled and observed morning peak we identified that the model is performing well when evaluating the same stations for $NO_2$ (as shown in Appendix F). While uncertainties in the meteorology and emissions could play a role, the authors believe that the dry deposition may be one of the primary processes affecting these results.

- Line 523: Again, I don't think you can conclude the issue is missing emissions without considering meteorological effects or grid scale of the base (10km x 10km) model. In fact, the modest improvements following huge increases in emissions in the alternative scenario, to me, indicate some of these other factors (e.g., PBL) likely have a stronger influence on modeled concentrations than emission perturbations. In addition, perhaps the speciation of NMVOC is what is incorrect. A speciation profile can have considerable uncertainty.

Answer: The authors agree with the reviewer that the modest improvements from the alternative scenario in some locations could be related to meteorology and therefore the discussion should not focus only on emissions. So, this was included in the discussion:

"In addition, the limited impact observed in certain locations may not solely be due to emission uncertainties. The results might be influenced by meteorology biases in parameters, such as PBL or wind fields. These factors together with the grid resolution of the model may explain the uncertainties identified in the results, especially when considering emissions from industrial point sources."

It is clear that the speciation profiles account for big uncertainties as we identified in a previous work (Oliveira et al. 2023). However, for some specific activities (e.g. coke manufacturing) we compared and tested profiles from different literature sources and saw a limited impact on the results. This point was already mentioned in the manuscript "This could indicate a limitation arising from the speciation profiles used, although the fractions were similar compared to other available profiles.".

**RC2**

Major comments:

Oliveria et al. utilize the CTM model to simulate BTX concentrations and compare them with observational data. However, several critical points are not clearly presented, especially regarding the VOC chemical reactions and the CTM model itself. If the paper clarifies these issues in the discussion, this paper deserves to be published.

BTX react with the OH radical, ozone, and the NO3 radical, with both OH and NO3 being challenging to measure in ambient conditions. Therefore, the model's performance in simulating ozone is crucial to demonstrate that the CTM model functions well based on the emission and meteorological inputs. I suggest that the author should include an evaluation of ozone, like NO2 provided in Appendix E.

Answer: The authors agree with the importance of including the evaluation of Ozone. So this was added in Appendix F.

Furthermore, there is a critical issue in the chemical process within the model simulation part: The CB05 reactions R128 and R138 use model lumped species XYL and TOL, but the author has added A2 and A3 reactions in this study and uses the same species names (XYL and TOL) for explicit xylene and toluene. There is no clear method to distinguish between the lumped and explicit species that share the same species name. As a result, the model results for XYL and TOL represent lumped species and not the explicit species (xylene and toluene). The GC-MS/FID

observed data for xylene and toluene are for explicit species. If so, the comparison does not match like with like. Please address this issue.

Specific comments:

1. Line 103: "The VOC speciation mapping disaggregates total VOCs ……" explains the emission and VOC speciation mapping for the CTM model. In this study, are the BTX emission data for explicit Toluene and Xylene, or are they for the model's lumped species TOL and XYL? The lumped species TOL and XYL are used to represent similar species like Ethylbenzene, Styrene, Indene, etc. The lumped XYL can reflects more than 80 species, and TOL can represents more than 50 species.

Answer: The authors acknowledge this critical comment by the reviewer, as this was previously overlooked by the authors. This is, in fact, a limitation in this work, as toluene and xylene are lumped species in CB05. This would still be the case even if we updated the chemical mechanism of MONARCH to CB7.

One solution would be to employ the 'tracer' method to add explicit species in the model as mentioned by Ge et al. (2024). While this option presents a greater implementation challenge, it also acknowledges uncertainties. Tracer methods in air quality models may face limitations in accurately representing complex atmospheric processes and chemical reactions due to model dependencies and system perturbations.

It is important to note that while TOL and XYL lump multiple species, in Spain the major contributions come from explicit toluene and xylenes, respectively. For TOL total mass, toluene contribution is 63% on average, followed by styrene (around 20% on average over the domain), while for XYL total mass, (m,o,p-) xylenes contribution is 41% on average, followed by 1,2,4-trimethylbenzene (11%). These contributions were quantified using the HERMESv3 model and the Spanish speciation information developed in a previous work by Oliveira et al. (2023).

Since the TOL and XYL species considered in MONARCH are lumped and not explicit, the comparison between modelled and observed concentrations presented in the previous version of the manuscript was not a direct comparison. To overcome this limitation, we used HERMESv3 and the speciation information reported by Oliveira et al. (2023) to estimate gridded and monthly emission ratios between explicit and CB05 lumped species (i.e., toluene/TOL, xylenes/XYL).

[Figure]

The gridded information was used to derive station type and monthly-dependent emission ratios, as presented in the following figure, which has been added in the Appendix:

[Figure]

We used the constructed ratios to adjust the TOL and XYL concentrations modelled by MONARCH, assuming that the ratios between explicit and lumped species from emissions can be linearly extrapolated to concentration values. This allowed us to analyse the results as explicit toluene and explicit xylene. This approach was clarified in the manuscript as follows: "To match explicit toluene and xylene observations, we assumed that ratios between explicit and lumped species from emissions could be extrapolated to concentrations. Spatially (gridded and per type of station) and temporally (monthly) resolved ratios (i,.e., toluene/TOL and xylene/XYL) were estimated using HERMESv3 and the speciation information reported by

Oliveira et al. (2023) and applied to the model outputs (see Appendix G). On average in Spain, for TOL total mass, toluene contribution is 63 % while for XYL total mass, (m,o,p-) xylene contribution is 41 %.". Additionally, all the figures were updated and the text in the manuscript was adapted.

For this work, we applied ratios, but another option, as previously mentioned, would have been to add explicit species in the mechanism, as done by Ge et al. (2024). This implementation will be explored in future developments. This was also added in the new section "4.2 Implications for future research" (previously called "Conclusions"): "The modelling simplifications to represent VOCs in chemical mechanisms are essential for computational efficiency. While necessary, they introduce uncertainties, with the main limitation for model evaluation being the restricted number of species with direct comparison to measurements. Future works using MONARCH should focus on including explicit species in the model, following a similar methodology as implemented by Ge et al. (2024). This approach is key to extend the current work and ensure a direct comparison between modelled and observed species..".

2.  Line 122: "Notably, the mechanism also accounts for explicit species, namely...", In the CB05 document (https://www.camx.com/Files/CB05_Final_Report_120805.pdf) , the XYL and TOL are "xylene and other polyalkyl aromatics" and "Toluene and other monoalkyl aromatics". They are lumped species.

Answer: The authors apologise for the mistake, this was fixed in the revised version as follows: "VOCs are lumped into several groups, such as: propionaldehyde and higher aldehydes (ALDX), acetaldehyde (ALD2), ethene (ETH), formaldehyde (FORM), internal olefin (IOLE), terminal olefin carbon bond (OLE), paraffin carbon bond (PAR), terpene (TERP), toluene and other monoalkyl aromatics (TOL), and xylene and other polyalkyl aromatics (XYL)."

3.  In Appendix A, what is the reference for the A1 reaction rate constant (k) (2.47E-12 e^(-206/T)) and the products (OH + 0.764*BENZRO2)? The rate constant (k) for Benzene in gas-phase chemistry mechanisms, CB6r2 and MCM3.3, is both 2.3E-12 e^(-190/T) (source: https://mcm.york.ac.uk/MCM/species/BENZENE). The inclusion of the OH radical as a product (OH + 0.764*BENZRO2) in the appears to be incorrect. If this additional OH radical product in A1 is considered in the CTM model, it will impact the OH concentration, VOCs concentration and reactions.

Answer: The authors would like to clarify that, as mentioned in section 2.1.2, the model uses the CB05 mechanism but with extensions from Sarwar et al., (2012), including benzene. This implementation follows a similar approach as CMAQ and therefore the rate and products differ from the values pointed out by the reviewer. The rates presented for benzene are from the SAPRC-99

(https://intra.engr.ucr.edu/~carter/pubs/s99doc.pdf) mechanism which provides a value of 2.47E-12. Moreover, it is important to mention that this extension doesn't impact CB05 as the reaction doesn't consume OH. Despite this we compared the rate expressions used in MONARCH and the one mentioned by the reviewer and the differences may have a limited impact, as shown below:

[Figure]

To make it clearer for the reader, we decided to replace the equations presented in Appendix A by a table with references for each rate. The table is as follow:

| Reactants | Products | Rate expression | Reference |
|---|---|---|---|
| BENZENE + OH | OH + 0.764*BENZRO2 | $2.47 \times 10^{-12} \times e^{-206/T}$ | Atkinson et al. (1989) |
| TOL + OH | 0.440HO$_2$ + 0.080XO$_2$ + 0.360CRES + 0.560TO$_2$ + 0.765TOLRO2 | $1.8 \times 10^{-12} \times e^{-355/T}$ | Le Bras (1997); Gery et al. (1989); Sarwar et al. (2012) |
| XYL + OH | 0.700HO$_2$ + 0.500XO$_2$ + 0.200CRES + 0.800MGLY + 1.100PAR + 0.300TO$_2$ + 0.804XYLRO2 | $1.7 \times 10^{-11} \times e^{-116/T}$ | Gery et al. (1989); Sarwar et al. (2012) |

**Table A2.** Reactions in the CB05 extended mechanism used in MONARCH for BTX. Note: References are provided for the different reactions and rates.

4.  Figure5: symbol for Max Observed Value (blue star) not in the figure.

Answer: In Figure 5 the authors agree that the symbol was challenging to see. So, we increased the size and colour of the marker in Figures 5, 8, and 11 and now the symbol for the max observed value is clearly visible.

**RC3**

Oliveria et al. employ HERMES and MONARCH to simulate ambient concentrations of BTX. The authors conducted a rigorous evaluation of BTX emissions from different sectors, temporal variation including weekday-weekend differences to an extent. The simulated concentrations are subsequently compared to ground-level observations in Spain. Further, the biases between observed and modeled data were explored through sensitivity tests. I appreciate the authors highlighting the need for extensive BTX measurements and regulations. This work is justifiable for publication in ACP. However, there are some shortcomings of the study that need to be addressed:

1. The title needs to highlight "model-evaluation"

Answer: We appreciate your feedback and have updated the title to emphasise "model-evaluation" as follows: "On the uncertainty of anthropogenic aromatic VOC emissions: model evaluation and sensitivity analysis"

2. Although sensitivity tests were performed for some source aspects that are driving the uncertainty, the authors need to acknowledge that ambient VOC levels are not only driven by emission sources but also by the atmospheric chemical mechanisms, meteorology, and transported pollutants from upwind regions. Please clarify uncertainties that these aspects may lead to.

Answer: The authors agree with the reviewer that the current version of the manuscript mainly pointed to the uncertainties of emissions without acknowledging the several uncertainties present in the whole modelling chain that could influence the results. So, we added this information in the revised manuscript when discussing the impact sensitivity from the industrial scenario:

"In addition, the limited impact observed in certain locations may not solely be due to emission uncertainties. The results might be influenced by meteorology biases in parameters, such as PBL or wind fields. These factors together with the grid resolution of the model may explain the uncertainties identified in the results, especially when considering emissions from industrial point sources."

To avoid providing extensive details on the evaluation of the meteorology, as it is out of the scope of this work, we provided in section 2.1.2 further references where MONARCH's meteorology has been evaluated.

Moreover, we also added the information on the uncertainty of the observations in the sensitivity analysis of the mopeds and motorcycles:

"Additionally, despite the increases in emissions, the model's low performance could also suggest other issues, such as reproducing localised atmospheric and meteorological characteristics (e.g. PBL). Additionally, as seen in Appendix H, the measurement methods vary. In Barcelona, the stations use Gas Chromatography-Mass Spectrometry-Flame Ionisation Detection (GC-MS-FID),

while in Madrid, the stations use GC-MS. The differences in measurement methods could lead to several uncertainties and discrepancies in the observed values between both cities (e.g., interferences). Similarly, variations in measurement methods have also been identified as a factor influencing $NO_2$ values (Steinbacher et al., 2007; Villena et al., 2012). While the experimental limitations of methodologies and instrumentation measuring VOCs are beyond the scope of this paper, they represent crucial variables to consider when performing model evaluations."

3. The actual ambient measurements of NMVOCs itself have uncertainty depending on the measurement method. This uncertainty is exacerbated when estimates or calculations are made without direct measurements. In Section 3.3.1, authors should address the biases inherent in reported NMVOC emissions derived from PRTR estimation/calculations. It is imperative to elucidate how this impacts the findings of the study. Additionally, the paper lacks an examination of previous research that delves into these aspects.

Answer: The authors agree that the VOC measurements account for several uncertainties depending on the measurement method. Upon analysing differences between urban traffic stations (section 3.3.2) in Madrid and Barcelona, it was found that benzene levels in Barcelona were five times higher than those in Madrid. Part of the difference could be attributed to differences in measurement methods, as the Madrid stations use GC-MS and Barcelona GC-MS-FID. This analysis was missing in the previous version of the manuscript and is further addressed in section 3.3.2, as well as a bullet point in the "Summary and Implications for future research" section (previously the conclusions section).

Regarding the uncertainties in the estimations/calculations of PRTR for the refineries, the authors agree with the reviewer that this should be mentioned in the manuscript. We compared the differences between PRTR and LPS for $NO_2$ (for which the emissions are mainly obtained through measurements) and VOCs (obtained mainly through estimations/calculations. This showed that the differences are much higher for VOCs than for $NO_2$. So, we added this information to the manuscript:

"It is essential to acknowledge that estimating or calculating emissions without direct comparisons to measured emission fluxes can introduce significant uncertainties and should be approached with caution. For example, in PRTR, for refineries, $NO_2$ values are mainly obtained through direct measurements, with differences from LPS across different facilities averaging below 36 %. However, for VOCs, the differences are substantial, with the smallest difference for a specific facility being 50 %."

4. The paper's structure resembles more of a "report" than a "manuscript," making it challenging to navigate the results. Certain sections could be consolidated to improve clarity. For instance, the technical comments provided below could serve as an illustration.

Answer: The structure of the results was also mentioned by another reviewer, so to address this, we made the following changes:

- Figures 5, 8, and 11 were merged and moved to the beginning of section 3.2.
- Figures 6, 9 and 12 related to the spatial mean bias were merged and moved to Appendix E since they provided similar information as Fig. 5,8 and 11.
- In several instances, the text was adapted accordingly.

With these changes, we condense the result section, which should improve the clarity and readability of the paper.

Technical comments:

1. To enhance reader comprehension and facilitate result comparison, consider consolidating standalone figures and their respective discussions. For instance, merge figures 5, 8, and 11; figures 7, 10, and 13; figures 6, 9, and 12, along with Tables 2, 3, and 4.

Answer: The complete answer is presented in the previous comment.

2. Line 456: "To assess this, …..." this information e.g., the use of LPS and PRTR in this paper has to be clarified in the method section.

Answer: The information regarding the source of the industrial emissions, i.e. LPS and PRTR was moved and clarified in section 2.1.1 as follows: "The industrial emissions at plant-level used in HERMESv3 are derived from the national Large Point Sources (LPS) database and from the Spanish Pollutant Release and Transfer Register (PRTR-Spain). Both databases are compiled and maintained by the Spanish Ministry for the Ecological Transition and the Demographic Challenge (Ministerio para la Transición Ecológica y el Reto Demográfico, MITERD), which estimates the emissions using the information provided by the corresponding industrial facilities (MITERD, 2023, 2022). A priority is given to LPS when both datasets provide values for the same facility, since the data reported by LPS is consistent with the official inventories.".

3. While the conclusion section exceeds traditional length norms, it effectively summarizes all aspects of the paper. I would suggest adding a separate section after the conclusion to specifically address the atmospheric implications of this research.

Answer: The authors changed and split the previous "Conclusions" section into "Summary and Implications for future research". We believe that this helps with the clarity of the section and improves the readability.

Editorial comments:

1. Abstract:
    1. Abbreviate HERMESv3 and MONARCH

Answer: Added.

    2. L9-11: reads odd, please rephrase for better readability.

Answer: We have revised to improve clarity, as follows: "Official emissions reported for these facilities were replaced by alternative estimates, resulting in varied improvements in the model's performance across different stations. However, uncertainties associated with industrial emission processes persist, emphasising the need for further refinement."

2. Introduction:
    1. L22: biomass burning can be from natural activity as well.

Answer: The authors agree with the reviewer, so to be clearer we replaced by residential biomass burning.

    2. L24: rewrite for better readability.

Answer: We have revised to improve clarity, as follows: "Meanwhile, biogenic VOCs (BVOCs) also play a crucial role in atmospheric chemistry. Globally, BVOCs represent a larger fraction of total VOCs and exhibit higher chemical reactivity compared to many anthropogenic VOCs (Guenther et al., 2006)."

    3. L26: clarify what you mean by human-induced atmospheric changes.

Answer: We have revised L24 to improve clarity, as follows: "Additionally, it is important to note that human-induced atmospheric changes through land use management increase oxidant levels which can also boost natural aerosol production like biogenic SOA  (Kanakidou et al., 2000)."

    4. L33: confusing as two different statistics for benzene are presented. Rewrite for clarity.

Answer: There was a typo and now the revised sentence is: "Also, Oliveira et al. (2023) showed that in Spain, toluene and xylene are in the top 5 main species contributing to OFP, while benzene, although having a low reactivity, is in the top 20 species contributing to OFP."

    5. L37: clarify if Huang et a. (2014) is a global estimate or site-specific.

Answer: The sentence was revised as follows: "Huang et al. (2014) found that SOA contributes about 30 to 77 % of PM2.5 mass concentrations in their study, which focused on severe haze pollution events in specific urban areas."

6. L45: The change in paragraph reads odd. Mention why EI is important, and give reasons and references.

Answer: This was rewritten as: "Within the air quality modelling chain, the emission inventories, while being a significant source of uncertainty, serve as key elements to understand air pollution origins, in forecasting applications, and to design effective emission abatement strategies (Russell and Dennis, 2000; Day et al., 2019)."

7. L51: mention that MITERD (2023) is EU-based and may be different for other regions of the world.

Answer: The authors revised the sentence and added "in Spain" to clarify that the example is specific to Spain.

8. L56: Clarify what you mean by overlapping uncertainty sources.

Answer: The authors revised the sentence as follows: "When evaluated against observed pollutant concentrations, they typically provide key insights on the validity of emission inventories, even with overlapping uncertainties arising from various sources across the entire evaluation chain. These uncertainties include those related to emissions, the capability of the CTM to accurately reproduce chemical and meteorological conditions, as well as a wide range of uncertainties in observational data (i.e. the measurement technique, instrumentation)."

9. L59: remove "in the literature"

Answer: Removed.

10. L60-63: rewrite the sentence for better readability.

Answer: The sentence was revised as follows: "first, the models use simplified chemical mechanisms. These mechanisms group the numerous individual VOC species into broader families based on known reactions or the number of carbon bonds they possess (e.g., the Carbon Bond 2005 chemical mechanism (CB05; Yarwood et al., 2005)). This grouping enables the models to accurately replicate O3-NOx chemistry with both acceptable precision and computational efficiency."

3. Data and Method:
    1. Clarify why was the year 2019 chosen in this paper.

Answer: This was clarified in section 2.1.2 as follows: "We chose to focus on the year 2019, as the emissions and observational data for 2022 had not been validated

at the time of our analysis, and using emission data from 2021 or 2020 would be significantly affected by the COVID-19 pandemic (Guevara et al., 2023)."

    2.  L110: clarify what you mean by state-of-the-art works.

Answer: To improve the readability the term "state-of-the-art" was replaced as follows: "...which performed a collection, review and comparison of profiles available from recent studies."

    3.  L126: ("i.e. cloud,….") rewrite for clarity.

Answer: The sentence was revised to improve clarity as follows: "The CB05 is well formulated for various tropospheric conditions, from urban to remote areas. It employs the Fast-J scheme to calculate photolysis rates (Wild et al., 2000), which considers the influence of clouds, aerosols and absorbers such as ozone."

    4.  L134: remove "for"

Answer: Removed.

    5.  L142: add citation/web link associated with Spanish ministry data.

Answer: Added.

    6.  Table 1: In the caption, add in Spain.

Answer: Added.

    7.  L 157-165: redundant information. Remove redundant lines that are evident in Table 1. Some additional info in this para can be added to the Table 1 caption.

Answer: We removed the redundant lines to improve readability, and moved some of the information to the caption of Table 1. The caption now shows: "Available number of stations in Spain by area classification and station type measuring benzene, toluene, and xylene in 2019 after applying a temporal coverage of 75 % and quality assurance criteria. The number of stations that only measure with daily resolution is shown in parentheses."

4.  Results:
    1.  L209: mention "figure not included" for clarity.

Answer: Added.

    2.  L195: use the Arabic numbering system to separate different groups for better readability.

Answer: Added.

3. L206: replace weekend effect with "weekday-weekend effect"

Answer: Replaced.

4. L363: clarify what these chemical processes affecting VOCs are.

Answer: We have added information based on other reviewers comments and revised L363 to improve clarity, as follows: "While the shift between modelled and observed morning peaks is also reported for benzene concentrations (see Fig. 6), the model performs well for $NO_2$. This suggests that, while there could be issues with the temporal disaggregation of emissions or the reproduction of key meteorological parameters (e.g., PBL height), the primary concern lies in chemical processes affecting VOCs, such as dry deposition. This should be further investigated as the concentrations were found to be quite sensitive to this parameter and there is a wide range reported in the literature.".

5. Conclusion:
   1. No need to abbreviate VOC or SOA.

Answer: This was revised.

**RC4**

The study by Oliveria et al. focuses on the uncertainties associated with the representation of VOCs in atmospheric emission inventories. They tested the spatiotemporal distribution of benzene, toluene, and xylene (BTX) as predicted by an atmospheric model (MONARCH), which utilized emissions produced by the High-Elective Resolution Modelling Emission System (HERMESv3) as input. The spatiotemporal distribution of BTX was compared with ground observations in Spain. The analysis considered different station classifications according to the measurement stations location, with respect to nearby emission sources (e.g., traffic, industrial, rural traffic, rural background etc.). The manuscript addresses an important, yet understudied topic, while the approach used is generally suitable.

However, although the methods apply an atmospheric chemistry model (MONARCH), the authors did not consider the effects of atmospheric dynamics and chemistry on the biases between simulations and observations. In several cases, the explanations given for such discrepancies are speculative, without adequately exploring causal effect. More information and/or analysis should be provided to support the suitability of MONARCH to adequately represent the spatial distribution of benzene, toluene, and xylene by ensuring the appropriate representation of their oxidation rates and their explicit inclusion in the model.

Answer: The authors agree with the reviewer that, while the scope of this work focuses on emissions uncertainties, it is important to acknowledge other potential sources of uncertainties. Therefore, in several instances of the manuscript, we mentioned possible uncertainties related to meteorology and their impact on modelled concentrations. Additionally, to provide more information to the reader on the model's performance when reproducing atmospheric patterns we added references (Brunner et al., 2015; Sicard et al.; 2021) in section 2.1.2. These works assessed the performance of MONARCH in reproducing different meteorological parameters, such as temperature, relative humidity, wind speed, wind direction, radiation, and PBL.

The authors agree that the information regarding explicit benzene, toluene, and xylene was overlooked in the original version of the manuscript. As this was also pointed out by reviewer 2, the full detailed answer can be found there. While the explicit representation would be ideal, the current intermediate mechanisms still lumps TOL and XYL. So, while this approach also accounts for limitations, we proposed an approach based on emission ratios. In the future, an implementation following the tracer methods by Ge et al. (2024) will be tested. This information was added to "4.2 Implications for future research".

Specific comments:

Line 5 – "HERMESv3" – Brief information should be provided so that the reader knows that this is an emission model.

Answer: We added the model full name "High-Elective Resolution Modelling Emission System (HERMESv3)" so it's clear that it is an emission model.

Line 6-" MONARCH" – provide brief information to inform the reader about the type of model it is.

Answer: We added the information that it is an online chemical transport model.

Lines 26-27 - The sentence should be clearer regarding the contribution of VOCs to SOA via those oxidants.

Answer: In response to a similar concern raised by another reviewer, we have further clarified the sentence as follows: "Additionally, it is important to note that human-induced atmospheric changes through land use management increase oxidant levels which can also boost natural aerosol production like biogenic SOA (Kanakidou et al., 2000)"

Line 43 – "They are continuously measured" - The meaning here is not clear to me - when or where are they measured?

Answer: To make it clearer for the reader, the authors rephrased as follows: "This study focuses on BTX because (1) they are continuously measured over time in various locations, providing an important temporal and spatial coverage,..."

Line 48 – "highest uncertainty" - In spatial/temporal distribution? concentrations? Sector wise?

Answer: The authors revised the sentence as follows: "VOCs are typically associated with the highest emission uncertainty."

Lines 58-70 – Not clear to me which information given here specifically refers to the UK.

Answer: To improve the readability, as also pointed out by another reviewer, we removed "in the literature" from the sentence.

Lines 65-66 – "Third, the availability and quality of observational data for VOCs are often limited in scope" - Can you briefly state in what ways? Currently, it is not clear to me how it differs from the second point you raise.

Answer: The authors would like to clarify that the second point concerns emission inventories and how they report VOCs. In contrast, the third point specifically addresses measurement data.

In summary, the three main reasons outlined for the limited amount of works are: 1) CTM models utilising chemical mechanisms, 2) emissions reported as total VOC, requiring speciation profiles to split into individual species, resulting in various uncertainties, and 3) observations are constrained to a limited number of species. In many cases, available information is derived from short campaigns lasting a month or less, making it challenging to compare against model outputs.

Lines 80-85: The manuscript could be made more concise by excluding this section.

Answer: The paragraph describing the paper's structure was removed.

Line 96 – Can you specify the spatial resolution applied?

Answer: The authors added the following sentence: "Specifically for this work, the model resolution was set to feed MONARCH at a spatial resolution of 0.1° by 0.1°."

Line 113- "sectional-bulk" - not clear to me what is meant by this.

Answer: In this context, "hybrid sectional-bulk" refers to the aerosol representation adopted in the model to describe the size distribution of the aerosol components, which combines both sectional and bulk approaches. "Sectional" refers to categorising aerosol particles into size bins based on their diameter. The "Bulk" approach simplifies and treats the aerosols as a homogenous mass without explicit

consideration of their size distribution. This hybrid approach allows for a balance of computational efficiency and accuracy.

In MONARCH, the aerosol module describes the lifetime of dust, sea salt, black carbon, organic matter (primary and secondary), sulphate and nitrate aerosols. The sectional approach is employed to model dust and sea salt aerosols, and other aerosol species are represented using a bulk description. These details are not included in the manuscript as they are not directly relevant to its scope. However, the cited work of Klose et al., 2021a and references therein provides detailed information on the design of the model.

Line 135 – "CAMS" – define at first appearance.

Answer: Revised.

Line 249 – "Urban and suburban industrial stations were also aggregated" – what is the rational for this aggregation?

Answer: The authors changed the sentence to include the rationale as follows: "Urban, suburban and rural industrial stations, when available, were also aggregated due to their similar observed range values and trends. This consolidation was relevant as there is only 1 urban station and 1 rural industrial station."

Line 277 – "from -0.21 to 0.92 µg.m-3" – Can you specify the bias in percentage?

Answer: The NMB values were added.

Lines 278 – 279 – "This is mainly attributed to underestimations of VOCs" – VOCs emission?

Answer: Revised as "This is mainly attributed to underestimations of VOC emissions,..."

Line 281 – "NO2" should be written with subscripted "2".

Answer: Revised.

Lines 312-313 – "Notably, underestimations are more pronounced during winter, suggesting a potential underestimation of road traffic cold start emissions" - For traffic? I don't see that this winter trend is significant when looking at Fig. 7 and Table 2

Answer: The authors agree that when evaluating the average of all the traffic stations, as presented in Fig.7 and Table 2, this might not be so clear to the reader. Despite this, in Table 2 we can see that the lowest MB values are in summer and the biggest in winter. This effect is more evident when looking at specific stations, e.g. stations located in Barcelona and Valencia, as presented in Figure 16.

Figure 7 - Specify H, M and DOW in the figure caption.

Answer: This was specified in the caption for the 3 pollutants as "... hourly ("H"), weekly (day of the week, "DoW"), and monthly ("M") cycles (µg.m$^3$)..."

Figure 9 - What is the difference between "Hourly, Daily" and "Daily"- what is the meaning of "Hourly, Daily"?

Answer: The authors agree that it was redundant since if hourly resolution is available so are daily values. So, to be more explicit, we replaced "Hourly, Daily" with "Hourly".

Line 363 – "chemical processes affecting VOCs" - This is not clear to me. Can you specify what kind of chemical reaction could lead to an earlier VOCs morning build-up compared to the measurements? Do you imply that benzene and toluene are formed by chemical reactions which occur in the morning? Could meteorological effects/stratification of the atmosphere could play a role here too?

Answer: While uncertainties in the meteorology and emissions could play a role, the authors believe that one of the main processes affecting these results is the dry deposition. This is mainly because, for the same stations, for NO$_2$ the model is performing well.

In the past, we identified that the concentrations are quite sensitive to changes in the dry deposition scheme used in the model. A resistance approach following Weseley (1989) is used here, where canopy resistances of individual species are scaled from observational-constrained O$_3$ resistances using their solubility properties (i.e. Henry's Law constant). We believe that dry deposition could play a major role to explain the discrepancies identified. This, while it is out of the scope of this work, should be further investigated as there is a wide range of values in the literature, e.g. for benzene 0.12-0.22, toluene 0.13-0.21 (https://webbook.nist.gov/cgi/cbook.cgi?ID=C108883&Units=SI&Mask=10#Solubility) Henry's Law constant.

This was added to the manuscript: "While the shift between modelled and observed morning peaks is also reported for benzene concentrations (see Fig. 6), the model performs well for NO$_2$. This suggests that, while there could be issues with the temporal disaggregation of emissions or the reproduction of key meteorological parameters (e.g., PBL height), the primary concern lies in chemical processes affecting VOCs, such as dry deposition. This should be further investigated as the concentrations were found to be quite sensitive to this parameter and there is a wide range reported in the literature."

Line 363 – "What do you mean by "temporal profiles"? Can you specify which physical or chemical processes could lead to the trend you refer to?

Answer: This was changed in the manuscript as mentioned in the previous comment.

Lines 405-407 - This is not clear to me - Can you explain why you believe that the large underestimation of traffic emissions in winter is related to cold-start emissions?

Answer: When comparing the different station classifications and focusing on the winter period, we identified major discrepancies, mainly within traffic stations. This suggests our model is able to reproduce emissions from other sectors. This observation is consistent with what we observed for benzene, particularly in Barcelona. In this case, we have 3 stations measuring benzene, i.e. 2 traffic and 1 background. Notably, the model performs well at the background station (presented in Figure G2), indicating accurate emission estimations for that area. However, discrepancies persist at the traffic stations, leading us to attribute them to processes within the road transport sector. As our model lacks adequate representation of VOC cold-start emissions and since cold-start emissions substantially impact emission factors, we conclude that the underestimations observed during winter are likely due to cold-start emissions.

Line 409 – "Indicating an issue related to VOC chemistry in MONARCH' – Can you explain why you necessarily attribute this issue to VOC chemistry? The same comment is relevant for toluene and benzene.

Answer: The authors gave a detailed explanation regarding this point in the previous comment regarding L363. To specify what we believe to be the main issue, we added in line 409 the reference to the dry deposition as follows: "...indicating a possible issue related to VOC processes in MONARCH (i.e. dry deposition)."

Lines 553 – multiple occurrences of "the'

Answer: Removed.

Line 601 – "the impact on air quality results" - What do you mean by this?

Answer: The authors changed the "the impact on air quality results" to ""the impact on benzene concentrations" to be clearer.

Lines 607-611 – There are several other reasons for differences in addition to the apparent not well enough representation of emissions from mopeds and motorcycles. Can you provide concrete evidence that these factors impose a dominant impact on differences between modeling and observations?

Answer: The authors agree that other factors are causing the differences between modelling outputs and observations. Another reviewer also highlighted this. As discussed in the manuscript, observed benzene levels in Barcelona were five times higher than those in Madrid. Further analysis revealed differences in the measurement method between both cities; in Madrid, the stations use GC-MS and

Barcelona GC-MS-FID. While the authors lack expertise in the experimental field, consultations with experts validated this hypothesis that the measurement method could lead to several uncertainties and discrepancies in the observed values.

While the experimental limitations of methodologies and instrumentation measuring VOCs are out of the scope of this paper, it is a crucial variable to keep in mind when performing model' evaluations. So, to address this critical aspect, we have incorporated additional information in Section 3.3.2, highlighting the differences in measurement techniques and their potential implications as follows:

"Additionally, despite the increases in emissions, the model's low performance could also suggest other issues, such as reproducing localised atmospheric and meteorological characteristics (e.g. PBL). Additionally, as seen in Appendix H, the measurement methods vary. In Barcelona, the stations use Gas Chromatography-Mass Spectrometry-Flame Ionisation Detection (GC-MS-FID), while in Madrid, the stations use GC-MS. The differences in measurement methods could lead to several uncertainties and discrepancies in the observed values between both cities. Similarly, variations in measurement methods have also been identified as a factor influencing $NO_2$ values (Steinbacher et al., 2007; Villena et al., 2012). While the experimental limitations of methodologies and instrumentation measuring VOCs are beyond the scope of this paper, they represent crucial variables to consider when performing model evaluations."

The information regarding uncertainties in the observations was also added to the "Summary and Implications for future research" section (section previously named "Conclusions"), as follows:

"Observed benzene levels in Barcelona's urban traffic areas were five times larger than the ones observed in Madrid. While the model performs well in replicating background station values in Barcelona, suggesting a potential underestimation of emissions from mopeds and motorcycles, caution is warranted when comparing model outputs against measurements obtained using different techniques (e.g. GC-MS in Madrid and GC-MS-FID in Barcelona) due to uncertainties and limitations for the different measurement methodologies."

Lines 612-613 – "suggesting that some sources are either not accurately represented in our model or are unaccounted for" - What about atmospheric chemistry effects and/or meteorological effects?

Answer: The authors agree with the reviewer that besides the emissions, there are several other sources of uncertainty affecting the results, e.g., the methods used in the measurements and the model performance reproducing the meteorological and chemical parameters. Therefore, this was revised in the manuscript and a paragraph was added as mentioned in the previous comment.

Lines 613-614 –please rephrase.

Answer: The authors revised the phrase as follows: "So, to improve the model's performance in these locations, it would be important to include city-specific characteristics that impact EF (e.g. topography)."

Table 7 – Specify more clearly what do you mean by "Scenario".

Answer: This was replaced by "Road Traffic Scenario" to improve clarity.

Line 622 – "evaluation studies" - Which evaluation studies?

Answer: To specify that we are referring to works on VOC we revised the sentence as follows: "Nevertheless, comprehensive model-based evaluation studies on VOC species remain scarce in the literature."

Conclusions section –This section should be termed differently, possibly Summary and Conclusions".  For instance, lines 619-629 do not contain conclusions.

Answer: The authors agree with the reviewer, so the previously called "Conclusions" section was restructured and called "Summary and Implications for future research".

Line 631 – "Annual emissions of benzene, toluene and xylene over Spain reach 11 kt, 36 kt, and 25 kt, respectively" - For what years? Is this maximum or average values?

Answer: This was revised as follows: "Total annual emissions in 2019 ….".

Lines 652-655 – Please rephrase for clarity.

Answer: The point was revised as follows: "A sensitivity analysis was tested for the identified industrial activities by estimating alternative emissions considering different sources (i.e. from the PRTR and LPS databases, or estimated for this work). The industrial sensitivity analysis overall showed some improvements, although the model is still strongly underestimating near some industrial sites. This underestimation may be linked to underestimations in the total emissions. Additionally, in some cases, the EF may be more generic, lacking industry-specific nuances and thereby introducing uncertainty in the estimation of the emissions."

Lines 661-663 – This does not seem like a conclusion to me.

Answer: The point, previously in Lines 661-663, is now part of the summary subsection and was completed with relevant information regarding measurement uncertainties, as follows: "Observed benzene levels in Barcelona's urban traffic areas were five times larger than the ones observed in Madrid. As for the background station in Barcelona, the model is performing well; this indicates a potential underestimation of mopeds and motorcycle emissions. However, since the

measured values are obtained using GC-MS in Madrid and GC-FID in Barcelona, there could be uncertainties due to differences in measurement techniques. Therefore comparing model outputs against different measurement techniques should be done with caution."

Lines 687-688 – It is speculative while included in Conclusion section.

Answer: This point was revised to include all the identified uncertainties along the manuscript: "MONARCH performs better in reproducing $NO_2$ than VOCs in the stations considered. This could potentially be related to several factors, such as: the fact that NOx industrial emissions are more robust due to better measured-based EF, chemical mechanisms are designed to reproduce $O_3$ and $NO_2$ while major challenges remain to describe VOCs, limited constraints in key processes like VOCs dry deposition, and scarce availability of VOCs measurements which account for several uncertainties."

Line 703- "improve the performance of air quality models" - Can you elaborate on in what ways this improvement can be achieved?

Answer: We reviewed the sentence as: "To conclude, this research enhances our understanding of VOCs emissions and uncertainties while also contributing to the improvement of air quality models. By identifying limitations in emission sectors, chemical mechanisms, and observations, we pinpoint critical elements for enhancement. As demonstrated by Travis et al. (2024), improving VOC emissions and chemistry can aid in better describing urban ozone pollution, consequently supporting the design of more effective pollution control strategies."

Line 707 – "focusing on assessing the impact on PM2.5" - If you add this here, explain briefly why assessing the impact of PM2.5 is important and better connect it to the context of your study.

Answer: To better connect with VOCs and explaining the importance of PM2.5, the authors added the following sentence: "Given the significant health and environmental implications associated with fine particulate matter and the crucial role of VOCs in its formation through secondary aerosol formation, expanding our analysis will provide a better understanding of the effectiveness of these strategies in improving overall air quality."

---

## Referee Report (RR1)

In general, the revised version demonstrates better structure and provides a clearer explanation for biases between modeling and observations. I recommend applying the 'tracer' method, to explicitly incorporate species in the MONARCH model. This would help address uncertainties that remain unresolved in the revised manuscript. For instance, it may shed light on the speculated effect of deposition velocity on biases between modeling and observation. While the authors have adequately addressed most of my comments, I provide in the following additional feedback on some of my original comments that require further attention.

Lines 26-27 - The sentence should be clearer regarding the contribution of VOCs to
SOA via those oxidants.
Answer: In response to a similar concern raised by another reviewer, we have further clarified the sentence as follows: "Additionally, it is important to note that human-induced atmospheric changes through land use management increase oxidant levels which can also boost natural aerosol production like biogenic SOA (Kanakidou et al., 2000)"

**Response**:
I'm uncertain why land use management would necessarily lead to increased oxidant levels. Isn't oxidant formation primarily associated with anthropogenic activities rather than land use changes?

Lines 58-70 – Not clear to me which information given here specifically refers to the UK.
Answer: To improve the readability, as also pointed out by another reviewer, we removed "in the literature" from the sentence.

**Response**: It is unclear whether the reasons mentioned after 'This is due to several reasons' specifically refer to the example provided for the UK or if their relevance is broader.

Line 249 – "Urban and suburban industrial stations were also aggregated" – what is the rational for this aggregation?
Answer: The authors changed the sentence to include the rationale as follows: "Urban, suburban and rural industrial stations, when available, were also aggregated due to their similar observed range values and trends. This consolidation was relevant as there is only 1 urban station and 1 rural industrial station."

**Response**: Given that there is only 1 urban station and 1 rural industrial station, wouldn't it be logical to mention that only suburban stations were aggregated?

Lines 312-313 – "Notably, underestimations are more pronounced during winter, suggesting a potential underestimation of road traffic cold start emissions" – For traffic? I don't see that this winter trend is significant when looking at Fig. 7 and Table 2
Answer: The authors agree that when evaluating the average of all the traffic stations, as presented in Fig.7 and Table 2, this might not be so clear to the reader. Despite this, in Table 2 we can see that the lowest MB values are in summer and the biggest in winter. This effect is more evident when looking at specific stations, e.g. stations located in Barcelona and Valencia, as presented in Figure 16.

**Response**: I cannot locate Figure 16. Could you please direct the reader to the table or figure where this trend can be observed? If this minor trend is only evident in a few stations, it would advisable to restrict the discussion to those specific stations.

Line 363 – "chemical processes affecting VOCs" - This is not clear to me. Can you specify what kind of chemical reaction could lead to an earlier VOCs morning build-up compared to the measurements? Do you imply that benzene and toluene are formed by chemical reactions which occur in the morning? Could meteorological effects/stratification of the atmosphere could play a role here too?

Answer: While uncertainties in the meteorology and emissions could play a role, the authors believe that one of the main processes affecting these results is the dry deposition. This is mainly because, for the same stations, for NO2 the model is performing well.

**Response**: The provided explanation is not convincing. Here, the discussion addresses the time of the peak rather than its magnitude. Different deposition values would likely affect the amplitude rather than the timing. Additionally, the agreement between simulated $NO_2$ and observations does not support this conclusion. Additionally, referring to 'chemical processes' in the context of dry deposition is problematic, while PBL is not a meteorological parameter."

Line 409 – "Indicating an issue related to VOC chemistry in MONARCH' – Can you explain why you necessarily attribute this issue to VOC chemistry? The same comment is relevant for toluene and benzene.

Answer: The authors gave a detailed explanation regarding this point in the previous comment regarding L363. To specify what we believe to be the main issue, we added in line 409 the reference to the dry deposition as follows: "...indicating a possible issue related to VOC processes in MONARCH (i.e. dry deposition)."

**Response**: Please refer to my response concerning the original comment on line 363.

Lines 612-613 – "suggesting that some sources are either not accurately represented in our model or are unaccounted for" - What about atmospheric chemistry effects and/or meteorological effects?

Answer: The authors agree with the reviewer that besides the emissions, there are several other sources of uncertainty affecting the results, e.g., the methods used in the measurements and the model performance reproducing the meteorological and chemical parameters. Therefore, this was revised in the manuscript and a paragraph was added as mentioned in the previous comment.

**Response**: I don't see that you addressed atmospheric chemistry and/or meteorological effects in the revised paragraph.

Line 643: "this could in the model could lead to further improvements" – please revise.

---

## Author Response (AR2)

**egusphere-2023-3145: On the uncertainty of anthropogenic aromatic VOC emissions: model evaluation and sensitivity analysis**

**Referee #2**

The revised version was a significant improvement. I have just one question regarding Appendix A. The additional reaction of Benzene is described as BENZENE + OH -> OH + 0.764*BENZRO2. However, the reference (SAPRC99 document) provided by the author (https://intra.engr.ucr.edu/~carter/pubs/s99doc.pdf) on page 271 shows that the reaction is listed as: BENZENE + HO. = #.236 HO2 + #.764 RO2-R + #.207 GLY + #.236 PHEN + #.764 DCB1 + #1.114 XC. While I understand that the tracer products won't affect the main reaction, but the list of chemical reactions should be consistent with the reference.

The authors thank the reviewer for taking the time to review the new version of the manuscript.

The mechanism extension focused solely on products of interest, without altering oxidant concentrations. This simplification was chosen to avoid tracking species not accounted for in the original mechanism. Notably, TOL + OH and XYL + OH reactions were originally present in the CB05 mechanism, not BENZ + OH. TOL and XYL reactions were extended by adding only one additional product because the others were consistent with CB05. This was not the case for BENZ where the mechanism is extended with additional simplified reactions introducing only the first-generation product of BENZ oxidation to represent the partition to SOA while neglecting the rest for the sake of simplicity. The authors agree with the reviewer that the description of the reactions should be consistent with the reference or clarify any modification introduced. So in Appendix A, we have acknowledged this by adding the additional information in the legend of BENZENE + OH reaction, as follows: "[a] Only the first-generation product that further reacts to produce SVOC is considered from the original reference.".

**Referee #4**

In general, the revised version demonstrates better structure and provides a clearer explanation for biases between modeling and observations. I recommend applying the 'tracer' method, to explicitly incorporate species in the MONARCH model. This would help address uncertainties that remain unresolved in the revised manuscript. For instance, it may shed light on the speculated effect of deposition velocity on biases between modeling and observation. While the authors have adequately addressed most of my comments, I provide in the following additional feedback on some of my original comments that require further attention.

The authors thank the reviewer for taking the time to review the new version of the manuscript. The answers to each specific comment can be found below in red.

We appreciate the suggestion of adopting the 'tracer' method to address the limitation of working with lumped species. While we acknowledge its potential, implementing it requires significant modifications to the MONARCH's chemical mechanism, making it infeasible for the current manuscript. While this is planned to be done in the future, we believe our approach allows us to advance in a better understanding of key aspects of VOCs representation in models related to emission uncertainties.

Lines 26-27 - The sentence should be clearer regarding the contribution of VOCs to SOA via those oxidants.

Answer: In response to a similar concern raised by another reviewer, we have further clarified the sentence as follows: "Additionally, it is important to note that human-induced atmospheric changes through land use management increase oxidant levels which can also boost natural aerosol production like biogenic SOA (Kanakidou et al., 2000)"

**Response:** I'm uncertain why land use management would necessarily lead to increased oxidant levels. Isn't oxidant formation primarily associated with anthropogenic activities rather than land use changes?

Answer: As highlighted by Kanakidou et al. (2000), oxidant formation is indeed primarily attributed to anthropogenic activities. When discussing changes in land use, our intention was to focus on the increase of industrial and agricultural areas, which are inherently part of anthropogenic activities. We recognize the potential for confusion and have accordingly revised the text to ensure its clarity. The revised version now reads as follows: "Additionally, it is important to note that human-induced atmospheric changes, driven by emissions from sources like industrial processes, transportation, and agriculture, increase oxidant levels, which can also enhance natural aerosol production like biogenic SOA (Kanakidou et al., 2000)."

Lines 58-70 – Not clear to me which information given here specifically refers to the UK.

**Answer:** To improve the readability, as also pointed out by another reviewer, we removed "in the literature" from the sentence.

**Response:** It is unclear whether the reasons mentioned after 'This is due to several reasons' specifically refer to the example provided for the UK or if their relevance is broader.

Answer: The authors acknowledge that the previous wording may have been misleading. To clarify that the reasons provided are general and not specific to the

example of the UK, we revised the sentence to: "This identified gap is due to several reasons:...".

Line 249 – "Urban and suburban industrial stations were also aggregated" – what is the rational for this aggregation?

**Answer:** The authors changed the sentence to include the rationale as follows: "Urban, suburban and rural industrial stations, when available, were also aggregated due to their similar observed range values and trends. This consolidation was relevant as there is only 1 urban station and 1 rural industrial station."

**Response:** Given that there is only 1 urban station and 1 rural industrial station, wouldn't it be logical to mention that only suburban stations were aggregated?

Answer: As shown in Table 1, as an example, for industrial stations measuring benzene, we have 1 urban, 8 suburban, and 1 rural. So, we aggregated all station types into "industrial" as they had similar observed range values and trends. The authors acknowledged that how it was written could be misleading, so we simplified the text to: "All industrial stations were aggregated due to their similar observed range values and trends."

Lines 312-313 – "Notably, underestimations are more pronounced during winter, suggesting a potential underestimation of road traffic cold start emissions" – For traffic? I don't see that this winter trend is significant when looking at Fig. 7 and Table 2

**Answer:** The authors agree that when evaluating the average of all the traffic stations, as presented in Fig.7 and Table 2, this might not be so clear to the reader. Despite this, in Table 2 we can see that the lowest MB values are in summer and the biggest in winter. This effect is more evident when looking at specific stations, e.g. stations located in Barcelona and Valencia, as presented in Figure 16.

**Response:** I cannot locate Figure 16. Could you please direct the reader to the table or figure where this trend can be observed? If this minor trend is only evident in a few stations, it would advisable to restrict the discussion to those specific stations.

Answer: We apologise for the error in our previous response. We intended to refer to Figure 11, which illustrates specific stations (i.e., Barcelona, and Valencia) demonstrating that underestimations are more pronounced in winter. While we use these stations as examples, this effect is not only evident for these stations but also generalised for the majority of traffic stations. As previously mentioned, Table 2 shows the lower performance of the model for traffic stations in winter compared to spring/summer. For instance, the correlation during summer is 0.70, dropping to 0.45 in winter.

Line 363 – "chemical processes affecting VOCs" - This is not clear to me. Can you specify what kind of chemical reaction could lead to an earlier VOCs morning

build-up compared to the measurements? Do you imply that benzene and toluene are formed by chemical reactions which occur in the morning? Could meteorological effects/stratification of the atmosphere could play a role here too?

**Answer:** While uncertainties in the meteorology and emissions could play a role, the authors believe that one of the main processes affecting these results is the dry deposition. This is mainly because, for the same stations, for NO2 the model is performing well.

**Response:** The provided explanation is not convincing. Here, the discussion addresses the time of the peak rather than its magnitude. Different deposition values would likely affect the amplitude rather than the timing. Additionally, the agreement between simulated $NO_2$ and observations does not support this conclusion. Additionally, referring to 'chemical processes' in the context of dry deposition is problematic, while PBL is not a meteorological parameter."

Answer: As previously mentioned, in Figure F1, we present observed and modelled values for $NO_2$ for the same stations measuring VOCs. Despite the bias, the model reproduces the morning peak (around 6:00/7:00) well. This suggests that the problem related to the earlier peak observed for the VOCs may not be directly related to atmospheric effects, as it would also be observed in modelled $NO_2$.

We acknowledge that we are currently not fully aware of the underlying problem. Regarding our initial hypothesis, we conducted a quick test (1 week) modifying the dry deposition calculation for benzene. The scheme used in our model relies on the Henry's law constant (H*) to scale the canopy resistance of a specific species to compute the overall deposition flux. For instance, for benzene, we adjusted the effective Henry's law constant (H*) from 0.16 to 0.12. In the figure below, the reference run is depicted in blue and the run with modified H* in red.

[Figure]

As presented in the figure, we see a slight modification of the diurnal profile rather than just changes in the amplitude. While we observe a small shift in the afternoon

peak, there are no changes in the morning peak, resulting in inconclusive findings. Therefore, further investigation and testing are necessary. To avoid misleading the reader with inconclusive results, we removed the reference to dry deposition and other initial hypotheses and replaced it with: "Further research is needed to understand the cause or combination of causes that could explain this shift."

Line 409 – "Indicating an issue related to VOC chemistry in MONARCH' – Can you explain why you necessarily attribute this issue to VOC chemistry? The same comment is relevant for toluene and benzene.

**Answer:** The authors gave a detailed explanation regarding this point in the previous comment regarding L363. To specify what we believe to be the main issue, we added in line 409 the reference to the dry deposition as follows: "...indicating a possible issue related to VOC processes in MONARCH (i.e. dry deposition)."

**Response:** Please refer to my response concerning the original comment on line 363.

Answer: The complete response can be found in the previous comment. This line was also changed accordingly: "At the hourly scale, the already discussed shift between observed and modelled peaks for benzene and toluene also occurs for xylene, requiring further investigation."

Lines 612-613 – "suggesting that some sources are either not accurately represented in our model or are unaccounted for" - What about atmospheric chemistry effects and/or meteorological effects?

**Answer:** The authors agree with the reviewer that besides the emissions, there are several other sources of uncertainty affecting the results, e.g., the methods used in the measurements and the model performance reproducing the meteorological and chemical parameters. Therefore, this was revised in the manuscript and a paragraph was added as mentioned in the previous comment.

**Response:** I don't see that you addressed atmospheric chemistry and/or meteorological effects in the revised paragraph.

Answer: The authors acknowledge and apologise for the oversight that, although the topic was addressed at various instances throughout the manuscript, in the specific sentence highlighted by the reviewer, this was not introduced. Consequently, we have included this information, and the revised sentence now reads: "Despite this, it is evident that the emissions are still underestimated, suggesting that some sources are either not accurately represented in our model, are unaccounted for, or the uncertainties in atmospheric chemistry and meteorological effects are contributing factors.".

Line 643: "this could in the model could lead to further improvements" – please revise.

The authors thank the reviewer for identifying the typo, this was revised as follows: "Due to the different topography of Barcelona compared to Madrid, introducing this aspect into the model could lead to further improvements in its performance."